



# Heterogeneous spatial and temporal pattern of surface elevation change and mass balance of the Patagonian icefields between 2000 and 2016

Wael Abdel Jaber[1], Helmut Rott[2,3], Dana Floricioiu[1], Jan Wuite[2], and Nuno Miranda[4]

[1]Remote Sensing Technology Institute (IMF), German Aerospace Center (DLR), Oberpfaffenhofen, Germany
[2]ENVEO IT GmbH, Innsbruck, Austria
[3]Institute of Atmospheric and Cryospheric Sciences, University of Innsbruck, Innsbruck, Austria
[4]European Space Agency (ESA) - ESRIN, Frascati, Italy

*Correspondence to:* Wael Abdel Jaber (wael.abdeljaber@dlr.de)

**Abstract.** The Northern and Southern Patagonian icefields (NPI and SPI) have been subject to accelerated retreat during the last decades with considerable variability in magnitude and timing among individual glaciers. We derive spatially detailed maps of surface elevation change (SEC) of NPI and SPI from bistatic SAR interferometry data of SRTM and TanDEM-X for two epochs, 2000–2012 and 2012–2016 and provide data on changes in surface elevation and ice volume for the individual glaciers and for the icefields at large. We apply advanced TanDEM-X processing techniques allowing to cover 90 % and 95 % of the area of NPI and 97 % and 98 % of the area of SPI for the two epochs, respectively. Particular attention is paid to precisely coregistering the DEMs, assessing and accounting for possible effects of radar signal penetration through backscatter analysis, and correcting for seasonality biases in case of deviations in repeat DEM coverage from full annual time spans. The results show a different temporal trend between the two icefields and reveal a heterogeneous spatial pattern of SEC and mass balance caused by different sensitivities in respect to direct climatic forcing and ice flow dynamics of individual glaciers. The estimated volume change rates for NPI are $-4.26 \pm 0.20 \ \mathrm{km^3\,a^{-1}}$ for epoch 1 and $-5.60 \pm 0.71 \ \mathrm{km^3\,a^{-1}}$ for epoch 2, while for SPI these are $-14.87 \pm 0.51 \ \mathrm{km^3\,a^{-1}}$ for epoch 1 and $-11.86 \pm 1.90 \ \mathrm{km^3\,a^{-1}}$ for epoch 2. This amounts to $0.047 \pm 0.005 \ \mathrm{mm\,a^{-1}}$ eustatic sea level rise for both icefields during the epoch 2000–2016. On SPI the spatial pattern of surface elevation change is more complex than on NPI and the temporal trend is less uniform. On terminus sections of the main calving glaciers of SPI temporal variations of flow velocities are a main factor for differences in SEC between the two epochs. Striking differences are observed even on adjoining glaciers, such as Upsala Glacier with decreasing mass losses associated with slowdown of flow velocity between the two epochs, contrasting with acceleration and increase of mass losses on Viedma Glacier.

## 1 Introduction

The Northern and Southern Patagonian icefields (NPI and SPI) are the largest contiguous temperate ice bodies in mid-latitudes of the southern hemisphere. They stretch from 46.5° S to 47.5° S, respectively 48.3° S to 51.6° S, along the main ridge of the southern Andes and cover areas of about 4000 km² and 13000 km² (Davies and Glasser, 2012). The perturbation of the strong and consistent westerly flow caused by the Andes leads to one of the strongest precipitation gradients on earth (Garreaud et al.,





2013). Because the icefields are located on the only significant land mass between 45° S and Antarctica, they offer unique possibilities for studying the impact of changes in southern-hemisphere westerly flow on glacier evolution and for inferring Holocene climate history from glacial evidence (Rasmussen et al., 2007; Lopez et al., 2010; Glasser et al., 2011; Davies and Glasser, 2012; Garreaud et al., 2013).

Precise, spatially detailed data on changes of glacier area and volume and on the mass balance are essential for establishing reliable relations between climate signals and glacier records in order to reconstruct the past climate and to develop accurate predictive tools of glacier response to climate change (Fernández and Mark, 2016; Marzeion et al., 2017). The dynamic adjustment of a glacier to changing external forcing does not happen instantaneously. In particular for calving glaciers the dynamic behaviour and mass balance may be largely decoupled from direct climate forcing (Benn et al., 2007). The main outlet glaciers

of the Patagonian icefields are tidewater or freshwater calving glaciers, showing heterogeneous patterns of changes in frontal position and hypsometry. This stresses the need for spatially detailed geodetic repeat observations covering different epochs in order to resolve the complex pattern of glacial responses. High resolution topographic satellite data from SAR interferometry, as employed for the work reported in this paper, provide an excellent basis for handling these issues.

There has been a general retreat of SPI and NPI glaciers since the Little Ice Age (Davies and Glasser, 2012), however with

considerable variability in magnitude and timing of retreat for individual glaciers. Only few glaciers advanced intermittently during recent decades. The most striking case is the Pio XI Glacier showing a large cumulative frontal advance since 1945, including recently a general advancing period of its southern and northern branches starting in 2000 and 2005 (Wilson et al., 2016).

Geodetic mass balance estimates of NPI and SPI have been derived from various sources. The first remote sensing based

estimates of NPI and SPI volume change and mass balance were reported by Rignot et al. (2003), comparing Shuttle Radar Topography Mission (SRTM) elevation data of February 2000 with topographic maps of 1968/1975 as well as and of 1995 on a limited area of SPI. Elevation changes measured at low elevations were fitted to a polynomial function of elevation in order to extrapolate the results to higher elevations where the maps were affected by large gaps.

The SRTM DEM was used in several studies as baseline for deriving volume change of the icefields during periods spanning

the subsequent 10 to 15 years. Willis et al. (2012a, b) analyzed the ice volume change of NPI between 2000 and 2011 and of SPI between 2000 and 2012 by comparing the SRTM DEM with time series composed of 55 DEMs for NPI and 156 DEMs for SPI which were derived from data of the Advanced Spaceborne Thermal Emission and Reflection Radiometer (ASTER) operating on the Terra satellite. The authors used all suitable ASTER scenes and applied a weighted linear-regression to the time-series of elevations on a pixel-by-pixel basis in order to obtain the elevation change for the full time spans. Willis et al.

(2012b) assumed a 2 m radar signal penetration bias for SRTM without taking into account the melting state of the surface. Accordingly they also revised their previous volume loss estimate of NPI. This underlines the importance of correct treatment of radar signal penetration in case of interferometric DEMs. Dussaillant et al. (2018) determined the NPI volume change for 2000 to 2012 with two methods, on one hand by differencing the SRTM DEM and a SPOT-5 DEM from March 2012, on the other hand by fitting pixel-based linear elevation trends over 118 DEMs calculated from ASTER stereo images acquired

between 2000 and 2012. Icefield-wide rates of volume change by both methods agree very well.



Foresta et al. (2018) exploited swath processed CryoSat-2 interferometric data to produce maps of surface elevation change over the Patagonian icefields and estimate the mass balance for six years between April 2011 and March 2017. The maps cover 46 % of the total area of NPI, and 50 % of SPI with large gaps on several main glacier termini. Relations between CryoSat-2 elevation change and surface elevation in the SRTM DEM were used to fill the gaps in the SEC maps.

The TanDEM-X (TerraSAR-X add-on for Digital Elevation Measurements) mission (shortly TDM) (Krieger et al., 2007), composed of the two formation-flying radar satellites TerraSAR-X and TanDEM-X, is operational since December 2010. The mission opened up excellent new capabilities for high resolution topographic mapping of the global land surfaces including glaciers and ice sheets. Abdel Jaber (2016) generated DEMs based on comprehensive TanDEM-X data sets, over NPI mostly acquired in summer 2014 and over SPI between March 2011 and March 2012. DEM differencing versus the SRTM DEM

yielded volume loss rates over NPI for the period 2000 to 2014 and over SPI for the period 2000 to 2011/2012. Abdel Jaber (2016) presented also detailed information on the methods used for DEM generation and calibration, on error assessment and on the analysis of radar backscatter signatures of SRTM and TDM drawing conclusions on radar signal penetration. For Jorge Montt Glacier, featuring the highest thinning rate on the icefield, the geodetic mass balance for 2011 to 2014 was also derived from TDM DEMs. Malz et al. (2018) performed DEM differencing of TDM DEMs of December 2015 versus the

SRTM DEM over SPI. For the southern part of the SPI they further differentiated the SRTM elevation versus TDM DEMs from January–March 2012, autumn 2014 and winter 2016 and computed the TDM-based surface elevation change (SEC) for the epoch January–March 2012 to December 2015. Seasonal variations of surface elevation and of radar signal penetration were not taken into account for retrieving SEC rates.

The studies cited above use different data sets and methodological approaches and cover, at least to some extent, different

epochs. This impairs the comparison of results, the evaluation of temporal trends and the analysis of commonalities and differences between the two icefields. In order to tackle these issues, we derive volume change of NPI and SPI exclusively from bistatic SAR interferometry (InSAR) data (SRTM and TDM) for two multi-annual periods, spanning from 2000 to 2012 and from 2012 to 2016. Furthermore, for the main retreating glaciers of SPI an estimate of subaqueous ice loss is provided for the period 2000–2011/2012. The generation of surface topography products and the analysis of elevation change build

upon methods developed by Abdel Jaber (2016) with various upgrades regarding TDM data selection and processing methods. Particular attention is paid to the acquisition dates of the different DEMs, applying corrections for deviations of repeat DEM coverage from full annual time spans in order to avoid seasonality biases when deriving annual SEC rates. Significant effort is dedicated to the assessment of the wetness of the snow and firn surface through a careful analysis of the backscatter of SRTM and TDM, and to modelling and quantifying different sources of uncertainty.

The paper presents the first spatially detailed analysis of surface elevation change and the derived total net mass balance for two different epochs based on the same observation technique, including a catalogue of volume change 2000 to 2012 and 2012 to 2016 for all glaciers > 2 km$^2$ of NPI and > 9 km$^2$ of SPI. The results indicate a different temporal trend between the two icefields and reveal the complex spatial pattern of SEC and mass balance as result of intricate interdependencies between direct climatic forcing and effects of ice flow dynamics.



## 2 Data

For the generation of surface elevation change rate (SECR) maps we rely exclusively on pairs of multitemporal bistatic InSAR DEMs. This technique provides wide-coverage surface elevation, overcoming issues affecting optical DEMs, such as lack of contrast on smooth snow or the presence of clouds, as well as the limited spatial coverage and resolution of altimeters. In this study we exploit data from TanDEM-X and SRTM, the sole earth observation systems equipped with single-pass radar interferometers.

### 2.1 TanDEM-X

The primary objective of the TanDEM-X mission is the generation of a global, consistent DEM with high resolution and accuracy (Krieger et al., 2007). The main payload of the twin satellites is a SAR instrument operating at X-band (9.65 GHz), capable of a swath width of 30 km in the operational Stripmap single-pol (HH) mode. The global DEM product (DLR-EOC, 2018), whose performances are analyzed in Rizzoli et al. (2017) and Wessel et al. (2018), is the result of the combination of four years of bistatic data acquisitions with different baselines and geometries. This product is hence not suitable for the derivation of surface elevation changes, nevertheless we exploited the 0.4 arcsec (~ 12 m) release as a reference DEM of the region for various processing aspects (Sect. 3), after proper editing of unreliable samples.

In this study we processed single, selected TDM bistatic raw datatakes into so-called Raw DEMs (Rossi et al., 2012; DLR-CAF, 2010) using ITP, the operational TanDEM-X processor (Breit et al., 2012; Fritz et al., 2011), in order to generate two elevation maps completely covering the icefields in the years 2012 and 2015. The TDM data selection for each coverage was based on various criteria like the reduction of temporal span and of the number of datatakes, warm seasons to minimize SAR signal penetration, small height of ambiguity (HoA) to reduce interferometric noise and similar imaging geometry. Ideally data acquisitions should be at the end of the ablation season when the surface is at its lowest, but most importantly the two coverages should be acquired at the same time of year in order to minimize seasonal changes, which can be significant on the Patagonian icefields. Since data availability restricted fulfilling the last criterium, the residual temporal gap had to be compensated.

The TDM acquisitions used to generate the two elevation maps are summarized in Tables S1 and S2 in the Supplement for NPI and SPI, respectively. The footprints of the individual Raw DEMs are shown in Fig. S1. The first elevation map is composed of descending acquisitions from austral summer 2012. An exception are the western termini of NPI where, due to data unavailability we had to rely on an acquisition from May 2011. The second coverage is achieved with descending acquisitions from December 2015 (beginning of austral summer). On part of SPI we additionally processed three acquisitions from December 2011 acquired with the same geometry of the 2012 datatakes in order to measure seasonal elevation changes during summer. Three TDM datatakes from December 2015 (scenes 6 and 7 on NPI and 13 on SPI) feature a steep look angle (< 27°) leading to increased layover.

For each Raw DEM the ITP provides additional geocoded rasters (Rossi et al., 2010; DLR-CAF, 2010) which were used in different phases of this study: height error map (HEM), uncalibrated SAR amplitude, backscattering coefficient, interferometric coherence and flag mask indicating critical areas.





## 2.2 SRTM

The SRTM (Farr et al., 2007; Rabus et al., 2003) was launched 11 February 2000 and produced in 9 days of acquisition a near-global DEM (60° N–56° S) with 1 arcsec (∼ 30 m) posting. The main payload was a bistatic C-band (5.36 GHz) SAR capable of a 225 km swath achieved applying the ScanSAR technique to four sub-swaths featuring different polarization (HH,

VV, VV, HH) and look angles between 30° to 56°. The large number of interwoven acquisitions at higher latitudes contributed to both absolute and relative accuracy as well as to reducing voids: the 9 ascending and 9 descending datatakes covering the Patagonian icefields (Seal and Rogez, 2000) are listed in Table S3. The performance of SRTM was assessed among others by Rodriguez et al. (2005); Brown et al. (2005); Carabajal and Harding (2006); Wendleder et al. (2016). The main issue is the presence of long-wavelength height errors with magnitude up to ∼ 20 m globally and spatial variation scales of hundreds to

thousands of kilometres, mainly caused by residual roll errors due to the attitude adjustment manoeuvres of the Shuttle and by the applied absolute calibration of the sub-swaths.

The NASADEM (Crippen et al., 2016) is a new version of SRTM DEM, consisting of a complete reprocessing of the raw data, with improved phase unwrapping (significantly reducing voids) and an ICESat-based calibration, tackling issues such as limited absolute vertical accuracy and long-wavelength height errors. In this study we used a provisional version of

NASADEM (NASA JPL, 2018) as the elevation map of year 2000 for both icefields. The choice was done after comparing on a vast region surrounding the Patagonian icefields the NASADEM and the SRTM ver. 3 (SRTMGL1) (NASA JPL, 2013) to the TDM global DEM rescaled to 1 arcsec. The SRTMGL1 data set, besides suffering from a vertical offset of ∼ 1 m against the reference (statistics are given in Table S4), displays a stronger presence of long-wavelength elevation and geo-location biases ($\Delta h$ images are shown in Fig. S2) and a higher RMS when compared to the NASADEM. On the icefields the differences

between the two SRTM data sets are larger on NPI and in the very south of SPI.

We furthermore retrieved the SRTM radar brightness images (SRTMIMGR) (NASA JPL, 2014) for the sub-swaths covering the icefields (Table S3) with the purpose of assessing the melting state of the glacier surface. We also used the SRTM Water Body Data (SWBD) (Farr et al., 2007) for statistical and visualization purposes.

## 2.3 Glacier outlines

We relied on the Randolph Glacier Inventory (RGI) version 6 (RGI Consortium, 2017; Pfeffer et al., 2014), which contains improved basin divides of NPI by Rivera et al. (2007). We manually updated the RGI outlines at the glacier termini (including internal rocks) using the SAR amplitude, the DEM and optical images in order to reflect the exact extent of the glaciers at the time of acquisition of each elevation map (2000, 2012, 2015).



## 3 Method and error estimation

### 3.1 Methods for SEC and mass balance

#### 3.1.1 Raw DEM processing

The use of ITP to process the single Raw DEMs allows a great degree of flexibility with respect to processing parameters and
algorithms. The beginning and end times of each scene were adapted (up to ~ 30 s total length) in order to minimize the number
of scenes and to include the widest possible ice-free terrain suitable for DEM coregistration (Sect. 3.1.2). The ruggedness of
the topography of the study region with its steep mountains and intricate water bodies poses a significant difficulty for the
ITP operational algorithms of phase unwrapping (Lachaise, 2015) and absolute height determination (Rossi et al., 2012). We
hence relied on an alternative algorithm of ITP (Lachaise and Fritz, 2016) which tackles both issues by exploiting an external
reference DEM (Sect. 2.1).

The absolute phase simulated from the reference DEM is subtracted from the interferometric phase of the data. The fringe
frequency of the differential phase is significantly lower and its unwrapping is unproblematic as long as elevation differences
versus the reference DEM are not too large (maximum half the HoA). The absolute phase of the data is then reconstructed by
summing to the unwrapped differential phase the phase simulated from the reference DEM, this way removing any influence of
the latter on the relative elevation in output. The output Raw DEM is finally obtained by geocoding in ITP the absolute phase
of the data, implicitly determining an absolute phase offset (APO) value, on which the absolute height and the across-track
position of the Raw DEM depends. ITP allows to manually update the APO value and perform a new geocoding, for instance
to fine-tune the coregistration with a reference DEM, as described in Sect. 3.1.2.

#### 3.1.2 DEM coregistration

The master and slave DEMs may be affected by vertical biases with respect to each other, these can be constant (offset),
linear (tilt) or even varying with low frequency. They can furthermore be affected by horizontal shifts causing an additional
slope- and aspect-dependent elevation bias in the SEC which couples with the vertical bias resulting in a systematic error with
high potential impact on the volume change rate estimated over large areas. To obtain two consistent TDM elevation maps,
coregistered to each other and to the SRTM DEM we coregistered the single Raw DEMs and the SRTM DEMs of NPI and SPI
to the reference DEM (Sect. 2.1).

An error in the APO of a TDM Raw DEM leads to a vertical height offset, an across-track horizontal shift and a tilt around
the master flight trajectory (in order of impact, the latter being negligible in our Raw DEMs). These three effects are solved
by fine-tuning the APO through an accurate estimation of the height offset versus the reference DEM and by repeating the
geocoding with ITP. This method assures high precision by exploiting the geometrical parameters of the SAR acquisition and
allows avoiding critical aspects of the generic coregistration problem, accurately tackled by Nuth and Kääb (2011), such as
estimation of horizontal shifts, interpolation, etc.



To estimate the height offset we manually selected a large number of calibration regions (CRs) over stable terrain around the icefields relying on the SAR amplitude, the TDM slope and optical imagery. Tall vegetation was avoided because of physical changes and varying scattering phase centre at different incidence angles and radar frequencies. The CRs were chosen to be as flat as possible in order to isolate the actual vertical height offset. Layover and shadow regions were avoided as well as water

pixels, affected by low coherence. The footprints of the CRs are visualized in Fig. S1 and their features are summarized in Table S5.

From the elevation difference $\Delta h$ between the reference DEM and the single Raw DEMs (or the SRTM DEMs of NPI and SPI) we computed on each CR with index $r$ the mean $\mu_r$, the standard deviation $\sigma_r$ and the standard error of the mean $SE_r = \frac{\sigma_r}{\sqrt{N_r}}$, where $N_r$ indicates the number of spatially uncorrelated samples on CR $r$ and was estimated through a semivariogram

analysis as described in Sect. 3.3.2 and in particular computed with Eq. (2). A height offset estimate for each DEM was obtained through the weighted average $\delta h_{\text{off}} = \frac{\sum_r \frac{\mu_r}{SE_r^2}}{\sum_r \frac{1}{SE_r^2}}$. $\delta h_{\text{off}}$ ranged in absolute value between 0 m and 1.8 m for the TDM Raw DEMs and was used to fine-tune the APO. For the NASADEM it was equal to 0.3 m and 0.1 m in absolute value on NPI and SPI, respectively, and was subtracted after checking the absence of significant horizontal shifts with respect to the TDM reference DEM at 0.4 arcsec in the proximity of the icefields.

Furthermore range and azimuth tilts caused by baseline errors (Hueso González et al., 2010) were verified and found to be negligible for all Raw DEMs. Height consistency between overlapping Raw DEMs was also checked in order to ensure a seamless elevation map.

The coregistration procedure partly compensates the crustal uplift rates due to the glacial isostatic adjustment affecting the region, characterized by rates up to 40 mm a$^{-1}$ on the plateau of SPI and decreasing with distance as reported by Dietrich et al.

20  (2010).

### 3.1.3 Seasonal correction

To derive annual rates of surface elevation change and mass balance, seasonal variations of specific mass balance should be taken into account if the time span of the repeat DEMs does not exactly match yearly intervals. The annual mean SECR of the incomplete period is commonly extrapolated to the missing temporal gap. However, in summer increased melting and

compaction of snow/firn cause significant deviations from the annual mean SECR. In order to monitor such deviations and to compensate temporal gaps in our data sets we computed on SPI a high-resolution summer SECR map by differencing three additional Raw DEMs of December 2011 versus the corresponding same-beam Raw DEM of January/March 2012, already part of the 2012 DEM mosaic (Table S2). The December 2011 data cover SPI almost completely south of 49.4° S, and the central/eastern parts of the northern section. The time span is 99 days for the two western beams, and 33 days for the

easternmost beam. The resulting daily SECR, after conservative masking of artefacts and filtering (median and smoothing), is shown in Fig. 8.

In this study the 2012–2016 SECR is affected by a significant temporal gap during summer 2015/2016, which on a short time span of 4 years makes a correction necessary. The number of missing days (53 to 70 on NPI, 53 to 103 on SPI) varies according to the combination of TDM datatakes (Table S1 and S2 and Fig. S1). In order to compensate for the missing summer



days we relied on the filtered daily SECR of summer 2011/2012. This was used pixelwise where available on SPI, in particular the 33-days beam coincides on a region where 53 to 59 days must be compensated. For the missing part of Pio XI Glacier the hypsometric mean (aggregated in 100 m elevation bins) of low-loss glaciers (Pio XI, Perito Moreno, Grey, Tyndall, HPS 13, Europa, Penguin, Guilardi) was used (green curve in Fig. S3). On the rest of SPI (except Jorge Montt Glacier) the hypsometric mean of the 99-days summer SECR was used (blue curve in Fig. S3). On NPI (except scene 1) the hypsometric green curve in Fig. S3 was used since here calving fluxes are a small component of total mass balance whereas on SPI strong dynamic downwasting leads to higher negative SECR on glacier tongues.

In the scenarios addressed above we used the 2011/2012 summer SECR to substitute the missing days of summer 2015/2016. At the synoptic station Balmaceda (45.92° S; 71.68° W) near NPI the monthly mean air temperatures of summer 2011/2012 compared to summer 2015/2016 were higher in December and January (+2 °C, +1.5 °C), lower in February (−1.5 °C) and similar in March. A similar trend was measured at Punta Arenas (53.00° S; 70.85° W), leading us to assume relatively similar mean ice ablation rates in both summers.

On the ablation area of Jorge Montt Glacier (not covered by the December 2011 scenes) it is important to consider the increased dynamic downwasting compared to the average of the icefield. To this end we separately accounted for the surface lowering related to either surface mass balance or ice flow dynamics. For the latter we assumed a constant vertical velocity throughout the year, for the former the seasonal cycle was taken into account. We added to the dynamic component the seasonal correction for ablation. This correction was estimated by using an annual linear balance gradient of $-1.37 \, \mathrm{m \, a^{-1}}$ per 100 m of elevation up to the equilibrium line (based on field measurements on Perito Moreno Glacier (Stuefer et al., 2007)) scaled with the summer-to-annual ratio of ablation rate.

We applied a similar approach to scene 1 of NPI covering mainly the termini of S. Rafael, S. Quintin and Benito glaciers (Fig. S1). Being acquired on 28 May 2011, it exceeds by 189 and 200 days the 4 year time span to the overlapping slave scenes 4 and 5 (3 and 14 December 2015), respectively. The contribution due to both dynamic imbalance and melt was obtained by scaling the uncorrected 2011–2015 SECR to the actual excess period. The deviation from the mean annual ablation rate was computed with the balance gradient of Moreno Glacier (Stuefer et al., 2007), which accounts for reduced ablation during off-summer months.

A seasonal correction was also applied on SPI to the TDM elevations of 2012 with respect to the SRTM acquisition mid-date (17 February). The effect of the correction over the 12 year time span is small, being of main relevance for scenes 6/7 (acquired on 15 March 2012) covering $\sim 6000 \, \mathrm{km^2}$ with temporal gap of 38 days. The two 99-days beams of the 2011/2012 summer SECR (which shares the 2012 acquisitions to be corrected), were used pixelwise where available. Their hypsometric mean (blue curve in Fig. S3) was used elsewhere, with a reduction by 20 % to account for the late summer season (mid-February to mid-March). This scaling factor is based on a time series of daily air temperature measurements from 1995 to 2003 near the front of Perito Moreno Glacier. A similar correction was not necessary on NPI.

Finally the correction rasters were obtained by scaling pixelwise the daily correction rate by the temporal gaps in days and are shown in Fig. S4.



### 3.1.4 Derivation of SECR maps and estimation of mass balance

Two DEM mosaics were obtained for each icefield from the main TDM coverages by means of stacking, where the most reliable scene (evaluated through the height error, the look angle and the backscattering) was prioritized on overlapping regions. The WGS84 (EPSG:4326) projection with posting 0.4 arcsec was enforced through cubic convolution on a common geographic frame. Corresponding mosaics of the additional geocoded rasters computed by ITP were also obtained, as well as the SRTM DEM and its error layer. For each icefield two SECR rasters including seasonal correction were obtained differencing the DEM mosaics for the epochs 2000–2012 (12 years) and 2012–2016 (4 years), with the end of summer as reference start/end time of each epoch. To avoid biases of the mass balance we masked-out artefacts due to phase unwrapping, layover, shadow, etc. by means of the flag mask, thresholds on the SEC, morphological operators and visual inspection. The elevation of the water surface subject to frontal retreat, usually decorrelated, was manually edited in order to correctly capture the freeboard SEC.

By multiplying the average SECR with the corresponding glacier area over elevation intervals of 50 m the altitude-dependant volume change rate (VCR) was computed. The reference elevation used for the hypsometry is the 2012 TDM DEM (small voids are filled with the global DEM), which is common to the two investigation epochs. The mass balance was computed on the entire icefields as well as on single glaciers defined by the updated RGI glaciers outlines. The maximum extent of each glacier, either at the beginning or at the end of the observation period, was used to spatially capture all changes.

We used a glacier-wide density of $900 \pm 17 \, \mathrm{kg \, m^{-3}}$ for the conversion of the VCR to mass change rate. This value is commonly used for geodetic mass balance measurements and provides traceability for comparisons with other studies (Cogley, 2009). The main mass losses on the Patagonian icefields refer to ice areas, and for the accumulation areas assumptions on changes of the vertical profiles of snow/ice density would be speculative.

## 3.2 Impact of radar penetration

A critical issue affecting InSAR-based elevation data is the penetration of the radar signal in dry snow and firn. In this case the scattering phase centre is situated below the surface, causing an elevation bias in the DEM (Dall, 2007), ranging from decimetres to metres at C- and X-band. This represents an important source of local systematic error on the SEC and consequently on the resulting total net mass balance. The penetration depth depends on the microstructure and the dielectric properties of the snowpack, which are in turn strongly dependent on the liquid water content (LWC). Several models (Tiuri et al., 1984; Mätzler, 1987) show how at C- and X-band the penetration depth drops rapidly below 0.2 m already with a LWC of approximately 0.5 $\%^{\mathrm{vol}}$. We used the backscattering coefficient $\sigma^0$ as a proxy to assess the wetness status of the snow and firn (Stiles and Ulaby, 1980; Ulaby and Stiles, 1981; Mätzler, 1987). The C- and X-band radar return from the bare rough ice of the glacier termini is dominated by surface scattering so that penetration is not an issue here.

### 3.2.1 Assessment of TanDEM-X backscatter

The TanDEM-X sensor features an absolute and relative radiometric accuracy of 0.6 dB and 0.3 dB, respectively, allowing precise measurements of backscatter. For each Raw DEM we processed with ITP the geocoded backscatter image including





the annotated noise contribution. This typically varies between $-29$ dB and $-17$ dB along the range direction and can thus have a significant impact on $\sigma^0$ of weak scatterers such as smooth wet snow. The $\sigma^0$ mosaics corresponding to the 2012, 2015 and 2011 DEMs are shown in Fig. S5. No masking of artefacts was applied.

TDM austral summer datatakes were chosen in order to increase the likelihood of imaging wet snow and firn. The mid-range look angle ($\theta_l$) ranges between $35°$ and $45°$, except for scenes 6 and 7 of NPI and scene 13 of SPI which have steeper look angles (Tables S1 and S2). The satellite overpasses were at approximately 6:00 local time (UTC$-4$h), which is generally the coldest time of the day, although the plateaus of NPI and SPI usually feature limited daily variations of air temperature due to the dense clouds and strong precipitation occurring most of the year.

On the plateau of NPI (covered by scenes 2, 5, 6 in Table S1) $\sigma^0 < -18$ dB dominates up to approximately 2300 m of altitude in 2012 and 2015 confirming high LWC on most of the surface. Above this altitude $\sigma^0 > -10$ dB can be found on limited areas (particularly in December 2015), implying the presence of dry snow. Some regions with $-15 < \sigma^0 < -11$ dB are found on scene 2 ($\theta_l = 38.4°$) at altitudes below 2000 m. Given the season and time of day, these can possibly be explained by the formation a refrozen crust layer on top of wet snow or firn, implying an offset of the scattering phase centre within the few decimetres.

The backscattering of SPI is more heterogeneous compared to NPI. The 2015 coverage features $\sigma^0 < -19$ dB revealing wet snow on large parts of the plateau (particularly on the western margin). The $\sigma^0$ of the 2012 coverage is in average higher (especially on scene 4/5 acquired at the end of March), still, we assume that most of the surface was wet (or partially refrozen) and therefore the DEM is not affected by a significant elevation bias. The December 2011 coverage displays values of $\sigma^0 < -18$ dB imputable to wet snow on most of the plateau. Some isolated regions with higher $\sigma^0$ in the southern sector have been conservatively masked out in the 2011/2012 summer SECR.

Based on the analysis of the backscatter and of the SEC maps we manually outlined regions prone to signal penetration on each DEM mosaic (Fig. S5). We assigned a potential penetration height offset to each of these polygons according to its average $\sigma^0$ and look angle, to be included in the error budget. The offsets are based on empirical observations of the relationship between $\sigma^0$ and height offset performed on multiseasonal TDM Raw DEMs of NPI (Abdel Jaber, 2016).

### 3.2.2 Assessment of SRTM backscatter

The SRTM absolute and relative radiometric accuracy nominal values are 3 dB and 1 dB, respectively (Farr et al., 2007). The SRTMIMGR product provides the radar brightness $\beta^0$ at 1 arcsec corrected for flat earth for all the sub-swaths acquired during the mission. Lacking the orbital parameters of each acquisition, we coarsely removed the flat earth correction using the mid-look angle of each sub-swath, introducing this way an error up to $\pm 0.6$ dB and computed the backscattering coefficient using the provided local incidence angle ($\theta_{\mathrm{loc}}$) mask as $\sigma^0 = \beta^0 \cdot \sin\theta_{\mathrm{loc}}$ (Abdel Jaber, 2016).

Figure S6 shows the arithmetic mean ($\bar{\sigma}^0$) and the standard deviation computed pixelwise from the sub-swath $\sigma^0$ images covering the icefields (4 to 7 stacked pixels are usually found). The measure of spread supports the interpretation of $\bar{\sigma}^0$. While $\sigma^0$ is similar for the HH and VV polarizations of the sub-swaths, variations of several dB are induced by the wide range of look angles ($30°$ to $56°$) at parity of snow conditions (Ulaby and Stiles, 1981). Furthermore temporal variations of LWC due to





changing meteorological conditions cannot be excluded during the nine days of acquisition. On the other hand variations due to the diurnal temperature cycle are unlikely given the time of the Shuttle overpasses (Table S3).

Values of $\bar{\sigma}^0 < -22$ dB denoting the presence of wet snow are found on large sections of the plateaus. In the north-western part of SPI, a west-east gradient is visible (Fig. S6). Values of $\bar{\sigma}^0$ up to $-18$ dB are found in the 1800–1900 m range (the mean

elevation of the plateaus) and up to $-16$ dB up to approximately 2300 m. These can be an indicator of wet snow with a rough surface. Above 2300 m $\bar{\sigma}^0$ increases up to $-12$ dB (excluding steep slopes in layover). Here nocturnal freezing of the upper snow layer is more likely, implying a displacement of the scattering phase centre in the order of decimetres.

The SRTM elevations are not affected by a bias due to C-band radar signal penetration on most of the icefield. Areas at higher elevations with higher likelihood of penetration have been outlined (Fig. S6) and accounted for in the error budget.

Our conclusion is supported by synoptic data measured at an AWS near the front of the Perito Moreno Glacier at 198 m. Daily average air temperatures were extrapolated using a lapse rate of $-0.65$ °C/100 m (Bippus, 2007) to around 0 °C at 1800 m a.s.l. with a slightly positive trend during the 9 days of acquisition.

### 3.3 Uncertainty of SECR and mass balance

This section reports on the estimation of the different error sources affecting the SECR maps and the mass balance computed

with the geodetic method.

### 3.3.1 Random error

The random error of each SECR sample $\sigma_{\text{SECR}}$ was computed as the square root of the quadrature sum of the random errors of the subtracted elevation samples, scaled by $\Delta t$. For TDM elevations the random error is given in the HEM raster, which contains the interferometric standard error for each sample $(x, y)$ computed assuming a normally distributed error as (Rossi

et al., 2010):

$$\sigma_h(x, y) = \sigma_\phi(x, y) \frac{h_a}{2\pi} \tag{1}$$

where $h_a$ is the height of ambiguity and $\sigma_\phi(x, y)$ is the standard deviation of the interferometric phase which depends on the coherence and on the number of looks (Lee et al., 1994). The HEM does not include any systematic error components (phase unwrapping errors, etc.), these are discussed in Sect. 3.3.3. Concerning SRTM, the NASADEM also comes with a

corresponding height error map. The contribution of the pixelwise seasonal correction (Sect. 3.1.3) was also included where performed on SPI. The resulting random error maps for the two epochs are shown in Fig. S7.

### 3.3.2 Spatial correlation and spatial averaging

The standard error (SE) of a spatial average of several SECR samples can be computed as SE $= \sigma_{\text{SECR}}/\sqrt{N}$, where $N$ is the number of uncorrelated samples. To determine $N$ the spatial correlation of the SECR maps was estimated by means of

semivariograms. Two different regions of interest (ROIs), both verifying the assumptions of first- and second-order stationarity, were selected on ice-free terrain. ROI 1 features a relatively flat topography similar to the one of the CRs (Sect. 3.1.2), ROI 2




features varied slope and aspect distribution, simulating the icefield topography. The empirical omnidirectional semivariograms obtained on the two ROIs for the TDM–SRTM and TDM–TDM SECR were furthermore fitted with an exponential model and are shown in Fig. S8. Among the model parameters reported in Table S6 the range of the semivariogram is an estimate of the correlation distance $d_c$ of the SECR map, which was conservatively increased by $\sim 40\,\%$, to account for possible higher

slopes on the averaged regions, among other factors. For the TDM–SRTM and TDM–TDM SECR maps we used, respectively, $d_c = 120$ m and $d_c = 60$ m to compute the standard error of the mean height offset on each CR (Sect. 3.1.2) and, respectively, $d_c = 200$ m and $d_c = 100$ m to compute the standard error of the mean SECR on the elevation intervals. For the estimation of $N$ the theory of geostatistics was applied as in Rolstad et al. (2009) by integrating the exponential semivariogram model (they used a spherical model) in polar coordinates over a circular integration area $A$. The assumption of a negligible nugget

(representing the uncorrelated component of the variance for the applied sampling interval) leads to the following expression for the number of uncorrelated samples $N$ within $A$:

$$N = \left[ -\frac{2}{9} \frac{A_c}{A} \left( 3\sqrt{\frac{A}{A_c}} e^{-3\sqrt{\frac{A}{A_c}}} + e^{-3\sqrt{\frac{A}{A_c}}} - 1 \right) \right]^{-1}, \qquad (2)$$

where $A_c = \pi d_c^2$ is the correlation area. Equation (2) simplifies to $N = \frac{9}{2} \frac{A}{A_c}$ for the common case where $A \gg A_c$.

### 3.3.3 Systematic errors

Systematic errors are not reduced when spatial averaging is applied, they can hence have a significant impact on the mass balance of large areas. We defined four systematic error components.

1. An error linked to the coregistration to the reference DEM (Sect. 3.1.2) was defined for each Raw DEM and for the SRTM DEM of NPI and SPI as the interquartile range (IQR) of the $\mu_r$ (mean of $\Delta h$ on CR $r$) used to estimate the height offset. This error ranges between 0.04 m and 0.3 m. The corresponding systematic error on the SECR $\varepsilon_{\mathrm{reg}}$ is obtained

pixelwise as the square root of the quadrature sum of the of master and slave DEMs, scaled by $\Delta t$ in years.

2. To account for signal penetration we used the penetration height offsets assigned to critical regions on each DEM mosaic (Sect. 3.2) as local systematic errors, ranging between 1 m and 6 m according to $\sigma^0$, look angle and radar frequency. Furthermore a bulk systematic error of 0.1 m was assigned to all remaining pixels above 1000 m a.s.l. to account for undetected regions and small offsets on refrozen upper layer of snow and firn. The systematic error $\varepsilon_{\mathrm{pen}}$ was obtained as

above.

3. An additional bulk systematic error was assigned to all glacier samples to account for unmodelled sources (e.g. residual GIA effects, residual tilts, unmasked local errors due to PU or layover, etc.). This source includes effects of the curvature-dependent SEC bias caused by the different resolution of the SRTM and TDM DEMs affecting small regions mostly at high elevation (Abdel Jaber, 2016). This additional error was set to 0.05 m for TDM, while for SRTM it was set to 0.2

m on SPI and 0.3 m on NPI to account for residual low-frequency elevation biases (Sect. 2.2). The systematic error $\varepsilon_{\mathrm{add}}$ was obtained as above.



4. To compute the systematic error linked to the seasonal correction (Sect. 3.1.3), the three aforementioned sources were estimated separately for the summer 2011/2012 SECR. Here $\varepsilon_{\text{add}}$ was increased by a factor of 1.5 to account for the different temporal coverage. All three components were summed in quadrature and further increased by a factor of 3.0 on extrapolated regions (north of SPI and NPI). A pixelwise scaling by the number of corrected days and by the appropriate $\Delta t$ in years was applied, leading to a further systematic error component on the SECR, $\varepsilon_{\text{seas}}$.

The total systematic error of each SECR sample was obtained as $\varepsilon = \sqrt{\varepsilon_{\text{reg}}^2 + \varepsilon_{\text{pen}}^2 + \varepsilon_{\text{add}}^2 + \varepsilon_{\text{seas}}^2}$. The mean values of $\varepsilon$ and of its components for the four SECR maps are reported in Table S7.

### 3.3.4 Geodetic mass balance error

The geodetic method was applied to estimate the average SECR on separate elevation bins and the corresponding volume and mass change rates. The total error of the mean SECR on elevation bin $b$ was computed by summing in quadrature the mean systematic error $\overline{\varepsilon}_b$ on bin $b$ and the standard error of the spatial average on bin $b$ (negligible compared to $\overline{\varepsilon}_b$ on large integration areas), obtained as:

$$\text{SE}_b = \sqrt{\frac{\overline{\sigma_{\text{SECR}}^2}}{N_b}}. \tag{3}$$

where $N_b$ is computed according to Eq. (2).

In the geodetic method the mean of the valid SECR samples of bin $b$ is extrapolated to the unsurveyed area of the bin. On such gaps the total error was increased by a factor of 1.5 when computing the mass balance of a single glacier basin and a by factor of 3.0 when computing an entire icefield, to account for the across-basin variability of the SEC, particularly at lower elevations.

To calculate the volume change rate a 2 % error was assigned to the glacier area obtained from the updated outlines (Sect. 2.3). This value is higher than the RGI error suggested by Pfeffer et al. (2014) and in line with the empirical findings of Paul et al. (2013). We limited the uncertainty of the density used for the volume to mass change rate conversion to a relatively small value of $\pm 17\,\text{kg m}^{-3}$ (1.9 %), imposed by the maximum density of ice. In Sect. 4 the average SECR and VCR errors estimated for each bin are reported graphically on the hypsometric plots, while the errors estimated for the entire icefields and for the individual glaciers are reported in the results tables.

## 4 Results

The SECR maps of NPI and SPI after seasonal correction are shown in Fig. 1 for the two main epochs 2000–2012 (epoch 1) and 2012–2016 (epoch 2) along with the TDM DEM mosaic of 2012 used as hypsometric reference to analyze the elevation dependence. Unsurveyed areas in the SECR maps are relatively small and geographically evenly distributed, with the exception of the eastern margin of NPI in 2012–2016 because of layover caused by the steep incidence angle of scenes 6 and 7. In Table





the SECR, the volume change rate (VCR), the mass balance and the contribution to sea level rise are specified for the entire icefields. Table 2 provides SECR and VCR for NPI glaciers larger than 2 km², Table 3 for SPI glaciers larger than 35 km² and Table S8 in the Supplement for SPI glaciers larger than 9 km². The tables report also the measured basin areas (based on the updated RGI glacier outlines) and the percentage of SECR coverage for the two epochs. The reference hypsometry and

the distribution of unsurveyed areas are shown in Figs. S10 and S11 for NPI and SPI, respectively. The altitude dependence of SECR and VCR is shown in Fig. 9 for NPI and its main glaciers and in Figure 10 for SPI and its main glaciers, while additional glaciers are reported in Figs. S12 and S13. SECR and VCR are assembled in 50 m elevation bins using the surface of the 2012 TDM DEM as reference. The SECR averaged over each glacier basin is visualized in Fig. S9 together with the 2012 TDM DEM average surface topography.

NPI shows a similar pattern of elevation change during the two epochs, with the highest rates of thinning on the lowest sections of the glacier tongues, gradually decreasing up-glacier. Equilibrium state is reached on average at about 1800 m elevation (Fig. 9). On the south-western sector of the icefield and on San Quintin Glacier the thinning rates at elevations below 1200 m are slightly higher than in the northern and eastern sectors. All glaciers with an area larger than 20 km² show volume losses during both epochs except Leones Glacier revealing a modest increase in ice volume (Table 2). The volume loss rate of

NPI increased from epoch 1 (VCR = $-4.26$ km³ a$^{-1}$) to epoch 2 (VCR = $-5.60$ km³ a$^{-1}$). The three main glaciers (San Quintin, San Rafael, Steffen) account for 50 % of the NPI volume loss during epoch 1 and for 48 % of the NPI volume loss during epoch 2. During both epochs the highest SECR at basin scale was observed on HPN 1 (VCR = $-2.50$ m a$^{-1}$ and $-3.25$ m a$^{-1}$, respectively). On all glaciers larger than 20 km², except Arco Glacier and Leones Glacier (with positive mass balance), the loss rates were higher during epoch 2. San Quintin Glacier (Fig. 2, Fig. 9) shows the highest increase in volume loss (VCR =

$-0.60$ km³ a$^{-1}$ and $-0.92$ km³ a$^{-1}$). On San Rafael Glacier the loss rate increased slightly from epoch 1 to epoch 2 (VCR = $-0.81$ km³ a$^{-1}$ and $-0.87$ km³ a$^{-1}$), but the loss pattern changed (Fig. 3, Fig. 9). On the terminus below about 800 m a.s.l. the rate of surface lowering decreased, whereas in the upper reaches loss rates became larger.

On SPI the spatial pattern of surface elevation change is more complex and the temporal trend is less uniform. Contrary to NPI, the volume loss of SPI decreased from epoch 1 (VCR = $-14.87$ km³ a$^{-1}$) to epoch 2 (VCR = $-11.86$ km³ a$^{-1}$). The three

glaciers Upsala, Jorge Montt and Viedma account for 45 % of the SPI volume loss in epoch 1 and 58 % in epoch 2. On Upsala Glacier the rate of surface lowering decreased on the terminus from epoch 1 to epoch 2 (Fig. 4) associated with a slowdown of calving velocity. The losses increased on Jorge Montt Glacier (Fig. 5) and on Viedma Glacier. Very high loss rates are observed on the lower sections of the Jorge Montt terminus, with SECR up to about $-22$ m a$^{-1}$ during epoch 1 and $-30$ m a$^{-1}$ during epoch 2. The loss rates decrease gradually up-glacier, but the main sections of the accumulation area of these glaciers, up to

elevations of 1800 m to 2000 m, were affected by downwasting during both epochs (Fig. 10).

Other glaciers with volume loss rates > 0.5 km³ a$^{-1}$ are located in the northern sector of SPI (O'Higgins, Bernardo, Greve, Tempano, Occidental), in the centre/west sector of the icefield (HPS 12), and in the south-west (Tyndall Glacier). Thinning rates up to 40 m a$^{-1}$ are observed on the terminus of HPS 12 Glacier during epoch 1. The HPS 12 terminus, flowing through a deep, narrow fjord, retreated by almost 5 km between 2000 and 2012 and by 4 km between 2012 and 2015. In epoch 2 the

maximum SECR is even higher but the exact number is not known because of gaps in the 2015 DEM due to phase unwrapping



errors. Next to HPS 12 the highest loss rates at basin scale are observed on Jorge Montt Glacier (SECR = −4.01 m a⁻¹ and −4.95 m a⁻¹ during the two epochs) and on Upsala Glacier (SECR = −3.33 m a⁻¹ and −3.04 m a⁻¹).

The only glacier with positive mass balance in both epochs is Pio XI Glacier, showing a significant increase of VCR from 0.423 km³ a⁻¹ in epoch 1 to 1.26 km³ a⁻¹ in epoch 2 (Fig. 6). SEC rates in the elevation zones up to 1500 m a.s.l. have been positive during both epochs. During epoch 1 the elevation zone between 100 m and 400 m a.s.l. was the main source for gain in ice mass. During epoch 2 an additional source of significant mass gain was the elevation zone between 1000 m and 1500 m on the ice plateau (Fig. 10).

On the western sector south of HPS 12 (49.6° S) and on the eastern sector south of Upsala Glacier (49.9° S) the average loss rates are smaller than on the northern sector, but all glaciers covering areas > 35 km² and the majority of smaller glaciers show negative SECR during epoch 1 (Tables 3, S13). On the main ice plateau the surface elevation was either stable or the SECR was slightly negative during epoch 1, becoming slightly positive during epoch 2. During epoch 2 the mass balance of several glaciers of the southern sector switched from negative to slightly positive values. However, the termini of the majority of glaciers were thinning during both epochs. The largest contributors to the SPI mass deficit in the southern sector were Tyndall Glacier (VCR = −0.79 km³ a⁻¹ during epoch 1) and Grey Glacier (VCR = −0.44 km³ a⁻¹ during epoch 1). On both glaciers the volume loss rate decreased significantly during epoch 2, on Tyndall Glacier (VCR = −0.48 km³ a⁻¹) mainly due to decrease of losses above 700 m a.s.l. (Fig. 10) and on Grey Glacier (VCR = −0.07 km³ a⁻¹) at all elevations (Fig. 7, Fig. 10). Other glaciers with distinctly different hypsometric VCR between the two epochs are Penguin, Europa, Amalia, HPS 41 (Fig. S13).

Figure 8 shows a map of daily SECR on SPI during summer 2011/2012 based on DEM differencing spanning the period 18/12/2011 to 26/3/2012 (99 days), 7/12/2011 to 15/3/2012 (99 days) and 29/12/2011 to 31/1/2012 (33 days). During summer the signal of surface lowering in the accumulation areas is mainly related to firn compaction and melting of the top snow layers (values around −0.03 m d⁻¹ are observed on the plateau) whereas in the ablation areas ice melt and dynamic downwasting (varying from glacier to glacier) are the main factors. High loss rates (SECR >∼ −0.08 m d⁻¹) refer to areas that are subject to significant dynamic thinning, such as the lower terminus of Upsala and Viedma glaciers. Average summer melt rates for ice on the lower terminus of Moreno Glacier (at 300 m altitude) are about 0.05 m d⁻¹ (Stuefer et al., 2007). On a glacier in balanced state surface lowering due to melt is in summer partly compensated by uplift due to emergence.

The reported volume change and mass balance do not include subaqueous ice volume changes. Subaqueous losses are negligible in respect to the mass change of NPI since there are no large frontal retreats on water bodies. On SPI the main glaciers, and also many smaller ones, terminate in proglacial lakes or in oceanic fjords. For the main retreating glaciers of SPI Abdel Jaber (2016) estimated the subaqueous ice VCR at −0.73 ± 0.22 km³ a⁻¹ for the period 2000 to 2011/2012. This number is obtained by measuring or estimating various parameters at the glacier front, including the glacier width, the water depth, the freeboard height on the two dates and the retreat distance. A bulk error of 30 % is assigned to the total subaqueous volume change rate, accounting also for unsurveyed glaciers. For the basal cross-section at the calving front the shape of a semi-ellipse is assumed except for four glaciers for which bathymetric data is available enabling more precise estimates. For these glaciers a bulk error of 20% is assumed for the subaqueous volume changes, amounting for the whole period to −2.80 ± 0.56 km³ on the



main front of Upsala Glacier, $-0.68 \pm 0.14$ km$^3$ on Jorge Montt Glacier, $-0.59 \pm 0.12$ km$^3$ on Tyndall Glacier and $-0.05 \pm 0.01$ km$^3$ on Ameghino Glacier. The estimated subaqueous volume changes for the period 2000–2011/2012 are reported in Table S9, together with the frontal retreat distance.

## 5 Discussion

### 5.1 Spatial and temporal pattern of surface elevation and glacier volume change

Patagonian glaciers and icefields experienced area retreat and shrinkage since the Little Ice Age which accelerated during recent decades associated with tropospheric warming (Davies and Glasser, 2012). Our estimate of mass loss for both icefields during the period 2000 to 2016 is equivalent to $0.047 \pm 0.005$ mm a$^{-1}$ eustatic sea level rise. This corresponds to 6 % of the ensemble mean contribution to sea level rise of glaciers and ice caps for the period 2005–2016 of $0.74 \pm 0.18$ mm a$^{-1}$, based on global mass balance estimates from various sources (Cazenave et al., 2018). Between epoch 1 (2000–2012) and epoch 2 (2012–2016) the rate of mass loss of SPI and NPI combined decreased by 9 % with a contrasting temporal trend between the two icefields. The topographic data show significant losses in ice mass for both icefields, as reported in previous studies, revealing major differences in mass balance and temporal trends between individual glaciers. The spatially detailed maps of SECR during the two subsequent epochs, derived from bistatic InSAR DEMs, provide a sound basis for studying the heterogeneous pattern of glacier response on NPI and SPI.

Regarding the ice bodies at large, on NPI the average loss rate increased by 31 % from epoch 1 to epoch 2 (VCR = $-4.26 \pm 0.20$ km$^3$ a$^{-1}$ and $-5.60 \pm 0.71$ km$^3$ a$^{-1}$, respectively). This was reverse on SPI where the loss rate decreased by 20 % (VCR = $-14.87 \pm 0.51$ km$^3$ a$^{-1}$ and $-11.86 \pm 1.90$ km$^3$ a$^{-1}$, respectively). Reasons for the different behaviour are temporal changes of calving velocities, in particular on SPI, as well as a north-south gradient of air temperature increase in epoch 2 compared to epoch 1. Air temperatures, based on the European Centre for Medium-Range Weather Forecasts Interim Re-Analysis (ERA-Interim) (Dee et al., 2011; Berrisford et al., 2011) show for the ERA grid point 47.25° S, 73.5° W (NPI) in 850 hPa a mean annual temperature of +1.9 °C during the period 2000–2011 and +2.3 °C during 2012–2015. The corresponding values at the grid point 50.25° S, 73.5° W (southern SPI) are: +0.7 °C and +0.8 °C. The temperature difference between the two epochs was slightly larger during the main ablation period (1 November to 31 March): air temperature +4.1 °C (summer 2000/2001 to 2011/2012) and +4.8 °C (summer 2011/2012 to 2015/2016) on the NPI grid point, +2.4 °C and +2.7 °C on southern SPI. The NCEP/NCAR Reanalysis 850 hPa mean temperature (Kalnay et al., 1996) is about 1 °C lower, but shows a similar temporal and spatial trend. Over an area extending from 72.75° W to 74.25° W, 48.00° S to 51.75° S, covering SPI, the mean annual precipitation, derived from ERA Interim data, was slightly higher (8.4 %) in epoch 1 than in epoch 2 (Langhammer, 2017).

A main factor for the increased mass losses during epoch 2 on NPI is the higher air temperature compared to epoch 1, in particular during the main ablation period. Assuming a degree-day factor of 0.7 cm d$^{-1}$ on ice areas (Stuefer et al., 2007), the melt loss for an increase of surface temperature by 0.7 °C during November to March corresponds to an additional loss of 0.74 m water equivalent per year. The hypsometric plot for the whole icefield shows changes of SECR by about $-0.7$ m a$^{-1}$ up to elevations of 1200 m a.s.l., indicating increased melt losses during epoch 2 not only on glacier termini but also on



lower sections of the NPI plateau. At higher elevations the rate of surface lowering in both epochs, including the additional contribution during epoch 2, decreases gradually with elevation, reaching balanced state at about 1800 m in epoch 1 and about 2100 m in epoch 2 (Fig. 9). On NPI surface melt is the dominating process for mass depletion. During the period 2000 to 2009 the ice export due to calving amounted to about 20 % of the annual mass depletion by surface melt (Schaefer et al., 2013).

On lower sections of the main calving glaciers temporal variations of flow velocities are a main factor for the differences in SECR during the two epochs. Flow velocities near the calving front of San Rafael Glacier have reached magnitudes in excess of 7 km a$^{-1}$ in 2007 (Willis et al., 2012a; Abdel Jaber et al., 2014), but have slowed down to 4.4 km a$^{-1}$ afterwards showing little change between 2012 and 2016. Velocities 10 km from the ice front show a temporal peak in 2005 and a decrease by about 20 % until 2014 (Mouginot and Rignot, 2015). This is reflected in the hypsometric curve of SECR, showing reduced

loss rates below 800 m elevation during epoch 2 (Figs. 3 and 9). San Quintin Glacier, the largest glacier of NPI, reaches its maximum speed of about 1 km a$^{-1}$ at a distance of 27 km from the front (Abdel Jaber et al., 2014; Mouginot and Rignot, 2015). Between 2005 and 2014 the flow velocity 1 km upstream of the front increased by about 50 %. However, this caused only a minor additional increase of surface lowering on the glacier terminus (Fig. 9) because for this glacier the ice export due to calving accounts only for a very small part of total mass turnover (Schaefer et al., 2013).

On SPI calving fluxes play a larger role for mass turnover than on NPI. This is reflected in the change of the average hypsometric curve of SECR of the icefield between the two epochs (Fig. 10). In spite of slightly higher air temperatures during epoch 2 the average rate of surface lowering decreased at elevations below 400 m. Between 400 m and 1000 m elevation the differences between the two epochs are very small. On the ice plateau, between 1000 m and 2000 m, the loss rate decreased slightly, mainly brought about by minor changes on the southern sector of the icefield. Local increase in snow accumulation

may play a role.

    For six glacier basins the VCR between the two epochs changed by more than +0.2 km$^3$ a$^{-1}$, summing up to a combined decrease of volume losses by 2.20 km$^3$ a$^{-1}$ (Table 3). The change of VCR from epoch 1 to epoch 2 amounted for Pio XI to +0.74 km$^3$ a$^{-1}$, for Grey & Dickson to +0.37 km$^3$ a$^{-1}$, for Upsala & Cono to +0.33 km$^3$ a$^{-1}$, for Tyndall to +0.30 km$^3$ a$^{-1}$, for Europa to +0.24 km$^3$ a$^{-1}$, for Penguin to +0.22 km$^3$ a$^{-1}$ . There are three glaciers with major increase of losses during epoch

2 (VCR becoming more negative by $\geq$ 0.2 km$^3$ a$^{-1}$): the change of VCR for Jorge Montt is −0.36 km$^3$ a$^{-1}$, for Viedma −0.28 km$^3$ a$^{-1}$, for Bernardo −0.20 km$^3$ a$^{-1}$.

    The behaviour of Pio XI Glacier, with frontal advance and positive mass balance since many years is opposed to the general trend of SPI glaciers. The recent frontal advance trend started at the northern section of the terminus in 2006 and at the southern section in 2000 (Wilson et al., 2016). Between 2000 and 2014 a general slowdown of velocity was observed on the central and

southern sections of the terminus, whereas the northern section accelerated. This is reflected in the map of elevation change of the two epochs, showing the strongest increase of SECR in the northern section (Fig. 6).

    On Upsala Glacier the front retreated by 4 km between 2000 and 2014 and the calving velocity decreased significantly after reaching a maximum in 2009/2010 (Abdel Jaber et al., 2012; Mouginot and Rignot, 2015). This caused a major decrease in the thinning rate of the lower terminus during epoch 2 (Fig. 4).



The hypsometric curve of Grey and Tyndall glaciers shows little change in SECR on the lower terminus close to the calving front and decreasing loss rates in the upper reaches of the terminus and the accumulation area, an indication for surface mass balance as main cause for the change in SECR (Fig. 10). This is in line with our analysis of surface velocity from TerraSAR-X data between December 2011 and August 2016 showing only modest changes near the ice front and slowdown upstream. On

Tyndall Glacier the velocity on the central flowline 0.5 km from the front was $0.35 \pm 0.02$ km a$^{-1}$ in December 2011, $0.32 \pm 0.02$ km a$^{-1}$ in October 2013 and $0.35 \pm 0.02$ km a$^{-1}$ in August 2016. On Grey Glacier the velocity on the central flowline 3 km from the front, where the glacier splits into three branches, was $0.41 \pm 0.02$ km a$^{-1}$ in December 2011, $0.40 \pm 0.02$ km a$^{-1}$ in October 2013 and $0.37 \pm 0.02$ km a$^{-1}$ in April 2016. Further upstream there is a slowdown of approximately 20 % between 2011 and 2016 on both glaciers. Weidemann et al. (2018) computed the surface mass balance of both glaciers and estimated

the calving flux as residual of mean surface mass balance and geodetic mass balance over the period 2000 to 2014, pointing out that ice loss by surface ablation exceeds ice loss by calving. On Europa and Penguin, featuring steep narrow tongues, the SECR switched from slightly negative values to slightly positive values on the ice plateau above 1000 m elevation, indicating also a change in surface mass balance.

Jorge Montt Glacier experienced a frontal retreat by 11 km between 1990 and 2011 (Rivera et al., 2012) and a further

retreat by 2 km until 2016. The hypsometric profile shows high loss rates on the terminus at elevations up to 1000 m, with losses increasing during the second epoch (Table 3, Fig. 5) associated with major flow acceleration between 2007 and 2015 (Mouginot and Rignot, 2015). On Viedma Glacier the increased mass loss during epoch 2 is caused by a major increase of the thinning rate on the glacier terminus below 1000 m elevation, an indication for dynamic downwasting. This in accordance with increasing ice flow velocities. We derived surface velocity maps of Viedma Glacier from TerraSAR-X repeat pass data

of August/September 2010 and July/August 2016, showing a two-fold velocity increase at the calving front and the signal of acceleration reaching 15 km upstream.

The heterogeneous spatial pattern of elevation change on the two icefields and its temporal evolution are results of complex interdependencies between surface mass balance, responding directly to climate change signals, and effects of flow dynamics. Differences in surface elevation change and mass balance between individual glaciers and their temporal trends are particularly

pronounced on SPI, where the calving fluxes represent a main component of mass turnover for most glaciers. The elevation dependence of the SEC reveals that ice dynamics exerts a main control on topographic change not only on the glacier tongues, but also on several sections of the main ice plateau.

## 5.2  Comparison with previous estimates

A comparison of published results on volume change rates of SPI and NPI is reported in Table S10 for different epochs between

1968 and 2017, based on various methods including differencing of optical and/or interferometric DEMs and gravimetric time series of the GRACE mission. Similar comparisons are found in Malz et al. (2018) and Foresta et al. (2018). Our results are in line with geodetic mass balance results of NPI and SPI published by other authors, which suggest an overestimation of the mass losses retrieved from gravimetric time series (Chen et al. (2007), Ivins et al. (2011) and Jacob et al. (2012), the latter referring to Patagonia in general).



Our result for NPI during epoch 1 (VCR = $-4.26 \pm 0.20$ km$^3$ a$^{-1}$) is in line with the results of Abdel Jaber (2016) for the period 2000 to 2014 ($-4.40 \pm 0.13$ km$^3$ a$^{-1}$) and Willis et al. (2012a) for 2000 to 2011 ($-4.06 \pm 0.12$ km$^3$ a$^{-1}$) based on SRTM and ASTER DEMs. Willis et al. (2012b) recomputed the estimate applying a 2 m offset to the SRTM DEM to account for signal penetration resulting in larger losses (VCR = $-4.9 \pm 0.3$ km$^3$ a$^{-1}$). This correction is not comprehensible given the wet status of the snow surface during the summer acquisition of SRTM as evident from the backscatter data (Sect. 3.2.2, and Abdel Jaber (2016)).

Our VCR for epoch 1 is slightly lower than the results of Dussaillant et al. (2018) who applied both differencing of SPOT and SRTM DEMs (VCR = $-4.55 \pm 0.41$ km$^3$ a$^{-1}$) and derivation of temporal elevation trends from ASTER DEM time series ($-4.72 \pm 0.34$ km$^3$ a$^{-1}$). Our hypsometric curve of SEC shows up to 2800 m a similar behaviour as their ASTER_trend results, although with slightly lower losses at most elevations. Above 1000 m Dussaillant et al. (2018) report 35 % and 22 % of unsurveyed area for the SPOT-SRTM analysis and ASTER_trend respectively, mostly due to the lack of contrast or presence of clouds in the optical stereo images. For the same elevation band the unsurveyed area in our 2000–2012 SECR map is 6 % on NPI. On glaciers larger than 100 km$^2$ the SEC rates with both methods applied by Dussaillant et al. (2018) agree within error bars with our results. On two medium-sized glaciers, Exploradores (86 km$^2$) and Grosse (67 km$^2$), the average SECR of their two methods differs by more than 1.0 m a$^{-1}$, their ASTER_trend being $\sim 0.8$ m a$^{-1}$ higher than our SECR and $\sim 0.6$ m a$^{-1}$ higher than those of Willis et al. (2012a).

On SPI Willis et al. (2012b) estimate a VCR of $-21.2 \pm 0.5$ km$^3$ a$^{-1}$ for the period 2000–2011, a much larger value compared to the one reported here for epoch 1 (VCR = $-14.87 \pm 0.51$ km$^3$ a$^{-1}$) and to that of Abdel Jaber (2016) for 2000–2011/2012 (VCR = $-14.59 \pm 0.37$ km$^3$ a$^{-1}$). The discrepancy largely exceeds the 10 % VCR contribution they attribute to the 2 m correction for signal penetration in the SRTM DEM.

Malz et al. (2018) present SECR maps and mass balance of SPI for the period 2000–2015 based on SRTM and several TDM DEMs of December 2015. We used the same raw data at the end of our epoch 2. They do not account for missing summer days and report a VCR of $-13.2 \pm 3.6$ km$^3$ a$^{-1}$. Scaling our VCR results over the two epochs and accounting for the missing summer days in order to cover a period of 16 years, from mid-February 2000 to mid-February 2016, we obtain $-14.2 \pm 0.9$ km$^3$ a$^{-1}$. The difference can probably be explained by the missing 48 to 76 summer days required for spanning a full period of 16 years. Applying the method of Sect. 3.1.3 for the missing summer days, we obtain an icefield-wide average SECR value of $-0.12$ m a$^{-1}$, corresponding to an additional VCR of $-1.5$ km$^3$ a$^{-1}$. For the southern sector of SPI, Malz et al. (2018) show SECR maps and hypsometric curves for the periods 2000–2012 and 2012–2015, based on the same TDM raw data used in this study (scenes 7/8 and 13/14). The absence of a correction for 53/59 summer days at the end of the 4 year period leads to lower loss rates compared to ours for epoch 2.

Average SEC rates for single glaciers are reported by Willis et al. (2012b) and Malz et al. (2018). On several main glaciers, including Bernardo, Tempano, Occidental, Greve, Chico, Europa and Guilardi glaciers, a direct comparison is not possible because of different glacier outlines. Among main glaciers with similar area our SECR estimates are in general lower than those of Willis et al. (2012b). Among glaciers > 200 km$^2$, average SEC rates deviating by more than $-1.0$ m a$^{-1}$ from our results are reported for Tyndall, Pio XI and Perito Moreno glaciers.



Foresta et al. (2018) compute the geodetic mass balance of NPI and SPI for six glaciological years between 2011 and 2017 from SEC maps using swath processed CryoSat-2 (CS2) interferometric data with sub-kilometer spatial resolution. The acquisitions dates vary spatially for different pixels. The authors explain that seasonality biases are avoided due to the regular flight path of CS2 ensuring data acquisition within each pixel at the same epochs in each glaciological year. The data coverage

is relatively poor (46 % for NPI, 50 % for SPI), in particular on lower sections of glacier tongues. Termini of several main glaciers are not covered at all and the SECR data appear to be relatively noisy. To fill data gaps hypsometric average models are applied, using the values of polynomials (degree 1 to 3) fitted to the observed hypsometric SECR. This is performed for nine sub-regions in order to obtain mass balance estimates, including large groups of glaciers (NPI, SPI-G1, SPI-G2) featuring a non-negligible variability of the SEC pattern among basins, particularly at lower elevations.

A comparison between their VCR estimates and our results for 2012–2016 is provided in Table S11. The two data sets do not cover exactly the same period. The ERA Interim 850 HPa over NPI and SPI show for the average air temperature of the main ablation period (November to March) 2011 to 2017 agreement within 0.1 °C with 2012 to 2016, suggesting similar rates of surface melt during the two epochs. Foresta et al. (2018) report a mass balance of $-6.79 \pm 1.16\,\mathrm{Gt\,a^{-1}}$ for NPI and $-14.5 \pm 1.60\,\mathrm{Gt\,a^{-1}}$ for SPI using $900 \pm 125\,\mathrm{kg\,m^{-3}}$ for volume-to-mass conversion. This corresponds to a VCR of $-7.54 \pm 0.75$

$\mathrm{km^3\,a^{-1}}$ and $-16.11 \pm 1.43\,\mathrm{km^3\,a^{-1}}$, respectively. The volume losses are significantly higher than our results for epoch 2 in all sub-regions, overall 35 % higher for NPI and 36 % for SPI. Foresta et al. (2018) show also time series of cumulative mean observed elevation change for the nine sub-regions. For NPI and five sub-regions of SPI minima in the annual elevation are found in mid-winter for some years. This is not compatible with both the annual cycle of surface mass balance and the seasonal variation of flow velocities on glacier tongues, showing high velocities in summer and low velocities in winter (Stuefer et al.,

2007; Minowa et al., 2017).

## 6   Conclusions

We reported on a detailed study focussing on the climate-sensitive Northern and Southern Patagonian icefields, where high resolution maps of surface elevation change were obtained for the epochs 2000–2012 and 2012–2016 from bistatic InSAR DEMs allowing to derive the total net mass balance of most of the glacier basins. We rely on a re-processed version of the

SRTM C-band DEM featuring improved absolute height calibration and on a series of TanDEM-X Raw DEMs, processed with a robust phase unwrapping method, leading to almost complete coverage including narrow glaciers and high altitudes. Significant effort was dedicated to reduce systematic errors, especially critical for the mass balance of vast regions: a precise coregistration of the DEMs was performed, seasonal biases in the order of metres were corrected based on a complementary TDM summer SEC map, the backscatter coefficient of all acquisitions (including SRTM) was analyzed to assess signal penet-

ration. A comprehensive uncertainty estimation including all main error sources of the SEC maps and of the mass balance was also performed.

A similar pattern of elevation change is found on NPI for the two epochs, with lowering on most of the termini and well into the main ice plateau with an increasing temporal trend. Being mass depletion mainly driven by surface melt on NPI, this trend




is likely due to higher average air temperatures during epoch 2. The estimated volume change rate increased by 31 % from $-4.26 \pm 0.20$ km$^3$ a$^{-1}$ in epoch 1 to $-5.60 \pm 0.71$ km$^3$ a$^{-1}$ in epoch 2.

On SPI the spatial pattern and the temporal trend of SECR are more complex. The volume change rates decreased by 20 % from $-14.87 \pm 0.51$ km$^3$ a$^{-1}$ during epoch 1 to $-11.86 \pm 1.90$ km$^3$ a$^{-1}$ during epoch 2. Increased trend of thinning is measured on Upsala, Jorge Montt and Viedma. Pio XI displays increased thickening up to 1500 m. On the accumulation areas south of $\sim 49.5°$ S the SECR was either stable or slightly negative during epoch 1, turning to slightly positive in epoch 2. Air temperature remained relatively stable at the south of SPI, meaning a north-south gradient was present. This, coupled with a local increase in snow accumulation may be the cause of the decreased loss rates at elevations above 1000 m. The more complex behaviour of SPI glaciers is caused by the relevance of calving fluxes as a source of mass turnover on this icefield, where the effect of ice dynamics on surface elevation changes extends to the main ice plateau. Significant frontal retreat was observed on SPI during epoch 1, our corresponding coarse estimation of subaqueous volume loss is $-0.73 \pm 0.22$ km$^3$ a$^{-1}$ for the period 2000–2011/2012.

The eustatic sea level rise contribution of both icefields, excluding subaqueous changes, was estimated to be $0.048 \pm 0.002$ mm a$^{-1}$ in epoch 1 and $0.043 \pm 0.005$ mm a$^{-1}$ in epoch 2. Behind these numbers lies a complex interplay between surface mass balance, responding directly to climate change, and ice flow dynamics, mechanisms which regulate the heterogeneous spatial pattern and temporal evolution of the SEC on NPI and SPI.

This study confirms the potential of bistatic InSAR and particularly of the TanDEM-X mission for accurate, detailed and almost gapless mapping of surface elevation changes of large icefields even for small basins and tongues. We recommend the use of TanDEM-X data—with an appropriate coregistration and care for radar signal penetration—to map SEC of all types of glaciers, as recently shown also in the northern Antarctic Peninsula (Rott et al., 2018). We hope that our results will encourage the development of remote sensing missions capable of repeated bistatic InSAR observations allowing worldwide SEC mapping and mass balance estimations with improved temporal sampling.

*Data availability.* The SECR maps will be made available upon publication of the final version on http://cryoportal.enveo.at

*Author contributions.* WAJ, HR and DF conceived and designed the study. WAJ selected and processed the TanDEM-X DEMs and produced the results and their visualization. JW processed updated TerraSAR-X ice flow velocities. HR devised the seasonal correction and performed the glaciological analysis of the results. WAJ prepared the manuscript with contributions from all co-authors.

*Competing interests.* The authors declare that they have no conflict of interest.



*Acknowledgements.* The work was supported by the European Space Agency, ESA Contract No. 4000115896/15/I-LG, High Resolution SAR Algorithms for Mass Balance and Dynamics of Calving Glaciers (SAMBA). The TanDEM-X data were made available by DLR through the projects DEM_GLAC0787, XTI_GLAC0495, XTI_GLAC6663.



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





**Table 1.** Results over NPI and SPI for the two epochs. The reported area refers to the beginning of the epoch, the coverage of the SECR map is also reported. Subaqueous ice changes are not included.

| Icefield | Period | Area $[\mathrm{km}^2]$ | Cov. [%] | Average SECR $[\mathrm{m\,a}^{-1}]$ | Volume change $[\mathrm{km}^3\,\mathrm{a}^{-1}]$ | Mass change $[\mathrm{Gt\,a}^{-1}]$ | Sea level rise $[\mu\mathrm{m\,a}^{-1}]$ |
|---|---|---|---|---|---|---|---|
| NPI | 2000–2012 | 3975.3 | 95.4 | $-1.072 \pm 0.049$ | $-4.261 \pm 0.195$ | $-3.835 \pm 0.191$ | $10.594 \pm 0.527$ |
| NPI | 2012–2016 | 3914.2 | 89.8 | $-1.431 \pm 0.181$ | $-5.602 \pm 0.709$ | $-5.042 \pm 0.645$ | $13.927 \pm 1.783$ |
| SPI | 2000–2012 | 12999.0 | 98.0 | $-1.143 \pm 0.039$ | $-14.874 \pm 0.510$ | $-13.386 \pm 0.525$ | $36.979 \pm 1.450$ |
| SPI | 2012–2016 | 12846.8 | 97.0 | $-0.923 \pm 0.148$ | $-11.860 \pm 1.896$ | $-10.674 \pm 1.719$ | $29.485 \pm 4.747$ |




**Table 2.** Average surface elevation change rate (SECR) and volume change rate (VCR) for NPI and its glaciers larger than 2 km$^2$ for the two epochs. The reported area refers to the beginning of the epoch, the coverage of the SECR map is also reported. Subaqueous ice changes are not included.

| RGI Name | 2000–2012 | | | | 2012–2016 | | | |
|---|---|---|---|---|---|---|---|---|
| | Area [km$^2$] | Cov. [%] | Average SECR [m a$^{-1}$] | Volume change [km$^3$ a$^{-1}$] | Area [km$^2$] | Cov. [%] | Average SECR [m a$^{-1}$] | Volume change [km$^3$ a$^{-1}$] |
| NPI | 3975.3 | 95.4 | $-1.072 \pm 0.049$ | $-4.2609 \pm 0.1951$ | 3914.2 | 89.8 | $-1.431 \pm 0.181$ | $-5.6018 \pm 0.7087$ |
| San Quintin | 791.7 | 99.3 | $-0.758 \pm 0.039$ | $-0.5999 \pm 0.0313$ | 773.1 | 99.0 | $-1.188 \pm 0.143$ | $-0.9182 \pm 0.1106$ |
| San Rafael | 724.6 | 98.5 | $-1.117 \pm 0.058$ | $-0.8094 \pm 0.0419$ | 717.7 | 98.5 | $-1.213 \pm 0.187$ | $-0.8706 \pm 0.1344$ |
| Steffen | 430.0 | 98.6 | $-1.669 \pm 0.050$ | $-0.7178 \pm 0.0215$ | 421.0 | 97.8 | $-2.120 \pm 0.145$ | $-0.8926 \pm 0.0609$ |
| Colonia | 291.2 | 96.5 | $-0.859 \pm 0.042$ | $-0.2502 \pm 0.0124$ | 288.0 | 85.0 | $-1.010 \pm 0.149$ | $-0.2909 \pm 0.0429$ |
| Acodado | 269.8 | 98.2 | $-1.710 \pm 0.051$ | $-0.4614 \pm 0.0137$ | 265.3 | 96.9 | $-2.367 \pm 0.146$ | $-0.6279 \pm 0.0389$ |
| Benito | 163.4 | 98.7 | $-1.500 \pm 0.047$ | $-0.2452 \pm 0.0076$ | 158.9 | 97.6 | $-1.972 \pm 0.143$ | $-0.3133 \pm 0.0228$ |
| HPN 1 | 154.0 | 98.4 | $-2.498 \pm 0.061$ | $-0.3847 \pm 0.0095$ | 149.2 | 95.2 | $-3.249 \pm 0.154$ | $-0.4847 \pm 0.0230$ |
| Nef | 128.8 | 93.5 | $-0.750 \pm 0.040$ | $-0.0966 \pm 0.0052$ | 128.4 | 76.6 | $-1.045 \pm 0.155$ | $-0.1343 \pm 0.0199$ |
| Gualas | 128.3 | 96.9 | $-1.148 \pm 0.044$ | $-0.1468 \pm 0.0056$ | 124.6 | 95.6 | $-1.543 \pm 0.163$ | $-0.1922 \pm 0.0203$ |
| Exploradores | 86.4 | 57.9 | $-0.357 \pm 0.048$ | $-0.0308 \pm 0.0042$ | 86.4 | 57.3 | $-1.187 \pm 0.168$ | $-0.1025 \pm 0.0145$ |
| Pared Norte | 84.4 | 90.7 | $-1.339 \pm 0.048$ | $-0.1130 \pm 0.0041$ | 84.2 | 57.1 | $-1.369 \pm 0.172$ | $-0.1153 \pm 0.0145$ |
| Reichert | 73.2 | 90.1 | $-0.869 \pm 0.044$ | $-0.0636 \pm 0.0032$ | 71.9 | 87.1 | $-0.931 \pm 0.147$ | $-0.0669 \pm 0.0106$ |
| Grosse | 66.8 | 78.7 | $-0.763 \pm 0.047$ | $-0.0510 \pm 0.0031$ | 66.7 | 84.6 | $-1.320 \pm 0.151$ | $-0.0880 \pm 0.0101$ |
| Leones | 66.2 | 94.4 | $0.231 \pm 0.039$ | $0.0153 \pm 0.0026$ | 66.2 | 68.2 | $0.313 \pm 0.160$ | $0.0207 \pm 0.0106$ |
| HPN 4 | 65.7 | 97.9 | $-1.237 \pm 0.045$ | $-0.0813 \pm 0.0030$ | 65.7 | 92.7 | $-1.444 \pm 0.146$ | $-0.0948 \pm 0.0096$ |
| Soler | 50.4 | 95.3 | $-0.386 \pm 0.039$ | $-0.0194 \pm 0.0020$ | 50.5 | 75.2 | $-0.493 \pm 0.155$ | $-0.0249 \pm 0.0078$ |
| Fiero | 43.2 | 57.9 | $-0.482 \pm 0.062$ | $-0.0209 \pm 0.0027$ | 41.7 | 43.6 | $-0.949 \pm 0.187$ | $-0.0395 \pm 0.0078$ |
| Cachet | 37.2 | 95.5 | $-0.254 \pm 0.040$ | $-0.0094 \pm 0.0015$ | 36.9 | 86.7 | $-0.360 \pm 0.148$ | $-0.0133 \pm 0.0055$ |
| Pared Sur | 33.5 | 92.4 | $-1.210 \pm 0.049$ | $-0.0405 \pm 0.0016$ | 33.5 | 70.3 | $-1.543 \pm 0.163$ | $-0.0517 \pm 0.0055$ |
| Fraenkel | 31.5 | 99.6 | $-0.547 \pm 0.040$ | $-0.0173 \pm 0.0013$ | 30.9 | 97.5 | $-0.855 \pm 0.141$ | $-0.0264 \pm 0.0044$ |
| Arco | 26.3 | 97.8 | $-0.326 \pm 0.040$ | $-0.0086 \pm 0.0011$ | 26.3 | 85.5 | $-0.113 \pm 0.149$ | $-0.0030 \pm 0.0039$ |
| U-3 | 17.8 | 99.2 | $0.046 \pm 0.046$ | $0.0008 \pm 0.0008$ | 17.8 | 53.5 | $-0.092 \pm 0.175$ | $-0.0016 \pm 0.0031$ |
| Strindberg | 16.9 | 99.3 | $-0.510 \pm 0.043$ | $-0.0086 \pm 0.0007$ | 16.5 | 98.4 | $-1.284 \pm 0.142$ | $-0.0212 \pm 0.0023$ |
| U-2 | 15.9 | 90.4 | $-0.031 \pm 0.052$ | $-0.0005 \pm 0.0008$ | 15.9 | 53.3 | $-0.151 \pm 0.179$ | $-0.0024 \pm 0.0028$ |
| Bayo | 13.7 | 41.9 | $-0.413 \pm 0.060$ | $-0.0057 \pm 0.0008$ | 13.7 | 27.6 | $-0.754 \pm 0.188$ | $-0.0104 \pm 0.0026$ |
| U-4 | 13.4 | 87.1 | $-1.185 \pm 0.058$ | $-0.0159 \pm 0.0008$ | 13.4 | 67.4 | $-1.414 \pm 0.167$ | $-0.0190 \pm 0.0022$ |
| Pissis | 13.4 | 92.9 | $-0.455 \pm 0.051$ | $-0.0061 \pm 0.0007$ | 13.1 | 41.8 | $-0.382 \pm 0.187$ | $-0.0050 \pm 0.0025$ |
| U-6 | 10.8 | 69.6 | $-0.332 \pm 0.067$ | $-0.0036 \pm 0.0007$ | 10.8 | 26.9 | $0.167 \pm 0.205$ | $0.0018 \pm 0.0022$ |
| Cachet Norte | 10.2 | 86.7 | $0.135 \pm 0.056$ | $0.0014 \pm 0.0006$ | 10.2 | 54.2 | $-0.280 \pm 0.174$ | $-0.0029 \pm 0.0018$ |
| Hyades | 7.7 | 80.7 | $0.735 \pm 0.082$ | $0.0056 \pm 0.0006$ | 7.7 | 3.8 | $0.935 \pm 0.491$ | $0.0072 \pm 0.0038$ |
| RGI-17.15835 | 7.6 | 80.5 | $0.097 \pm 0.086$ | $0.0007 \pm 0.0007$ | 7.6 | 30.9 | $-1.210 \pm 0.234$ | $-0.0092 \pm 0.0018$ |
| Verde | 7.0 | 78.4 | $-0.094 \pm 0.080$ | $-0.0007 \pm 0.0006$ | 6.9 | 83.9 | $-0.631 \pm 0.170$ | $-0.0044 \pm 0.0012$ |
| RGI-17.15869 | 6.4 | 99.9 | $-0.069 \pm 0.058$ | $-0.0004 \pm 0.0004$ | 6.4 | 71.6 | $-0.023 \pm 0.162$ | $-0.0001 \pm 0.0010$ |
| Cristal | 5.7 | 94.0 | $-0.091 \pm 0.059$ | $-0.0005 \pm 0.0003$ | 5.6 | 69.6 | $0.201 \pm 0.169$ | $0.0011 \pm 0.0009$ |
| U-5 | 5.6 | 92.3 | $-0.614 \pm 0.062$ | $-0.0034 \pm 0.0003$ | 5.6 | 45.4 | $-0.252 \pm 0.183$ | $-0.0014 \pm 0.0010$ |
| Andree | 6.0 | 100.0 | $-0.688 \pm 0.057$ | $-0.0041 \pm 0.0003$ | 5.4 | 99.8 | $-1.114 \pm 0.147$ | $-0.0061 \pm 0.0008$ |
| Mocho | 5.3 | 91.1 | $0.329 \pm 0.078$ | $0.0018 \pm 0.0004$ | 5.3 | 23.5 | $0.831 \pm 0.242$ | $0.0044 \pm 0.0013$ |
| RGI-17.15816 | 5.1 | 67.5 | $-0.102 \pm 0.109$ | $-0.0005 \pm 0.0006$ | 5.1 | 61.6 | $-0.444 \pm 0.194$ | $-0.0023 \pm 0.0010$ |
| RGI-17.15827 | 4.5 | 87.5 | $0.087 \pm 0.086$ | $0.0004 \pm 0.0004$ | 4.5 | 85.0 | $-0.681 \pm 0.174$ | $-0.0030 \pm 0.0008$ |
| U-7 | 3.1 | 88.7 | $0.956 \pm 0.109$ | $0.0029 \pm 0.0003$ | 3.1 | 3.2 | $1.292 \pm 0.860$ | $0.0039 \pm 0.0026$ |
| Circo | 2.9 | 62.6 | $0.044 \pm 0.113$ | $0.0001 \pm 0.0003$ | 2.9 | 66.8 | $-0.592 \pm 0.197$ | $-0.0017 \pm 0.0006$ |
| RGI-17.15812 | 2.6 | 100.0 | $0.469 \pm 0.074$ | $0.0012 \pm 0.0002$ | 2.6 | 99.3 | $-0.519 \pm 0.152$ | $-0.0014 \pm 0.0004$ |
| Mormex | 2.5 | 87.1 | $-0.552 \pm 0.125$ | $-0.0014 \pm 0.0003$ | 2.5 | 80.1 | $-1.349 \pm 0.203$ | $-0.0034 \pm 0.0005$ |
| RGI-17.15850 | 2.3 | 56.4 | $-0.515 \pm 0.176$ | $-0.0012 \pm 0.0004$ | 2.3 | 22.0 | $0.196 \pm 0.347$ | $0.0004 \pm 0.0008$ |
| RGI-17.15868 | 2.0 | 100.0 | $-0.194 \pm 0.078$ | $-0.0004 \pm 0.0002$ | 2.0 | 61.0 | $0.196 \pm 0.187$ | $0.0004 \pm 0.0004$ |





**Table 3.** Average surface elevation change rate (SECR) and volume change rate (VCR) for SPI and its glaciers larger than 35 km$^2$ for the two epochs. The reported area refers to the beginning of the epoch, the coverage of the SECR map is also reported. Subaqueous ice changes are not included. The list is continued for glaciers up to 9 km$^2$ in Table S8 in the Supplement.

| RGI Name | 2000–2012 | | | | 2012–2016 | | | |
|---|---|---|---|---|---|---|---|---|
| | Area $[\text{km}^2]$ | Cov. $[\%]$ | Average SECR $[\text{m a}^{-1}]$ | Volume change $[\text{km}^3\,\text{a}^{-1}]$ | Area $[\text{km}^2]$ | Cov. $[\%]$ | Average SECR $[\text{m a}^{-1}]$ | Volume change $[\text{km}^3\,\text{a}^{-1}]$ |
| SPI | 12999.0 | 98.0 | −1.143 ± 0.039 | −14.8738 ± 0.5105 | 12846.8 | 97.0 | −0.923 ± 0.148 | −11.8595 ± 1.8964 |
| Pio XI | 1237.6 | 99.4 | 0.420 ± 0.037 | 0.5232 ± 0.0467 | 1246.7 | 98.5 | 1.010 ± 0.166 | 1.2593 ± 0.2067 |
| Viedma | 978.8 | 98.4 | −1.987 ± 0.051 | −1.9446 ± 0.0502 | 971.3 | 98.9 | −2.291 ± 0.154 | −2.2251 ± 0.1495 |
| Upsala + Cono | 848.9 | 99.2 | −3.331 ± 0.075 | −2.8278 ± 0.0639 | 823.5 | 99.3 | −3.039 ± 0.157 | −2.5021 ± 0.1291 |
| OHiggins | 765.0 | 99.7 | −1.164 ± 0.038 | −0.8902 ± 0.0294 | 764.6 | 98.2 | −1.110 ± 0.152 | −0.8484 ± 0.1159 |
| Bernardo | 540.7 | 99.9 | −1.319 ± 0.038 | −0.7129 ± 0.0208 | 531.6 | 99.7 | −1.717 ± 0.144 | −0.9126 ± 0.0765 |
| Jorge Montt | 491.9 | 99.8 | −4.008 ± 0.085 | −1.9714 ± 0.0418 | 471.2 | 98.6 | −4.947 ± 0.158 | −2.3309 ± 0.0745 |
| Penguin | 469.8 | 99.7 | −0.117 ± 0.030 | −0.0551 ± 0.0140 | 469.8 | 99.2 | 0.359 ± 0.135 | 0.1687 ± 0.0636 |
| Greve | 428.9 | 99.8 | −1.867 ± 0.047 | −0.8007 ± 0.0202 | 419.2 | 99.4 | −2.006 ± 0.131 | −0.8410 ± 0.0548 |
| Europa | 405.9 | 99.7 | −0.276 ± 0.030 | −0.1122 ± 0.0120 | 405.8 | 99.3 | 0.322 ± 0.129 | 0.1307 ± 0.0522 |
| Tempano | 334.2 | 100.0 | −1.861 ± 0.046 | −0.6218 ± 0.0154 | 327.0 | 99.8 | −2.189 ± 0.133 | −0.7157 ± 0.0435 |
| Grey + Dickson | 310.0 | 99.8 | −1.429 ± 0.042 | −0.4430 ± 0.0131 | 304.4 | 96.4 | −0.239 ± 0.154 | −0.0726 ± 0.0468 |
| Tyndall | 311.0 | 99.3 | −2.525 ± 0.058 | −0.7851 ± 0.0179 | 302.2 | 98.4 | −1.591 ± 0.133 | −0.4809 ± 0.0402 |
| Perito Moreno | 263.5 | 96.4 | −0.246 ± 0.037 | −0.0649 ± 0.0098 | 263.5 | 92.3 | 0.379 ± 0.236 | 0.0998 ± 0.0622 |
| Chico | 239.6 | 99.6 | −1.141 ± 0.036 | −0.2735 ± 0.0087 | 238.2 | 99.4 | −1.519 ± 0.126 | −0.3619 ± 0.0301 |
| Occidental | 233.1 | 99.6 | −2.681 ± 0.060 | −0.6251 ± 0.0139 | 222.6 | 94.0 | −2.883 ± 0.136 | −0.6416 ± 0.0302 |
| HPS 13 | 213.8 | 99.9 | −0.136 ± 0.038 | −0.0291 ± 0.0081 | 213.8 | 99.7 | 0.180 ± 0.150 | 0.0384 ± 0.0320 |
| HPS 31 | 167.0 | 95.0 | −0.200 ± 0.034 | −0.0333 ± 0.0057 | 167.1 | 92.6 | 0.116 ± 0.185 | 0.0193 ± 0.0309 |
| Guilardi | 165.7 | 99.6 | −0.446 ± 0.029 | −0.0740 ± 0.0048 | 165.5 | 99.2 | 0.212 ± 0.122 | 0.0350 ± 0.0201 |
| HPS 19 | 163.2 | 99.8 | −0.036 ± 0.028 | −0.0058 ± 0.0046 | 163.2 | 99.4 | 0.313 ± 0.123 | 0.0510 ± 0.0201 |
| Lucia | 164.6 | 98.4 | −0.806 ± 0.033 | −0.1326 ± 0.0054 | 162.3 | 97.0 | −1.097 ± 0.125 | −0.1780 ± 0.0204 |
| Amalia | 163.5 | 100.0 | −0.712 ± 0.031 | −0.1164 ± 0.0051 | 161.1 | 99.7 | −0.169 ± 0.126 | −0.0273 ± 0.0204 |
| HPS 12 | 165.5 | 89.6 | −5.055 ± 0.095 | −0.8365 ± 0.0157 | 155.0 | 85.6 | | |
| HPS 34 | 153.2 | 99.1 | −0.229 ± 0.029 | −0.0351 ± 0.0045 | 153.2 | 98.2 | 0.280 ± 0.130 | 0.0429 ± 0.0199 |
| Spegazzini | 120.0 | 98.1 | −0.245 ± 0.030 | −0.0295 ± 0.0035 | 120.0 | 98.3 | 0.216 ± 0.130 | 0.0259 ± 0.0156 |
| Asia | 113.7 | 99.8 | −0.331 ± 0.029 | −0.0376 ± 0.0033 | 113.7 | 99.2 | 0.142 ± 0.122 | 0.0162 ± 0.0138 |
| Calvo | 104.3 | 98.2 | −0.250 ± 0.039 | −0.0260 ± 0.0041 | 104.3 | 93.6 | 0.528 ± 0.211 | 0.0551 ± 0.0220 |
| Bravo | 104.7 | 99.6 | −1.083 ± 0.036 | −0.1135 ± 0.0037 | 102.5 | 99.3 | −1.185 ± 0.126 | −0.1215 ± 0.0129 |
| HPS 15 | 99.3 | 99.8 | −0.081 ± 0.029 | −0.0081 ± 0.0028 | 99.3 | 99.6 | −0.023 ± 0.123 | −0.0023 ± 0.0122 |
| Ofhidro | 84.1 | 99.9 | −0.531 ± 0.030 | −0.0447 ± 0.0025 | 81.2 | 99.7 | −1.188 ± 0.124 | −0.0964 ± 0.0101 |
| Pascua | 81.9 | 98.9 | −1.740 ± 0.045 | −0.1425 ± 0.0037 | 79.6 | 96.3 | −2.263 ± 0.132 | −0.1802 ± 0.0105 |
| HPS 29 | 79.4 | 98.4 | −0.170 ± 0.030 | −0.0135 ± 0.0024 | 79.4 | 97.9 | 0.347 ± 0.124 | 0.0276 ± 0.0098 |
| HPS 41 | 79.9 | 94.6 | −1.327 ± 0.040 | −0.1061 ± 0.0032 | 73.0 | 85.9 | 0.195 ± 0.130 | 0.0143 ± 0.0095 |
| RGI-17.04863 | 75.3 | 99.6 | −2.597 ± 0.059 | −0.1954 ± 0.0045 | 71.9 | 96.6 | −1.881 ± 0.130 | −0.1353 ± 0.0094 |
| Pingo | 70.2 | 99.8 | −0.581 ± 0.030 | −0.0408 ± 0.0021 | 69.7 | 99.1 | 0.745 ± 0.123 | 0.0519 ± 0.0086 |
| HPS 28 | 68.7 | 96.0 | −0.365 ± 0.031 | −0.0251 ± 0.0021 | 68.7 | 97.4 | 0.351 ± 0.124 | 0.0241 ± 0.0085 |
| HPS 10 | 67.6 | 96.1 | −0.492 ± 0.032 | −0.0333 ± 0.0021 | 66.8 | 92.7 | −0.390 ± 0.126 | −0.0260 ± 0.0084 |
| RGI-17.04982 | 62.0 | 99.8 | −0.143 ± 0.030 | −0.0088 ± 0.0018 | 62.0 | 98.6 | 0.209 ± 0.123 | 0.0130 ± 0.0076 |
| Ameghino | 59.8 | 92.9 | −2.002 ± 0.051 | −0.1198 ± 0.0030 | 59.3 | 93.0 | −1.930 ± 0.132 | −0.1144 ± 0.0078 |
| Agassiz | 54.4 | 99.7 | −0.360 ± 0.031 | −0.0196 ± 0.0017 | 54.3 | 99.8 | 0.230 ± 0.123 | 0.0125 ± 0.0067 |
| Balmaceda | 56.7 | 94.0 | −2.736 ± 0.066 | −0.1552 ± 0.0038 | 54.0 | 96.5 | −2.389 ± 0.133 | −0.1291 ± 0.0072 |
| RGI-17.04843 | 54.4 | 97.9 | −1.813 ± 0.047 | −0.0986 ± 0.0026 | 53.5 | 90.8 | −0.971 ± 0.127 | −0.0520 ± 0.0068 |
| HPS 9 | 54.2 | 99.3 | −0.651 ± 0.032 | −0.0353 ± 0.0018 | 52.4 | 96.0 | −0.955 ± 0.126 | −0.0501 ± 0.0066 |
| HPS 38 | 52.3 | 98.1 | −2.456 ± 0.058 | −0.1284 ± 0.0030 | 50.0 | 95.6 | −1.774 ± 0.129 | −0.0888 ± 0.0065 |
| Frias | 48.6 | 61.1 | −0.667 ± 0.049 | −0.0324 ± 0.0024 | 48.6 | 96.5 | −1.366 ± 0.131 | −0.0663 ± 0.0064 |
| Onelli | 49.7 | 88.0 | −0.708 ± 0.037 | −0.0352 ± 0.0019 | 48.4 | 94.3 | −1.333 ± 0.180 | −0.0645 ± 0.0087 |
| Oriental | 47.8 | 97.9 | −0.523 ± 0.031 | −0.0250 ± 0.0015 | 47.4 | 97.8 | −0.881 ± 0.125 | −0.0418 ± 0.0059 |
| RGI-17.04904 | 46.6 | 99.4 | −0.779 ± 0.033 | −0.0363 ± 0.0016 | 45.5 | 98.1 | −0.276 ± 0.122 | −0.0126 ± 0.0056 |
| RGI-17.05363 | 41.4 | 98.4 | −0.197 ± 0.031 | −0.0082 ± 0.0013 | 41.4 | 98.0 | −0.597 ± 0.125 | −0.0247 ± 0.0052 |
| Mayo | 41.4 | 90.4 | −0.506 ± 0.034 | −0.0209 ± 0.0014 | 41.4 | 95.2 | −0.404 ± 0.126 | −0.0167 ± 0.0052 |
| RGI-17.04963 | 40.4 | 99.4 | −0.088 ± 0.031 | −0.0036 ± 0.0012 | 40.4 | 98.6 | 0.336 ± 0.124 | 0.0136 ± 0.0050 |
| RGI-17.04915 | 38.9 | 96.6 | −0.236 ± 0.034 | −0.0092 ± 0.0013 | 38.9 | 95.5 | 0.310 ± 0.127 | 0.0121 ± 0.0049 |
| Melizo Sur | 37.5 | 92.6 | −0.221 ± 0.034 | −0.0083 ± 0.0013 | 37.0 | 91.6 | −0.401 ± 0.130 | −0.0148 ± 0.0048 |
| HPS 8 | 35.5 | 99.8 | −0.369 ± 0.033 | −0.0131 ± 0.0012 | 35.4 | 98.8 | −0.625 ± 0.124 | −0.0221 ± 0.0044 |





**Figure 1.** SECR maps of NPI and SPI for the two epochs (a) 2000–2012 and (b) 2012–2016. Unsurveyed areas are marked in yellow. (c) The TDM DEM of 2012 used as reference for the geodetic mass balance.



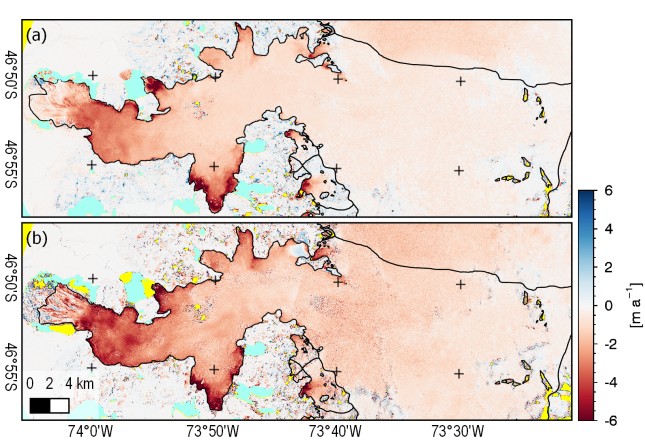

**Figure 2.** SECR of S. Quintin: (a) 2000–2012 , (b) 2012–2016.



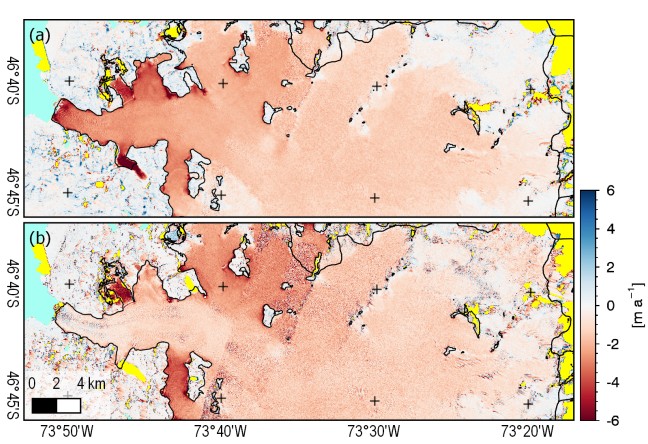

**Figure 3.** SECR of S. Rafael: (a) 2000–2012, (b) 2012–2016.





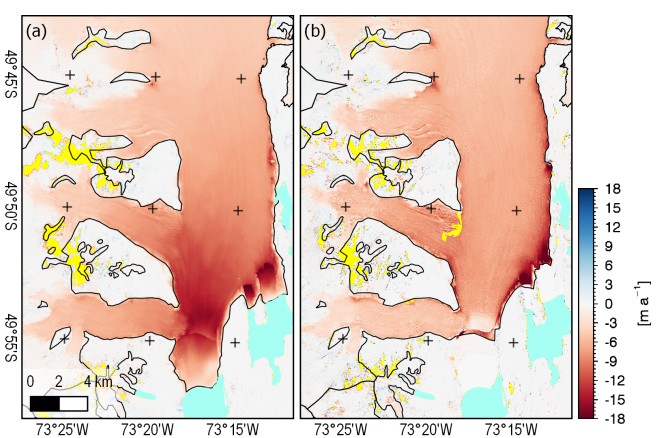

**Figure 4.** SECR of Upsala: (a) 2000–2012, (b) 2012–2016.



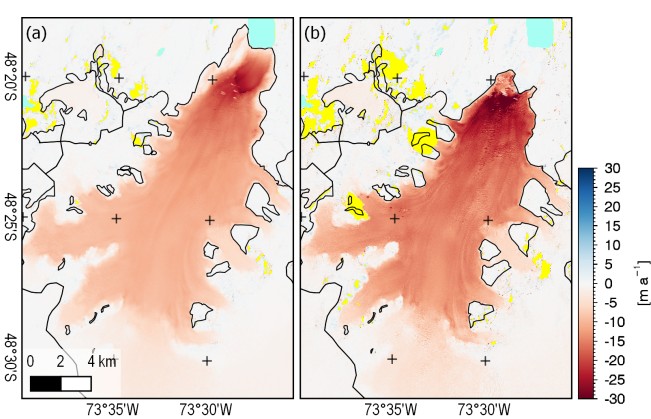

**Figure 5.** SECR of Jorge Montt: (a) 2000–2012 , (b) 2012–2016.




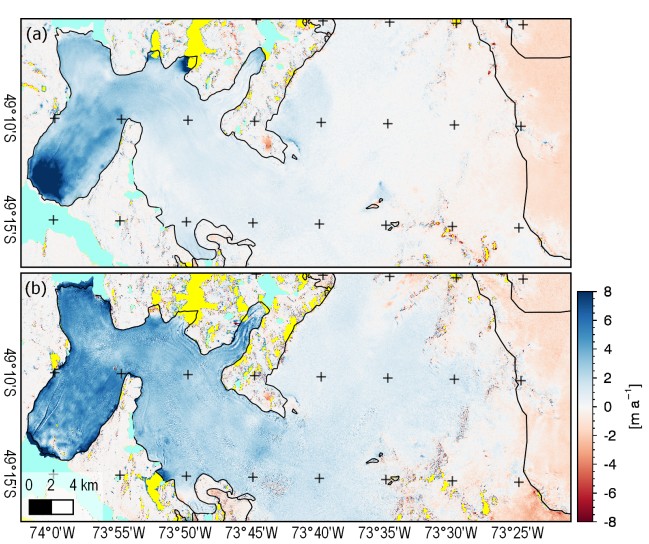

**Figure 6.** SECR of Pio XI: (a) 2000–2012 , (b) 2012–2016.



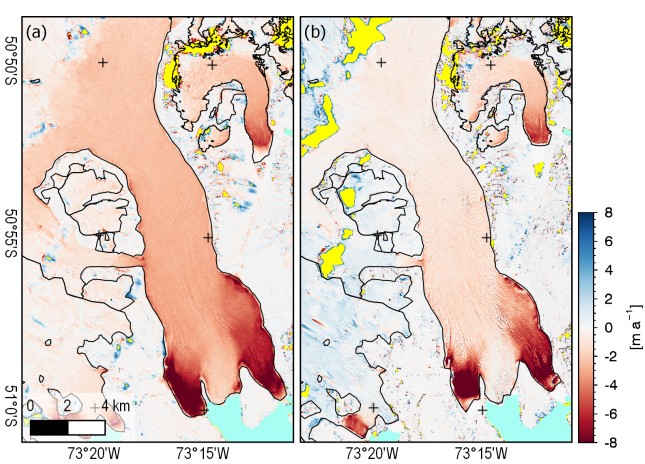

**Figure 7.** SECR of Grey: (a) 2000–2012 , (b) 2012–2016.





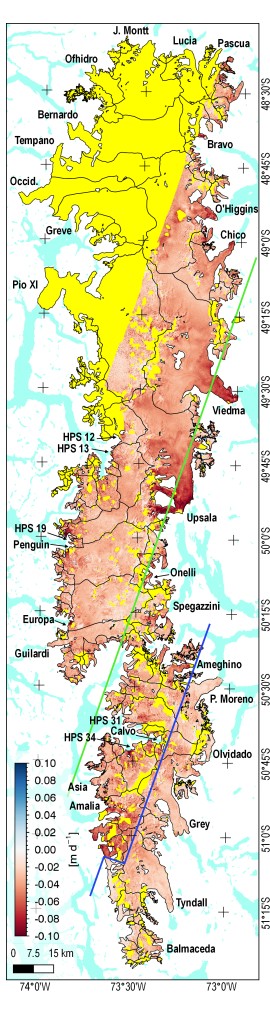

**Figure 8.** Daily SECR of SPI during summer 2011/2012. Acquisition dates north of green line: 18/12–26/3 (time span: 99 days), between green and blue lines: 7/12–15/3 (99 days), south of blue line: 29/12–31/1 (33 days). Unsurveyed areas are marked in yellow.



**Figure 9.** Surface elevation, volume and mass change rates (SECR, VCR, MCR) versus altitude in 50 m intervals for NPI and its main glaciers for epochs 2000–2012 (red) and 2012–2016 (blue). The hypsometric curve of 2011/2012 is shown in grey.



**Figure 10.** Surface elevation, volume and mass change rates (SECR, VCR, MCR) versus altitude in 50 m intervals for SPI and its main glaciers for epochs 2000–2012 (red) and 2012–2016 (blue). The hypsometric curve of 2012 is shown in grey.