# Peer review of "Heterogeneous spatial and temporal pattern of surface elevation change and mass balance of the Patagonian icefields between 2000 and 2016"

_The Cryosphere, 2018_

## Referee Comment (RC1) · Anonymous Referee #1 · 9 Jan 2019

General comment

This manuscript applies advanced SAR processing techniques in order to derive spatially detailed maps of surface elevation change (SEC) of the Northern and Southern Patagonian icefields, NPI and SPI respectively. The text is well written, the figures are of good quality, the tables are clear and informative, the topic is of high interest and the results are very interesting, confirming previous studies showing as a whole a strong negative ice volume change with high spatial variability. The authors analyse in detail different sources of errors and gives precise estimations of uncertainties for every studied glacier, different data sets and analysed periods. The results are not totally

novel, since the data sets employed in this manuscript were recently used in a paper published by Malz et al., (2018) with some variations in dates, error assessments and study area. Unfortunately, the results are not totally comparable between them since the glacier basins and dates are not the same. These differences preclude a precise estimation of discrepancies, but in general the results are statistically similar. The main contribution of Abdel Jaber et al, is their claimed much smaller uncertainties due to several correction that Malz et al didn't applied. After considering the analysis performed by Abdel Jaber et al, I think the error assessment is much more rigorous, effectively addressing many error sources of the data sets, but it is too ambitious when trying to extrapolate parameters from single stations/glaciers for correcting some issues related to the whole icefields. In those cases, is better to live with higher uncertainties and not adding more doubts as I think were added when using Perito Moreno as model for altitudinal gradients for example.

Thanks to the detailed error analysis, this manuscript provides a state-of-the art estimation of surface elevation change of both Patagonia Icefields. Unfortunately, the last data set of 2015 does not come from the end of ablation season, therefore, the authors applied a seasonal corrections to the derived surface elevation changes in order to provide estimates that correspond to full seasonal cycle. This seems to be a weak point of the study design. I guess this was caused by the availability of TanDEM-X imagery, however, employing a more recent datatake from the end of ablation season (later than 2015/16), would have considerably strengthen the importance of this contribution. Assuming that the surface elevation changes in summers 2011/2012 and 2015/2016 are equal, based on similarity of monthly mean air temperature records in some neighbouring weather stations is very arguable.

The other weak aspects of the manuscript are the assumptions regarding altitudinal gradients only supported by Perito Moreno glacier data. This glacier cannot be considered representative for the entire icefield due to extreme longitudinal gradients of climate and mass balance associated with orographic barrier of the Patagonian Andes.

[Figure]

I would be keen to see maps of systematic error similar to Fig S7 that shows random error. This is especially important for some major outlet glaciers of SPI (Jorge Montt, Pio XI, O'Higgins, Viedma and Upsala) where seasonal correction was in the order of several meters (Fig. S4b).

This work is largely based on the results and methods reported by Abdel Jaber (2016) PhD thesis. In order to avoid undesirable repetitions, I guess the thesis can be quoted only a couple of times, and then assuming that the results are the one obtained in this manuscript.

In synthesis, I think this manuscript is highly valuable and fits very well with the aim and scope of the journal.

Some specific comments:

P1 L20: ... and ..., respectively

P2 L9: any other reference? this is only review

P3 L8-9: You frequently refer to Abdel Jaber (2016), how different is its work from this submission? I presume is roughly speaking the same.

P3 L23-24: Later you refer to Abdel Jaber (2016) as a source for subaqueous ice loss estimates

P5 L15-20: Crippen et al. (2016) provided only a general description of NASADEM, and as far as I now, its performance has not been thoroughly compared with SRTMGL1. This I guess is one of main differences of this work compared to a very similar study by Malz et al. (2018) - how does it impact your final results? Were there really less voids compared to SRTMGL1?

P6 L25: How different were your glacier outlines from those used by Malz et al? The Randolph inventory is known to have problems in many places. Maybe you can discuss about this.

P7 L10: There is a problem when quoting equations along the whole manuscript. Only Eq. 2 is mentioned in the text and some equations are between lines without label numbers, don't helping in the fluent reading process.

You refer here to Eq. (2) well before introducing it (and before Eq. (1))

P8 L20. The temperature and balance altitudinal gradients in the SPI are highly different between east-west margins or northern - southern parts etc. Maybe you can check a recent paper by Bravo et al in JGR (DOI: 10.1029/2018JD028857) and comment on this.

P8 23-36: Can you please clarify this part? For example Perito Moreno and Jorge Montt have very different climatic setting (Lenaerts et al., 2014), I wonder if Perito Moreno is a best choice for a reference in this case.

P8 L30-32: Again, issue of transferability of parameters of Perito Moreno to entire SPI. Why was it not necessary for NPI, because of the day of datatake, I guess?

P8 L36-37; Fig S4: There are sharp boundaries between zones corresponding to different timespans, do they propagate to the final product introducing discontinuities?

P9 L19: Uncertainty bound on glacier-wide density seems to be too low. Cogley (2009) refer to Sapiano et al. (1998) 6% estimate as reasonable. In similar work, Malz et al. (2018) provide three scenarios of different densities, it is their main source of uncertainty for the final results.

P10 L7-10: Please back it up with some reference

P10 L24: Finally, what exactly was the criterion for masking regions prone to penetration? Was it only manually outlined based on expert knowledge?

P11 L8-9: See comment above

P11 L11: Bippus (2007) assumed this lapse-rate for summer season on Perito Moreno, however as far as I know this value was not based on measurements. Additionally, she

accounted for an off-glacier location of AWS, resulting in additional temperature offset. Maybe you can compare your numbers with Bravo et al 2019

P12 L26: Is 0.1 m based on literature?

P13 L1-5: I think that the error linked to the seasonal correction may be underestimated as it does not seem to cover all uncertainties related to the transferability of hypsometric averages shown in Fig. 8 (see previous comments).

P15 L26: It that is true than your seasonal correction should use lower density in accumulation area.

P15 L30: Perito Moreno glacier

P16 L28: Results were recently published in Frontiers - Langhammer et al. (2018)

P16 L37: Again, I doubt that Perito Moreno is representative for entire SPI and SPI, Steufer (2007)

P17 L26: This issue is a critical factor in the whole analysis of the elevation changes in the high plateau of the icefields. We know that the accumulation is extremely high, an in between few days you can have huge accumulation events. I think this high temporal variation of snow fall must be taken into account. See Schwikowski et al 2013 for snow accumulation on the SPI.

P18 9? What analysis? It is missing in methods and results sections. Maybe you wanted to quote Abdel jabber 2012? P18 L25: See comment above P33 Fig.4: Why is there a sharp transition in the terminal part of the glacier on panel a? Frontal retreat I guess?

New references mentioned in this review:

Bravo, C. et al (2019) Air Temperature Characteristics, Distribution and Impact on Modeled Ablation for the South Patagonia Icefield. JGR, DOI: 10.1029/2018JD028857

Langhamer, L., Sauter, T., & Mayr, G. J. (2018). Lagrangian Detection of Moisture Sources for the Southern Patagonia Icefield (1979-2017). Frontiers in Earth Science, https://doi.org/10.3389/feart.2018.00219

Lenaerts, J. T., Van Den Broeke, M. R., van Wessem, J. M., van de Berg, W. J., van Meijgaard, E., van Ulft, L. H., & Schaefer, M. (2014). Extreme precipitation and climate gradients in Patagonia revealed by high-resolution regional atmospheric climate modeling. Journal of Climate, 27(12), 4607-4621.

Sapiano, J.J., W.D. Harrison and K.A. Echelmeyer. 1998. Elevation, volume and terminus changes of nine glaciers in North America. Journal of Glaciology, 44(146), 119-135.

Schwikowski, M., M. Schläppi, P. Santibañez, A. Rivera and Casassa G. (2013): "Net accumulation rates derived from ice core stable isotope records of Pío XI glacier, Southern Patagonia Icefield". The Cryosphere 7, 1635-1644. doi.org/10.5194/tc-7-1635-2013

---

## Referee Comment (RC2) · Anonymous Referee #2 · 22 Jan 2019

General Comments: The study presented by Wael Abdel Jaber and co-authors is an overview of surface elevation change rate (SECR) and geodetic mass balance (MB) values for the Southern Patagonia Icefield (SPI) and Northern Patagonia Icefield (NPI) for the two epochs 2000-2012 and 2012-2016. The results are calculated on the entire icefield as well as on glacier basis, mean SECR and volume change rates (VCR) are listed in a table including observed area and error budget. For most important glaciers the hypsometric distribution of those variables is depicted in graphs. The study provides a detailed description of the error analysis and several steps to correct for biases and penetration and ablation uncertainties. The language is correct and understandable. The subject is of high interest to the community, the method and study areas are

not completely novel. In the last years, there have been publications covering the study area with the same topic (Foresta, Dussaillant, Malz, Abdel Jaber himself), but partly using different approaches. This new study cites and discusses those adequately. I recommend to add the recent work of Braun et al. (2019) which also includes SPI and NPI, but only covers the first observation period (2000-2011/15). The authors point out two aspect as main progress to previous studies: 1) The comprehensive and simultaneous observation of both icefields at two epochs. 2) The variety of corrections and assumptions made to guarantee a precise observation of SECR and following products. The line of argumentation is clear as far as (1) is concerned and thus I support publication in TC. Nevertheless, concerning (2), revisions should be performed to significantly improve the traceability of results and assure the validity of some of the applied steps described in the method section before publication. Methods: The utilization of several thresholds or distinct values is not always transparently explained. At some decisive points, it remains vague if the method or decision follows own reasoning, own previous work or an external reference (cf. specific comments) The correction for the observation date in epoch 2, for not being at the end of ablation period, is an unprecedented venture. However, it forms also a weak point of the study. In the reviewer's opinion, the error induced to the SECR (Epoch 2 – Epoch1) by this step is not adequately represented by the mapped datasets nor is it transparently addressed as error contribution in the text. Moreover, an interpolation of missing areas based on only two weather stations and adjusted to sparse hypsometrical patterns has to be regarded rather experimental compared to the robust methodology used for the rest of the study. It is hard to judge the validity of the seasonal correction. A $\Delta h$ map outside the icefields and the unfiltered dataset $\Delta h$ could help justifying, at least for the observed parts(cf. specific comments) The error indicated for SECR is spectacularly low in this paper. Although there is a section explaining the calculation it is not totally clear, why a DEM comparison could come up with such low elevation error budget. It appears, the systematic error budget, as the main contributor, is calculated partly in favor of a small total error. Some steps along this path should be under discussion or
described in more detail for traceability (cf. specific comments). Structure: The work is based on the PhD thesis of Abdel Jaber (2016). However, since it is sometimes difficult to follow what is actually new in contrast to what was already in place, that presents the reader with challenges. A clear line between parts that were newly implemented and those that were adopted needs to be drawn by the authors. I recommend that the authors revise the methods and result section with regard to this aspect to make the paper a full stand-alone document. This also concerns the length of some descriptions that could be kept more concise for this paper, with reference to the thesis (or other original source).

Specific comments:

P 6 l27 ...(in order of impact, the latter being negligible in our Raw DEMs)." This and further statements could be corroborated by a similar Figure as Fig S 2 for SRTM-TDM, displaying same $\Delta$h for outside the icefields for SRTM-TDM(Ep1) and TDM(Ep1)-TDM(Ep2).

P 7 ll3 -13 The weighted averaging of the offset values leaves the question if a spatial pattern was analysed and fitted by an offset function. A simple averaging could lead to regional maladjustment, if the sign / magnitude of the offset is a function of geographic position (tilted dataset, described in this manuscript p6 l20). For the precision of the applied method a mapped $\Delta$h (cf. comment to P6 ll26) could be convincing.

P7 ll13 How is the absence of horizontal shifts checked? The detection is slope dependent (cf. Nuth and Kääb (2011)), thus cannot be efficiently performed on an area without slope as the CRs (avr. Slope below 4°).

p7 ll23 Please provide reference

P 7 ll30 What kind of filtering was applied? It would be interesting to see the original dataset and a $\Delta$h map outside the icefields.

P8 ll10 What does similar mean here? +-0°C? Please add a number for consistency.

P7 ll32 -p8 37 A comprehensive series of comments concerning the temperature variability and spatially variable ablation patterns resulting in a rather speculative adjustment in the seasonal correction section is given by referee #1. I agree on those.

P8 ll28-32 Please explain the justification of 20% reduction in correlation to a temperature value. Based on what assumption does it translate into a percentage?

P9 ll8 Can you please add more information to increase reproducibility when data gets available: what threshold on SEC values? What morphological operators?

P9 ll16-19 Where are the 17 kg m$^{-3}$ uncertainty taken from? Citation of Cogley et al. (2009) is misleading here, because reader would expect a reference for the density uncertainty. I found it to be mentioned in Abdel Jaber (2016), but it seems to be taken from Gardner et al. (2012) – this is not referenced here. Anyway, why using this value when recent large area studies like Brun et al. (2017), Dussaillant et al. (2018), Malz et al. (2018) use 60 kg m-3? Choosing that latter value would lead to comparable error budget.

P9 ll 31 Please provide reference

p10 ll21-24 and P11 ll8 Why manual outlines? What is the decision to delimit these areas based on? If that information can be found in Abdel Jaber (2016) it should be indicated (or the original study it referes to).

P11 ll3-7 Is any of the values mentioned in these paragraphs used for determining the outlines? What is the interpretation of the sigma0 ranges based on? Abdel Jaber (2016) / other? Please reference it.

P11ll 17 First sentence would be well supported by a formula. Is the HEM for the TDM elevations calculated by the phase difference to the interferometric phase of 12 m TDM products? Is it always TDM 12m as a reference (also in 3.3.3 (1))? It is mentioned once briefly in 2.1, but I think I should be emphasised there, that it is especially used as reference for elevation error assessment.

P11 ll25: How was it included? Add some mathematical explanation of the error propagation through seasonal correction. Is it sqrt($\sigma$t1$^2$ + $\sigma$t2$^2$ +$\sigma$seas$^2$) for each pixel?

P12 ll14 Enhance precise and illustrative explanation to this whole section 3.3.3. The reader is interested how exactly the systematic error is calculated, for it is key to the low elevation error budget presented. Please provide formulas to enhance comprehensibility. That could spare some explanatory text passages, that are less illustrative.

p 12 ll14 Is the IQR of the areas that were adjusted (CR, calibration) addressed as the measure of error (validation) on each DEM? I do not agree with this method from a scientific perspective. On top, choosing the IQR reduces or eliminate slope dependent effects (avr. Slope below 4°, IQR slope?). But on glacier these are present for sure, so they are a source of systematic error to be addressed in the budget. It would be more reliable if validation is performed on the entire DEM (glaciers excluded), but assessed with regard to absolute elevation and slope.

P12 ll21 Why 1 – 6 m? Reference, calculation or explanation for decision should be provided. Where does that assumption 1000 m.a.s.l come from? Please provide reference.

P13 ll1 According to this paragraph: for interpolated seasonal correction, the last epsilon term should dominate the quadrature sum and thus the total SECR error ,if I understand correctly. What does 'increase by a factor of three' mean in this context? Times 3 (*3) ? I compared SECR uncertainty value for extrapolated glaciers (e.g. Jorge Montt, Bernardo, Tempano) in Tab. 3 with values for not extrapolated glaciers. First ones are not near triple of latter. And they should even be higher than triple, following this paragraph: scaling by year (divided by 0.27.. for 99 days for example) is performed as well as a *1.5 increase for the timespan difference. Please explain where I've gone wrong and/or revise the explanations in this paragraph.

P13 ll9 A formula containing the total SECR error would be helpful for traceability. Is it $\delta$SECR = sqrt($\varepsilon$b$^2$ + SE$^2$). Just to make sure I got the method correctly and the

comment above (ll1) is justified.

P13 ll15 I would assume a factor of 3 to be very low for the icefields concerning a factor 5 was applied e.g. by Brun et al. 2017 in High Mountain Asia, whereas the variability of SECR patterns in the icefields (especially SPI) is rather high.

P13 ll19 A formula for the complete error propagation throughout mass balance computation would be appropriate.

P15 ll19 The processes described should be perfectly correct. However, I doubt the values found through the seasonal correction analysis are able to significantly support this interpretation. As mentioned, I assume this daily SECR as a study design feature hard to accept. Also, a precise description of the method that smoothed the SECR field in Fig 8 would be of interest– or even better a display of the original data (SECR field). If it is clearly shown, that the process introduces more precision to the data, than it introduces measurement/ interpolation uncertainty (also regarding comments to 3.3.3) I am willing to accept it. So far, I find it difficult to support it.

P15 ll28 For the subaqueous loss Abdel Jaber (2016) is referenced. But for the basal cross-sections an original source should be cited.

P18 ll 9 It is unclear here if that paragraph refers to previous work (Abdel Jaber 2016) or a different publication. Any citation would help. Also I would suggest a reference to the Figures displaying those datasets (provided in the supplement if it is own work)

Technical Corrections:

p4 l1 'Method and error estimation'

P7 l1 Check formula. This way it says $\delta$hoff is equal $\delta$hoff times the factor.

Also the distinction, when formulas are a) formatted as objects to be numbered b) written as part of continuous text c) omitted, but have a text description instead is not clearly structured. This should be reconsidered thoroughly.

P14 l 'Figs' Fig. /Figure consistency Check throughout the text, also Table /Tab.

---

## Author Comment (AC1) · 23 Jan 2019

**Interactive comment on "Heterogeneous spatial and temporal pattern of surface elevation change and mass balance of the Patagonian icefields between 2000 and 2016"**

*Authors*: Abdel Jaber, W., Rott, H., Floricioiu, D., Wuite, J., Miranda, N.

The Cryosphere Discuss., https://doi.org/10.5194/tc-2018-258

**Authors' response to Anonymous Referee #1**

Referee comments are shown in **black**, our response in **blue**. Line numbers refer to the manuscript version (pdf) of 23 November 2018.

**General comments:**

**Comment:** This manuscript applies advanced SAR processing techniques in order to derive spatially detailed maps of surface elevation change (SEC) of the Northern and Southern Patagonian icefields, NPI and SPI respectively. The text is well written, the figures are of good quality, the tables are clear and informative, the topic is of high interest and the results are very interesting, confirming previous studies showing as a whole a strong negative ice volume change with high spatial variability. The authors analyze in detail different sources of errors and gives precise estimations of uncertainties for every studied glacier, different data sets and analyzed periods.

**Response:** Firstly, we want to thank the anonymous Referee for the time and effort put in this detailed and thorough review. We carefully evaluated all comments and suggestions, which are extremely valuable in improving the paper. We are very glad about the positive feedback. We particularly value the Referee's appreciation of the error estimation, in which we invested significant effort and which we believe is one of the strong points of this manuscript with respect to previous studies. Throughout this response, as well as in the paper, we refer to the surface elevation change (in m) as "SEC" and to the surface elevation change rate (in m a$^{-1}$ or m d$^{-1}$) as "SECR".

We recognize that the focus of this review is on the seasonal correction which we apply for filling gaps in the rate of surface elevation change, in particular for the four-year time span of epoch 2012 to 2016. This is an important side issue aimed at reducing the uncertainty in mean annual SECR. The main contribution of the paper is the generation of spatially detailed data sets on surface elevation and volume change of both NPI and SPI for two different epochs, including the assessment of communalities and differences in glacier behaviour between the two ice fields and the two epochs. We are not aware of any other publication showing homogenous high resolution data sets on surface elevation or volume change of both icefields for two different epochs.

**Comment:** The results are not totally novel, since the data sets employed in this manuscript were recently used in a paper published by Malz et al., (2018) with some variations in dates, error assessments and study area. Unfortunately, the results are not totally comparable between them since the glacier

basins and dates are not the same. These differences preclude a precise estimation of discrepancies, but in general the results are statistically similar. The main contribution of Abdel Jaber et al, is their claimed much smaller uncertainties due to several correction that Malz et al didn't applied. After considering the analysis performed by Abdel Jaber et al, I think the error assessment is much more rigorous, effectively addressing many error sources of the data sets, but it is too ambitious when trying to extrapolate parameters from single stations/glaciers for correcting some issues related to the whole icefields. In those cases, is better to live with higher uncertainties and not adding more doubts as I think were added when using Perito Moreno as model for altitudinal gradients for example.

Thanks to the detailed error analysis, this manuscript provides a state-of-the art estimation of surface elevation change of both Patagonia Icefields. Unfortunately, the last data set of 2015 does not come from the end of ablation season, therefore, the authors applied a seasonal corrections to the derived surface elevation changes in order to provide estimates that correspond to full seasonal cycle. This seems to be a weak point of the study design. I guess this was caused by the availability of TanDEM-X imagery, however, employing a more recent datatake from the end of ablation season (later than 2015/16), would have considerably strengthen the importance of this contribution.

**Response:** While in fact some TanDEM-X raw data (Level-0) used in this study are the same as those used by Malz et al., (2018) we would like to highlight the main differences with respect to the latter publication. Malz et al., (2018) provides a SECR with full coverage of SPI for the epoch 2000-2015/2016 only, from which they derive the geodetic mass balance at an icefield level (with uncertainty) and at a basin level (without uncertainty). They furthermore provide two SECR maps covering the periods 2000-2012 and 2012-2015 which are based on SRTM and on the same pairs of TanDEM-X raw data that we use but they are restricted to the southern part of SPI (~2106 km², ~16% of SPI). Furthermore they do not exploit them to provide any mass balance estimation of the covered glaciers.

We argue that one of the strong points of this study is the use of a homogeneous set of methods and data to achieve SECR maps of both Patagonian icefields, featuring a very high coverage, for two similar observation periods. These maps are used to compute the geodetic mass balance (with uncertainty) of all glacier basins larger than 2 km² on NPI and larger than 9 km² on SPI. Hypsometric plots (with uncertainty) for both epochs are also reported for 15 main glaciers of NPI and 24 of SPI.

As correctly noted by the Referee a critical point of our study design is linked to the missing summer days in the "slave" elevation mosaic of the 2012-2015 SEC, caused by its acquisition time in December 2015, whereas February 2016 would have constituted the ideal setup. We hence understand that most of the constructive critique is linked to this issue. We firstly confirm the Referee's presumption: the choice of this dataset was indeed forced by the absence of more suitable coverages of the icefields at the beginning of this study.

As described in the manuscript, the seasonal elevation loss occurring on the icefields during the missing summer days is non-negligible. We show a map of measured seasonal changes on most of the SPI for a different summer, that of 2011/2012 (Fig. 8) which, however, was characterized by the same average air temperature as summer 2015/2016. We are not aware of any published seasonal SECR map of SPI.

Neglecting the impact of summer seasonal elevation variations on the SECR maps and on the corresponding geodetic mass balance would cause some bias, in particular for the second epoch. We

hence devise different correction strategies for different sections of the icefields, according to the availability of SECR datasets and complementary data. We are aware of the limitations of such strategies and take into account these limitations in the error budget. Detailed information on the points correctly raised by the Referee is provided below. We are convinced that the seasonal corrections improve the accuracy of the mean annual SECR compared to the approach by filling temporal gaps with average SECR without taking into account the missing days in the melting season.

**Comment:** Assuming that the surface elevation changes in summers 2011/2012 and 2015/2016 are equal, based on similarity of monthly mean air temperature records in some neighbouring weather stations is very arguable.

**Response:** We agree that these weather stations, in some distance from the icefields, are not the best choice for checking summer temperature over the icefields. We checked the ERA Interim temperature data, showing agreement within 0.1°C for both summers. The 850 hPa mean summer temperatures (December, January, February) are for point 47.25 S, 73.55 W over NPI: 5.9° C (2011/2012), 6.0° C (2015/2016) and for point 50.25 S, 73.55 W over southern SPI: 3.6° C (2011/2012), 3.5° C (2015/2016).

**Comment:** The other weak aspects of the manuscript are the assumptions regarding altitudinal gradients only supported by Perito Moreno glacier data. This glacier cannot be considered representative for the entire icefield due to extreme longitudinal gradients of climate and mass balance associated with orographic barrier of the Patagonian Andes.

**Response:** Here we explain again the strategy and procedures for the seasonal corrections (described in Section 3.1.3 of the manuscript), providing some additional details. The term seasonal correction refers to the difference between mean annual SECR without accounting for seasonal differences in SEC of the missing days and mean annual SECR taking seasonal differences into account. The seasonal correction is assessed separately for the two considered epochs.

Motivations for deriving SECRs over annual periods are: (i) to provide the basis for estimating the annual mass balance (MB), a key climate parameter; (ii) to facilitate comparisons of SECR, volume change and MB from different epochs; (iii) to facilitate comparisons with results of other studies (based on different observation techniques and also on MB modelling).

If repeat observations are not available in exactly annual or multi-annual (365.25xN days) repeat intervals, commonly the mean (average daily) SECR of the (incomplete) epoch is extrapolated to fill the missing temporal gap (or to subtract the contribution of excess days) in order to match annual time spans. This approach introduces a bias in annual SECR in case of seasonal variations. The magnitude of the bias depends on the percentage of missing (or excess) days and the amplitude of the seasonal cycle.

The bias in annual SECR due to deviation from exactly (multi-) annual intervals can be reduced by accounting for seasonal differences in SEC. In our case the temporal mismatch vs. exactly annual repeat observations (time spans of 12 years and 4 years) varies for the two epochs and for different sections of the ice fields. For the 2000 to 2012 time span the mismatch in percentage of the full period is very small. For the different TanDEM-X (TDM) tracks vs. SRTM over SPI the number of missing days corresponds to 0.1 % to 1.0 % of the 12 year period. Nevertheless, for epoch 2000-2012 over SPI we apply a seasonal correction where the temporal mismatch exceeds 15 days. On NPI no seasonal correction is applied because here the number of missing days is only 0.1% of the 12 year period.

In the following we provide details for the 2012 to 2016 data because the impact of seasonal corrections is more important due to the shorter time span. For NPI the number of missing days amounts to 3.6 % to 4.8 % of the 4-year time span, depending on the track. For the main sections of SPI the corresponding numbers range from 3.6 % to 5.0 %, except for a small subarea where it is 7 %.

Depending on the availability of additional TDM DEM data, the following procedures for seasonal corrections are applied for different sections of the icefields:

- TDM acquisitions of December 2011 cover the southern, central and north-eastern sections of SPI (59.4 % of the SPI area, manuscript Figure 8). These data are used to compute daily SECR over summer periods 2011/2012 by DEM differencing vs. TDM data of March 2012 (99 days, covering the main part of the SPI area) and vs. TDM data of 31 January 2012 (33 days, for a small section). According to ERA Interim the average temperatures of summer 2011/2012 and summer 2015/2016 agree within 0.1°C (see details above). This means that the SECR maps of summer 2011/2012 (scaled to the length of the missing period) can well be used as substitute for the missing days in summer 2015/2016.

- For glaciers that are not covered by the summer 2011/2012 SECR map we carry out two approaches, explained in manuscript Section 3.1.3 and repeated here. For the majority of these glaciers we use daily SECR in dependence of elevation (aggregated in 100 m elevation bins, Fig. S3) derived from the SECR maps of summer 2011/2012. For the missing part of Pio XI Glacier we use the hypsometric SECR curve of low-loss glaciers (green curve in Fig. S3), based on summer 2011/2012 SECR of Perito Moreno, Grey, Tyndall, HPS 13, Europa, Penguin and Guilardi glaciers. For the rest of SPI (except Jorge Montt Glacier) we use the hypsometric curve of the 99 days (blue curve in Fig. S3). On NPI, except S. Rafael, S. Quintin and (a very small frontal area of) Benito glaciers, we use the hypsometric green curve of Fig. S3 since here calving fluxes are a small component of total mass balance.

- For glaciers that are not covered by the summer 2011/2012 SECR maps and are subject to significant dynamic downwasting (Jorge Montt, S. Rafael, S. Quintin glaciers) we separate the SEC components in the ablation areas related to surface mass balance (SMB) from dynamic downwasting. The seasonal correction accounts for the difference in SMB between summer and the rest of the year for the glacier area below the equilibrium line. For this purpose estimates on the elevation dependence of specific SMB during summer and during the full year are needed. To our knowledge, up to know the only multi-year time series of ablation measurements (including separation of net balance during summer and the rest of the year) on any glacier of SPI and NPI has been performed on Moreno Glacier (Stuefer et al., 2007). We use these data to compute the ratio between daily SECR during summer vs. the rest of the year. Using this ratio, and the hypsometric SMB curve of Moreno Glacier, we estimated the increased contribution due to surface melt during summer relative to the full year up to the equilibrium line and apply this to the missing days. For the dynamic downwasting component we use the average SECR of the full period.

Regarding the validity of the P. Moreno mass balance gradient for Jorge Montt, S. Rafael, S. Quintin and Benito glaciers: We compared published SMB data for the terminus of NPI west coast glaciers with SMB computed with the balance gradient of P. Moreno Glacier, showing good agreement between these two estimates. We computed also mass balance gradients for west coast glaciers using the published balance

gradient of Chico Glacier (northern SPI) accounting for the different temperature lapse rates between east and west coast glaciers (Bravo et al., 2019). Notably, these two approaches yield quite similar results, pointing out that use of the Moreno balance gradient is a reasonable choice. See details in the Appendix.

**Comment:** I would be keen to see maps of systematic error similar to Fig S7 that shows random error. This is specially important for some major outlet glaciers of SPI (Jorge Montt, Pio XI, O'Higgins, Viedma and Upsala) where seasonal correction was in the order of several meters (Fig. S4b).

**Response:** The systematic error maps do not feature a strong spatial variation, with exception of regions affected by high probability of signal penetration and regions not covered by the summer 2011/2012 SECR. The systematic error maps are per se not very informative and were hence not included in the Supplement. We instead preferred providing quantitative values of the different systematic error components in Table S7. The systematic error linked to the seasonal correction is considered as a bulk error derived from the systematic error of the summer 2011/2012 SECR (see Sect. 3.3.3 and also the response to specific comment below), which was increased on specific regions to take into account the extrapolation through hypsometric average values on regions not covered by the summer 2011/2012 SECR. The weighting by the number of corrected days was also applied.

We noticed that the description of Figure S4 might lead to confusion and we will change it. Figure S4 represents the rasters (in meters) added during processing to the original SEC rasters (covering the entire observation period) to compensate for the missing/exceeding days.

**Comment:** This work is largely based on the results and methods reported by Abdel Jaber (2016) PhD thesis. In order to avoid undesirable repetitions, I guess the thesis can be quoted only a couple of times, and then assuming that the results are the one obtained in this manuscript.
In synthesis, I think this manuscript is highly valuable and fits very well with the aim and scope of the journal.

**Response:** This work is based on methods applied in the Abdel Jaber (2016) PhD thesis which were further developed and adapted to this study. Abdel Jaber (2016) reported SECR maps of NPI (2000-2014) and SPI (2000-2011/2012) with mass balance only at an icefield level. The results presented in this study are completely novel (we note that a limited number of TanDEM-X raw data are in common but were newly processed). For this reason in the manuscript we treat Abdel Jaber (2016) as a separate scientific study. The good agreement between the results of this study and those obtained by Abdel Jaber (2016) by separate processing represents, in our opinion, an added value for both publications.
The only results reported in the manuscript and stemming from Abdel Jaber (2016) are the subaqueous ice volume changes and frontal distance variation of the main SPI glaciers in the epoch 2000-2011/2012 (Table S9) and the icefield-wide subaqueous volume change rate reported in Sect. 4.

**Specific comments:**

**Comment:** P1 L20: ... and ..., respectively

**Response:** Wording changed to: "They stretch from 46.5° S to 47.5° S and from 48.3° S to 51.6° S, respectively, along the…"

**Comment:** P2 L9: any other reference? this is only review

**Response:** References added:

*Åström, J. A., Vallot, D., Schäfer, M., Welty, E. Z., O'Neel, S., Bartholomaus, T.. C., Liu,, Y., Riikilä, T. I., Zwinger, T., Timonen, J., and Moore, J. C.: Termini of calving glaciers as self-organized critical systems, Nature Geoscience, 7, DOI: 10.1038/NGEO2290, 2014.*

*Benn, D. I., Warren, C. R. and Mottram, R. H.: Calving processes and the dynamics of calving glaciers, Earth-Science Reviews, 82, 143-179, 2007.*

*Warren, C. R. and Aniya, M.: The calving glaciers of southern South America, Global and Planetary Change, 22, 59-77, 1999.*

**Comment:** P3 L8-9: You frequently refer to Abdel Jaber (2016), how different is its work from this submission? I presume is roughly speaking the same.

**Response:** Please see the response to the general comment above.

**Comment:** P3 L23-24: Later you refer to Abdel Jaber (2016) as a source for subaqueous ice loss estimates

**Response:** Please see the response to the general comment above.

**Comment:** P5 L15-20: Crippen et al. (2016) provided only a general description of NASADEM, and as far as I now, its performance has not been thoroughly compared with SRTMGL1. This I guess is one of main differences of this work compared to a very similar study by Malz et al. (2018) - how does it impact your final results? Were there really less voids compared to SRTMGL1?

**Response:** Voids, mostly caused by phase unwrapping errors, were filled using ASTER GDEM2 in the SRTMGL1 dataset (SRTM version 3). In the NASADEM (available in its void-filled and SRTM-only versions) voids were not very critical on great part of the icefield. We agree with the Referee that the performance of NASADEM has not been assessed yet. We hence performed our own assessment with the main focus being on sources of systematic errors, and particularly long-wavelength elevation biases. While not perfect (particularly around NPI) we concluded that NASADEM is certainly a step forward in this regard compared to SRTMGL1. We hence proceeded with this dataset for the production of the SECR maps and the computation of the mass balance. SECR maps based on SRTMGL1 were not produced; a quantitative comparison of volume change rates is hence not available.

**Comment:** P6 L25: How different were your glacier outlines from those used by Malz et al? The Randolph inventory is known to have problems in many places. Maybe you can discuss about this.

**Response:** Assuming reference to P5 L25. For the glacier outlines of NPI and SPI we use the Randolph Glacier Inventory in its latest version 6 with our own modifications. In RGI v6 significant changes

compared to the previous version are found on NPI, where a complete new set of outline was introduced based on those published by Rivera et al. (2007), while no changes are found on SPI. We are aware of the limitations of the RGI outlines, particularly in the definition of the internal divides between adjacent basins. We improved the external borders of the glacier outlines as described in Sect 2.3, whereas we did not modify the internal divides. The TanDEM-X DEMs have potential for such an improvement but this exceeded the scope of this study. We preferred to use the latest RGI for comparability of results, limiting changes to the termini.

The SPI outlines of Malz et al., (2018) are reportedly based on RGI version 5 with refinements based on optical images. Some non-negligible differences between our and their glacier outlines can be appreciated by comparing our Figure S9a with their Figure 2b. In particular differences are visible on Bernardo, Tempano, Occidental, Greve glaciers (the latter three appear as a single basin in their paper). Other differences are found on Chico/Viedma glaciers, Europa/Guilardi glaciers, and on smaller glaciers.

**Comment:** P7 L10: There is a problem when quoting equations along the whole manuscript. Only Eq. 2 is mentioned in the text and some equations are between lines without label numbers, don't helping in the fluent reading process. You refer here to Eq. (2) well before introducing it (and before Eq. (1))

**Response:** Agreed, many equations were written inline in order to reduce space and provide a flowing narrative. We will improve the manuscript on this issue.

**Comment:** P8 L20. The temperature and balance altitudinal gradients in the SPI are highly different between east-west margins or northern - southern parts etc. Maybe you can check a recent paper by Bravo et al in JGR (DOI: 10.1029/2018JD028857) and comment on this.

**Response:** Many thanks for referring to this recently published paper providing very valuable data for advancing the modelling of surface/atmosphere exchange processes and surface mass balance across the icefield. We performed computations on mass balance gradients using different lapse rates on east and west coast glaciers and comparisons with Moreno balance gradients. See details above and in the Appendix.

**Comment:** P8 23-36: Can you please clarify this part? For example Perito Moreno and Jorge Montt have very different climatic setting (Lenaerts et al., 2014), I wonder if Perito Moreno is a best choice for a reference in this case.

**Response:** See the detailed description on the seasonal correction above and the Appendix.

**Comment:** P8 L30-32: Again, issue of transferability of parameters of Perito Moreno to entire SPI. Why was it not necessary for NPI, because of the day of datatake, I guess?

**Response:** The 2000-2012 SECR of NPI was not corrected because the SRTM and the TanDEM-X data takes were acquired approximately in the same days of February. The seasonal changes on the scene with index 1, not involving summer days, were deemed to be negligible and much less than the systematic errors on a 12-year time frame. See also the detailed description on the seasonal correction above and the Appendix.

**Comment:** P8 L36-37; Fig S4: There are sharp boundaries between zones corresponding to different time spans, do they propagate to the final product introducing discontinuities?

**Response:** The discontinuities are caused by the different Δt in days to be corrected between adjacent master-slave pairs. These do not propagate in a noticeable way to the final SECR product. In fact light discontinuities might already be present in the uncorrected SECR, the correction would, at least in theory, reduce such discontinuities by compensating different amplitude of seasonal changes according to the actual Δt in days.

**Comment:** P9 L19: Uncertainty bound on glacier-wide density seems to be too low. Cogley (2009) refer to Sapiano et al. (1998) 6% estimate as reasonable. In similar work, Malz et al. (2018) provide three scenarios of different densities, it is their main source of uncertainty for the final results.

**Response:** The main scope of this study is to provide reliable volume change rate estimates at a basin scale (Tables 2, 3, S8). These can then be converted to mass change rates using a constant density assumption and a corresponding error of choice. Furthermore we also wanted to provide a reference mass change rate at an icefield level. For this purpose we decided to use the common scenario of glacier-wide density of 900 kg m$^{-3}$ facilitating the comparison of results with other studies. We agree that the assigned error of 17 kg m$^{-3}$ is a small one (1.8%) for firn areas. However, the main mass losses on the Patagonian icefields refer to ice areas. For the accumulation areas assumptions on changes of the vertical profiles of snow/ice density would be speculative. Furthermore, the core of the error estimation of this study focuses on the volume change rate estimates.

**Comment:** P10 L7-10: Please back it up with some reference

**Response:** References added:

*Garreaud, R., Lopez, P., Minvielle, M., and Rojas, M.: Large Scale Control on the Patagonia Climate, J. Climate, 26, 215–230, 2012.*

*Schaefer, M., Machguth, H., Falvey, M., and Casassa, G.: Modeling past and future surface mass balance of the Northern Patagonian Icefield, J. Geophys. Res.-Earth, 118, 571–588, doi:10.1002/jgrf.20038, 2013.*

**Comment:** P10 L24: Finally, what exactly was the criterion for masking regions prone to penetration? Was it only manually outlined based on expert knowledge?

**Response:** In each DEM mosaic, regions prone to signal penetration were not masked-out, instead they were manually outlined and assigned a certain penetration bias, which was then used to compute the corresponding systematic error component of the SECR (Sect. 3.3.3). The penetration bias was assigned based on the average σ$^0$ within the region. Measurements of signal penetration in TanDEM-X data over NPI at varying σ$^0$ are reported in Abdel Jaber (2016). These, together with expert knowledge on relations between X-band σ° and signal penetration were used to assign the penetration bias values. During summer the top snow layers on the main ice plateau are either wet (low σ°) or include melt/freeze metamorphic layers with rather small penetration for X-band signals also in frozen state (Reber et al., 1987).

*Reber, C., Mätzler, C., and Schanda, E.: Microwave signatures of snow crusts, modelling and measurements, Int. J. Remote Sensing, 8 (11), 1649 – 1665, 1987.*

**Comment:** P11 L8-9: See comment above.

**Response:** The average $\sigma^0$ of the SRTM acquisitions were analyzed to reach this conclusion. This is another novel aspect of this publication; we are not aware of an empirical assessment of the backscatter of C-band SRTM to assess possible signal penetration in glaciological remote sensing studies. Furthermore, the good agreement between our volume change rates 2000 to 2012 over NPI and the results of Dussaillant et al. (2018) based on optical data confirm the validity of our approach regarding signal penetration. See also their conclusion: " our study confirms the lack of penetration of the C-band SRTM radar signal into the NPI snow and firn except for a region above 2,900 m a.s.l."

**Comment:** P11 L11: Bippus (2007) assumed this lapse-rate for summer season on Perito Moreno, however as far as I know this value was not based on measurements. Additionally, she accounted for an off-glacier location of AWS, resulting in additional temperature offset. Maybe you can compare your numbers with Bravo et al 2019

**Response:** Thank you for this input. However, we want to stress that the conclusion about signal penetration is primarily based on the backscatter assessment. Regarding penetration see also the comment above.

**Comment:** P12 L26: Is 0.1 m based on literature?

**Response:** The bulk systematic error of 0.1 m for the remaining pixels above 1000 m a.s.l. accounts for undetected regions (a very small percentage of the total area) and possible small offsets on refrozen upper layer of snow and firn, affecting only very small areas; see response above. The typical X-band one-way penetration depth of frozen crust is 0.1 m (Reber et al, 1987).

**Comment:** P13 L1-5: I think that the error linked to the seasonal correction may be underestimated as it does not seem to cover all uncertainties related to the transferability of hypsometric averages shown in Fig. 8 (see previous comments).

**Response**: See also the response to the general comment and Sect. 3.3.3. The systematic error linked to the seasonal correction is derived from the systematic error the summer 2011/2012 SECR used for the correction on most of the glaciers. This systematic error was increased to account for the different year to correct, although summer air temperatures were nearly the same (see response above). It was also increased on regions where the summer 2011/2012 SECR has no coverage (NPI, north-west SPI and many gaps throughout SPI) in order to account for the transferability of hypsometric averages. Such increases might even lead to an overestimation of this error source; this is accepted as it tends towards a more conservative error budget.

**Comment:** P15 L26: It that is true than your seasonal correction should use lower density in accumulation area.

**Response:** P15 L26 does neither refer to seasonal correction nor to density in the accumulation area. Regarding the comment above concerning seasonal correction, there is no point for using snow density, because the correction refers to seasonal differences (summer vs. rest of the year) in surface elevation.

**Comment:** P15 L30: Perito Moreno glacier

**Response:** Corrected.

**Comment:** P16 L28: Results were recently published in Frontiers - Langhammer et al. (2018)

**Response:** Thank you; we will include the reference to this publication.

**Comment:** P16 L37: Again, I doubt that Perito Moreno is representative for entire SPI and SPI, Steufer (2007)

**Response:** There seems to be some misunderstanding regarding the use of Perito Moreno seasonal balance gradients. This is not used for entire SPI and NPI, but to estimate the impact of seasonal differences in surface lowering due to ablation for altogether only four glaciers of the two icefields and only for the epoch 2012-2016. See also the detailed description on the seasonal correction above and the Appendix.

**Comment:** P17 L26: This issue is a critical factor in the whole analysis of the elevation changes in the high plateau of the icefields. We know that the accumulation is extremely high, an in between few days you can have huge accumulation events. I think this high temporal variation of snow fall must be taken into account. See Schwikowski et al 2013 for snow accumulation on the SPI.

**Response:** At first, we want to point out that we are well aware on the spatial and temporal variability of accumulation on SPI. The work reported in this paper does not deal with single events, but addresses the retrieval of spatially detailed maps on changes in surface elevation and volume over SPI and NPI, providing mean values over epochs of 12 years and 4 years. The contribution of single events during this period is implicitly included in this analysis. Regarding spatial variability, this is fully taken into account by the high spatial resolution of the TanDEM-X elevation data and the high percentage of spatial coverage (Table 3). Regarding the summer periods 2011/2012 vs. 2015/2016, there is no indication on exceptional events as according to ERA Interim there is perfect agreement for air temperature and the difference in precipitation over SPI between these two summers is 15 % (probably within the uncertainty of precipitation estimates for this region).

**Comment:** P18 9? What analysis? It is missing in methods and results sections. Maybe you wanted to quote Abdel jabber 2012?

**Response:** Assuming reference to P18 L4. Agreed, the sentence is not well-formulated. The analysis refers to ice flow velocity results external to this study and based on TerraSAR-X. We will reformulate.

**Comment:** P18 L25: See comment above P33 Fig.4: Why is there a sharp transition in the terminal part of the glacier on panel a? Frontal retreat I guess?

**Response:** Yes, exactly. The sharp transition is in fact a physical signal in the elevation difference between the glacier front in 2012 (abrupt step in elevation) and what in year 2000 was the glacier surface (smooth increase in elevation).

**New references mentioned in this review:**

Bravo, C. et al (2019) Air Temperature Characteristics, Distribution and Impact on Modeled Ablation for the South Patagonia Icefield. JGR, DOI: 10.1029/2018JD028857

Langhamer, L., Sauter, T., & Mayr, G. J. (2018). Lagrangian Detection of Moisture Sources for the Southern Patagonia Icefield (1979-2017). Frontiers in Earth Science, https://doi.org/10.3389/feart.2018.00219

Lenaerts, J. T., Van Den Broeke, M. R., van Wessem, J. M., van de Berg, W. J., van Meijgaard, E., van Ulft, L. H., & Schaefer, M. (2014). Extreme precipitation and climate gradients in Patagonia revealed by high-resolution regional atmospheric climate modeling. Journal of Climate, 27(12), 4607-4621.

Sapiano, J.J., W.D. Harrison and K.A. Echelmeyer. 1998. Elevation, volume and terminus changes of nine glaciers in North America. Journal of Glaciology, 44(146), 119- 135.

Schwikowski, M., M. Schläppi, P. Santibañez, A. Rivera and Casassa G. (2013): "Net accumulation rates derived from ice core stable isotope records of Pío XI glacier, Southern Patagonia Icefield". The Cryosphere 7, 1635-1644. doi.org/10.5194/tc-7- 1635-2013

**APPENDIX**

**Estimation of mass balance gradients for Jorge Montt Glacier and for the NPI glaciers S. Rafael. S. Quintin and Benito**

In the current version of the paper (Abdel Jaber et al., 2018) the mass balance (MB) gradients of Moreno Glacier are used to estimate the seasonal difference of surface lowering in the ablation area related to the surface mass balance (SMB) for J. Montt Glacier and for three glaciers on the west coast of NPI. In the review the use of the Moreno MB gradient for these glaciers is questioned, pointing out the different temperature lapse rates between the east and west side of SPI measured along a transect crossing the northern section of the icefield (Bravo et al., 2019). Taking this into account, we computed the MB gradient for west side glaciers based on the MB gradient for a glacier on the east side of the norther section of SPI, accounting for the different lapse rates.

*Mass balance gradients:*

- For Chico Glacier Rivera (2005) shows a MB gradient based on modelling. The (close to linear) value in the ablation area is: $\Delta b_n/\Delta z = 0.015$ m w.e./(m a).

- The ratio of the west/east temperature lapse rate (LR) at the transect Occidental to O'Higgins glaciers is: LR(west) = 0.76 LR(east) (Bravo et al., 2019):
- Chico Glacier is adjacent to O'Higgins Glacier. Using the balance gradient of Chico Glacier and the west/east ratio of LR results in the MB gradient: $\Delta b_n/\Delta z$ (west) = 0.0114 m w.e./(m a)
- For comparison: The MB gradient for Moreno Glacier: $\Delta b_n/\Delta z$ (Mor) = 0.0122 m w.e./(m a)

*Specific mass balance on the terminus of tidewater calving glaciers at 50 m a.s.l.:*

- Equilibrium line altitude (ELA) is used as reference for specific mass balance $b_n = 0$ at ELA.
  Mean ELA NPI west (S. Rafael. S. Quintin): 1200 m (Rivera et al., 2007).
  Mean ELA J. Montt Glacier: 1100 m (Rivera et al., 2012)
- The resulting $b_n$ at 50 m a.s.l using $\Delta b_n/\Delta z$ (west) is: NPI west $b_n$ = 13.11 m w.e. $a^{-1}$; Jorge Montt $b_n$ = 11.97 m w.e. $a^{-1}$.
- The resulting $b_n$ at 50 m a.s.l using $\Delta b_n/\Delta z$ (Moreno) is : NPI west $b_n$ = 14.03 m w.e. $a^{-1}$; Jorge Montt $b_n$ = 12.81 m w.e. $a^{-1}$.
- Comparison with published (modelled) $b_n$ for NPI western outlet glaciers (on the terminus near the glacier front): 14 m w.e $a^{-1}$ (Schaefer et al, 2013).

**Conclusion:**

The computed $b_n$ for the terminus of NPI western outlet glaciers, using the Moreno Glacier MB gradient, shows good agreement with $b_n$ reported by Schaefer et al. (2013). This confirms the use of the Moreno Glacier MB gradient as a reasonable option for computing the seasonal correction for these glaciers. For $b_n$ on the terminus of J. Montt Glacier we did not find any published data. However, considering the comparatively short distance to NPI (300 km) it can be assumed that the MB gradient is rather similar. Therefore (and for better traceability) we use the Moreno MB gradient for estimating the SMB-related seasonal correction also for J. Montt Glacier. The use of the MB gradient for west coast glacier, derived from the Chico Glacier MB gradient, would not cause any significant difference for the seasonal correction. , backing up this solution.

**Error estimate:** for $b_n$ $\pm20$ %; for the ratio $b_n$ summer vs. full year $b_{n,s}/b_{n,yr}$ $\pm20$ %; combined $\pm28$ %.

*References:*

*Rivera, A.: Mass balance investigations at Glacier Chico, Southern Patagonia Icefield, Chile. Ph.D. thesis, Univ. of Bristol, U.K., 2005.*

*Rivera, A., Benham, T., Casassa, G., Bamber, J., and Dowdeswell, J. A.: Ice elevation and areal changes of glaciers from the Northern Patagonia Icefield, Chile, Global and Planetary Change, 59, 126–137, 2007.*

*Rivera A., Koppes M., Bravo C. and Aravena J.C.: Little Ice Age advance and retreat of Glaciar Jorge Montt, Chilean Patagonia, Clim. Past., 8, 403–414, 2012.*

*Schaefer, M., Machguth, H., Falvey, M., and Casassa, G.: Modeling past and future surface mass balance of the Northern Patagonian Icefield, J. Geophys. Res.-Earth, 118, 571–588, doi:10.1002/jgrf.20038, 2013.*

---

## Author Comment (AC2) · 6 Mar 2019

**Interactive comment on "Heterogeneous spatial and temporal pattern of surface elevation change and mass balance of the Patagonian icefields between 2000 and 2016"**

*Authors*: Abdel Jaber, W., Rott, H., Floricioiu, D., Wuite, J., Miranda, N.

The Cryosphere Discuss., https://doi.org/10.5194/tc-2018-258

**Authors' response to Anonymous Referee #2**

Referee comments are shown in **black**, our response in **blue**. Line numbers refer to the manuscript version (pdf) of 23 November 2018.

**General comments:**

**Comment:** The study presented by Wael Abdel Jaber and co-authors is an overview of surface elevation change rate (SECR) and geodetic mass balance (MB) values for the Southern Patagonia Icefield (SPI) and Northern Patagonia Icefield (NPI) for the two epochs 2000-2012 and 2012-2016. The results are calculated on the entire icefield as well as on glacier basis, mean SECR and volume change rates (VCR) are listed in a table including observed area and error budget. For most important glaciers the hypsometric distribution of those variables is depicted in graphs. The study provides a detailed description of the error analysis and several steps to correct for biases and penetration and ablation uncertainties. The language is correct and understandable. The subject is of high interest to the community, the method and study areas are not completely novel. In the last years, there have been publications covering the study area with the same topic (Foresta, Dussaillant, Malz, Abdel Jaber himself), but partly using different approaches. This new study cites and discusses those adequately. I recommend to add the recent work of Braun et al. (2019) which also includes SPI and NPI, but only covers the first observation period (2000-2011/15). The authors point out two aspect as main progress to previous studies: 1) The comprehensive and simultaneous observation of both icefields at two epochs. 2) The variety of corrections and assumptions made to guarantee a precise observation of SECR and following products. The line of argumentation is clear as far as (1) is concerned and thus I support publication in TC. Nevertheless, concerning (2), revisions should be performed to significantly improve the traceability of results and assure the validity of some of the applied steps described in the method section before publication.

**Response:** We thank the anonymous referee for the detailed review and the appreciation of our work. Although similar studies were published already we want to point out the main novelties of our manuscript. We provide the first geodetic mass balance for NPI and SPI also for a recent epoch (2012-2016) by TanDEM-X DEM differencing. Besides the entire icefields we give average SECR and VCR (incl. error) for individual glacier basins (up to 9km$^2$ on SPI and 2km$^2$ on NPI) and hypsometric plots of main glaciers (incl. error bars). We used the same method as for the preceding epoch (2000-2012) and this

allows the comparison of individual glacier and icefield behavior in the two epochs. Also we present an up to now unique analysis of the backscatter coefficients of all SAR acquisitions (SRTM and TanDEM-X) to assess the error due to signal penetration, a known issue when using InSAR based DEMs. Abdel Jaber et al. (2016) is a doctoral thesis. It has not been published in any scientific journal, neither in its entirety nor any part of it, but it is available online to everybody. The thesis, reporting many details on the methods used for SECR and VCR, provides also the basis for the technical approach applied in this paper. This review asks for many details on techniques for TanDEM-X DEM differencing and retrieval of SECR which may be relevant for a technical paper on DEM differencing, but is not the scope of our paper. We want to point out that the methods section on the current version of the paper exceeds in terms of information content previous papers on SPI and NPI retrievals of volume change. We provide in this response information on specific technical issues raised by the referee.

Thanks for pointing to the recently published paper by Braun et al. (2019). We will make reference to this paper. Regarding DEM differencing SRTM-TanDEM-X, for SPI, the numbers seem to be based on Malz et al. (2018). The reported number for NPI 2000 - 2011/2015 lacks traceability regarding the TanDEM-X data used and processing methods so that in depth comparison with our results is not possible.

**Comment:** Methods: The utilization of several thresholds or distinct values is not always transparently explained. At some decisive points, it remains vague if the method or decision follows own reasoning, own previous work or an external reference (cf. specific comments)

**Response:** We thank the referee for pointing out this aspect. We provide relevant information in the response to specific comments.

**Comment:** The correction for the observation date in epoch 2, for not being at the end of ablation period, is an unprecedented venture. However, it forms also a weak point of the study. In the reviewer's opinion, the error induced to the SECR (Epoch 2 – Epoch1) by this step is not adequately represented by the mapped datasets nor is it transparently addressed as error contribution in the text. Moreover, an interpolation of missing areas based on only two weather stations and adjusted to sparse hypsometrical patterns has to be regarded rather experimental compared to the robust methodology used for the rest of the study. It is hard to judge the validity of the seasonal correction. A Δh map outside the icefields and the unfiltered dataset Δh could help justifying, at least for the observed parts(cf. specific comments)

**Response:** We are aware that the correction applied for the missing days in the ablation season to complete the 4 years period of epoch 2 has some limitations. On the other hand performing such a correction for the short period is fundamental for obtaining reliable annual SECR for comparisons with other results. This reduces a possible bias compared to the case without seasonal correction. For the details of the procedure please refer to our response to referee #1, (AC#1, pages 3-5). The description in Section 3.1.3 of the paper will be improved by reformulating the text and adding the requested Δh maps in the Supplement if deemed to be necessary.

**Comment:** The error indicated for SECR is spectacularly low in this paper. Although there is a section explaining the calculation it is not totally clear, why a DEM comparison could come up with such low elevation error budget. It appears, the systematic error budget, as the main contributor, is calculated partly in favor of a small total error. Some steps along this path should be under discussion or described in more detail for traceability (cf. specific comments).

**Response:** The errors of SECR and of VCR (Table 1) are comparable to most of the recent results obtained by other authors based on elevation change approach (see Table S10). Therefore we do not understand the reason for the reviewer's statements "spectacularly low" error and "the systematic error is calculated in favor of a small total error". Furthermore, the agreement between our volume change rate 2000 to 2012 over NPI and the results of Dussaillant et al. (2018) is well within the combined error bound. This supports the validity of our error estimate, as the results of Dussaillant at al. are based on completely different data sets and methods. As suggested, we will provide further clarification on the error estimate in the revised manuscript and Supplement.

**Comment:** Structure: The work is based on the PhD thesis of Abdel Jaber (2016). However, since it is sometimes difficult to follow what is actually new in contrast to what was already in place, that presents the reader with challenges. A clear line between parts that were newly implemented  and those that were adopted needs to be drawn by the authors. I recommend that the authors revise the methods and result section with regard to this aspect to make the paper a full stand-alone document. This also concerns the length of some descriptions that could be kept more concise for this paper, with reference to the thesis (or other original source).

**Response:** The referee #1 had a similar concern which we answered in our response (AC#1 page 5). We will revise the two sections of the manuscript in this respect.

**Specific comments:**

**Comment:** P 6 l27 ...(in order of impact, the latter being negligible in our Raw DEMs)." This and further statements could be corroborated by a similar Figure as Fig S 2 for SRTMTDM, displaying same Δh for outside the icefields for SRTM-TDM(Ep1) and TDM(Ep1)- TDM(Ep2).

**Response:** Off glacier SECR close to some termini were kept in the detailed maps shown in Figures 2 - 7 for this purpose. We do not see any urgent need for adding these 4 figures (on a marginal topic) as it would inflate the already extensive size of the paper. Even so, we can add these figures in the Supplement if deemed to be necessary.

**Comment:** P 7 ll3 -13 The weighted averaging of the offset values leaves the question if a spatial pattern was analysed and fitted by an offset function. A simple averaging could lead to regional maladjustment, if the sign / magnitude of the offset is a function of geographic position (tilted dataset, described in this manuscript p6 l20). For the precision of the applied method a mapped Δh (cf. comment to P6 ll26) could be convincing.

**Response:** As mentioned above we can include the 4 figures in the Supplement if deemed necessary.

**Comment:** P7 ll13 How is the absence of horizontal shifts checked? The detection is slope dependent (cf. Nuth and Kääb (2011)), thus cannot be efficiently performed on an area without slope as the CRs (avr. Slope below 4∘)

**Response:** The horizontal shifts (in our case possibly acting in the ground range direction) were not checked analytically directly on our datasets, but relying on visual analysis of all available off-glacier

terrain. Analytical checks using the method of Nuth and Kääb (2011) was done for the TDM-SRTM SEC datasets during the preparation of the thesis (Abdel Jaber et al., 2016) corroborating the validity of this calibration procedure. Because the same method was applied for this paper as for the thesis, the conclusions regarding this procedure can be adopted for this work.

**Comment:** p7 ll23 Please provide reference

**Response:** Rivera (2004) Fig. 6.3 shows in January clearly higher density in the upper metres of snowpits in the accumulation area of Chico glacier, compared to density in September and October. We will add the reference:

Rivera, A.: Mass balance investigations at Glaciar Chico, Southern Patagonia Icefield, Chile, Ph.D. thesis, Univ. of Bristol, UK, 2004.

**Comment:** P 7 ll30 What kind of filtering was applied? It would be interesting to see the original dataset and a Δh map outside the icefields.

**Response:** Since the Summer 2011/2012 daily SECR is not the main result of this study and it is used only for the seasonal correction, we applied: (i) conservative masking on glaciated terrain of regions with high backscattering and peaks in the daily SECR values followed by (ii) 2-step filtering with sliding window: (a) median filters with kernel size 9 and (b) smoothing with kernel size 9. The raster posting is 0.4 arcsec. This way the localized seasonal changes or outliers were eliminated and thus the SECR map can be used for the purpose of compensating the temporal gap in 2015/2016.

**Comment:** P8 ll10 What does similar mean here? +-0◦C? Please add a number for consistency.

**Response:** The 2 stations data we used are confirmed by ERA Interim temperature data (see also response AC#1 page 3) which provide even higher agreement between the summer epochs. According to ERA Interim the average temperatures of summer 2011/2012 and summer 2015/2016 agree within 0.1°C. This means that the SECR maps of summer 2011/2012 (scaled to the length of the missing period) can well be used as substitute for the missing days in summer 2015/2016. We will add this info in this paragraph.

**Comment:** P7 ll32 -p8 37 A comprehensive series of comments concerning the temperature variability and spatially variable ablation patterns resulting in a rather speculative adjustment in the seasonal correction section is given by referee #1. I agree on those.

**Response:** Please see the detailed response to referee #1 in AC#1 pages 3 - 5

**Comment:** P8 ll28-32 Please explain the justification of 20% reduction in correlation to a temperature value. Based on what assumption does it translate into a percentage?

**Response:** For epoch 1, although the daily SECR is from the same year (ablation season 2011/2012) and was obtained from December to March, the days which have to be compensated are in late summer and therefore we reduced the estimate for ablation by 20 % compared to the summer average. As we mentioned in the manuscript, this scaling factor is based on a time series of daily air temperature

measurements from 1995 to 2003 near the front of Perito Moreno Glacier (Stuefer et al., 2007). Furthermore, we want to point out that this correction factor applied on the hypsometric curve in Fig S3 affects only a very small area.

**Comment:** P9 ll8 Can you please add more information to increase reproducibility when data gets available: what threshold on SEC values? What morphological operators?

**Response:** We did not include these details because we do not think that this is an interesting point and would inflate an already very long paper. For each of the 4 SECR maps we produced a raster starting from the flag mask (FLM) layer that resulted from the processing with ITP which provides roughly the regions affected by layover and shadow. Thresholds $\Delta h/\Delta t < -10$ m/a and $> +6$ m/a were applied. A morphological operator of closing followed by a 5 x 5 median filter was applied on the mask raster in order to "clean" the mask, avoiding noise due to thresholding.

**Comment:** P9 ll16-19 Where are the 17 kg m-3 uncertainty taken from? Citation of Cogley et al. (2009) is misleading here, because reader would expect a reference for the density uncertainty. I found it to be mentioned in Abdel Jaber (2016), but it seems to be taken from Gardner et al. (2012) – this is not referenced here. Anyway, why using this value when recent large area studies like Brun et al. (2017), Dussaillant et al. (2018), Malz et al. (2018) use 60 kg m-3? Choosing that latter value would lead to comparable error budget.

**Response:** Yes, the uncertainty $\pm 17$ kg m$^{-3}$ comes from (Gardner et al, 2012). Regarding this issue, we want to mention again the statement in response to Referee 1, that he main scope of this study is to provide volume change rate estimates at basin scale (Tables 2, 3, S8). These can then be converted to mass change rates using a constant density assumption (which provides full traceability). Furthermore we wanted to provide a reference mass change rate at an icefield level; these are the only four numbers in the paper reporting mass change estimates (Table 1). We decided to use the common scenario of glacier-wide density of 900 kg m$^{-3}$ facilitating the comparison of results with other studies. We agree that the assigned error of 17 kg m$^{-3}$ is a small one (1.8%) for firn areas. However, the main mass losses on the Patagonian icefields refer to ice areas. For the accumulation areas assumptions on changes of the vertical profiles of snow/ice density would be speculative. The changes in SEC on the firn plateau are small, suggesting that an error value of 60 kg km$^{-3}$ is probably an overestimation. In case, we can use different bulk estimates for uncertainty of density in the firn and ice areas in order to revise the error estimates for the icefield wide mass changes in Table 1.

**Comment:** P9 ll 31 Please provide reference

**Response:** Following reference will be added:

DLR-CAF. 2013 (October).TerraSAR-X Ground Segment Basic Product Specification Document. 1.9 edn. German Aerospace Center (DLR) - Cluster Applied Remote Sensing (CAF). TX-GS-DD-3302.

**Comment:** p10 ll21-24 and P11 ll8 Why manual outlines? What is the decision to delimit these areas based on? If that information can be found in Abdel Jaber (2016) it should be indicated (or the original study it referes to).

**Response:** The outlining was performed manually based on the backscatter coefficient ($\sigma^0$) and taking also into account the corresponding elevation to ensure that areas of high backscattering are on the smooth firn plateau and not in the ice areas. A fixed thresholding of the $\sigma^0$ layers would have added noise because this would include regions of rough ice. The penetration height offsets assigned to each region are based on the relation between difference in $\sigma°$ for dry and wet snow and $\Delta h$ (Figure 8.12 of Abdel Jaber (2016), Sect. 8.4).

**Comment:** P11 ll3-7 Is any of the values mentioned in these paragraphs used for determining the outlines? What is the interpretation of the sigma0 ranges based on? Abdel Jaber (2016) / other? Please reference it.

**Response:** The $\sigma°$-values and related interpretation are based on multi-year experimental and theoretical work on X-band and C-band radar signal interaction with snow and ice by two of the co-authors, including several field campaigns on Alpine glaciers related to ERS-1/ERS-2, ASAR of Envisat, Shuttle Radar SIR-C/ X-SAR SRL-1 and -2, and TerraSAR-X. See e.g. Nagler and Rott (2000); Floricioiu and Rott (2001). Concerning the SRTM data (Sect.3.2.2), to which this comment refers, such an analysis was already performed in (Abdel Jaber, 2016) and the processing was not repeated for his study. Only the analysis of TDM backscatter (Sect. 3.2.1) is new because the data used in this study have not been used in Abdel Jaber (2016).

Floricioiu D. and Rott, H.: Seasonal and short-term variability of multifrequency, polarimetric radar backscatter of alpine terrain from SIR-C/X-SAR and AIRSAR data. IEEE Trans. Geosc. Rem. Sens., Vol.39(12), 2634 –2648, 2001.

Nagler T. and Rott, H.: Retrieval of wet snow by means of multitemporal SAR data. IEEE Trans. Geosc. Rem. Sens., Vol 38(2), 754-76, 2000.

**Comment:** P11ll 17 First sentence would be well supported by a formula. Is the HEM for the TDM elevations calculated by the phase difference to the interferometric phase of 12 m TDM products? Is it always TDM 12m as a reference (also in 3.3.3 (1))? It is mentioned once briefly in 2.1, but I think I should be emphasised there, that it is especially used as reference for elevation error assessment.

**Response:** We can add the formula, even if trivial and therefore in our opinion not necessary. The HEM does not depend on the reference DEM (the global TDM DEM) but it is processed by ITP for each TDM RAW DEM. The HEM is only the interferometric error and reflects point per point the actual error. This is the alternative of computing it over ice free terrain, as other authors did. This error is used to compute the random error of each sample. When averaging on an elevation bin this error becomes anyway negligible compared to the systematic components.

In 2.1 (p 4 lines 13-14) we state only that the global TDM DEM is used as reference for the processing, the details of how this is used are given in 3.1.1. It is also used for the DEM coregistration (see sect. 3.1.2 p 6 lines 24-25). We never mention that we used the global TDM DEM in error calculation.

**Comment:** P11 ll25: How was it included? Add some mathematical explanation of the error propagation through seasonal correction. Is it sqrt($\sigma t1_2 + \sigma t2_2 + \sigma seas_2$) for each pixel?

**Response:** We can add the formula, even if trivial. However, in our view we should rather not increase the paper length by adding such material.

**Comment:** P12 ll14 Enhance precise and illustrative explanation to this whole section 3.3.3. The reader is interested how exactly the systematic error is calculated, for it is key to the low elevation error budget presented. Please provide formulas to enhance comprehensibility. That could spare some explanatory text passages, that are less illustrative.

**Response:** We will revise this section accordingly without inflating it too much.

**Comment:** p 12 ll14 Is the IQR of the areas that were adjusted (CR, calibration) addressed as the measure of error (validation) on each DEM? I do not agree with this method from a scientific perspective. On top, choosing the IQR reduces or eliminate slope dependent effects (avr. Slope below 4◦, IQR slope?). But on glacier these are present for sure, so they are a source of systematic error to be addressed in the budget. It would be more reliable if validation is performed on the entire DEM (glaciers excluded), but assessed with regard to absolute elevation and slope.

**Response:** Assuming the comment concerns p 12 ll17-20.

The IQR is chosen to characterize the spread of the elevation difference between the TDM Raw DEM and the global TDM DEM within each scene over calibration regions covered by that scene. Large difference means larger uncertainty of the DEM calibration. The IQR was chosen instead of standard deviation because the distribution is not Gaussian. A tilt of a specific Raw DEM was excluded through the comparison to the reference global TDM DEM. If still present it would lead to an increase of the spread, since the calibration regions (visible in Figure S1) are relatively well distributed geographically.

Slope dependent elevation offsets are caused by horizontal shifts between the two DEMs, which we assumed to be negligible compared to vertical ones. The glaciers are rather flat, except on the small regions of the mountain ranges sticking out of the plateau. Therefore, the combination of small shifts and small slopes would make these effects negligible. We focus on vertical offsets. Evaluating on the entire off-glacier surface would have been another possibility. But this would have been biased by the higher slopes of the off glacier mountains of Patagonia and would lead to a larger error than what we found on the glaciers which have low and moderate slopes. Since this error is linked to the calibration procedure we compute it on the calibration regions themselves.

**Comment:** P12 ll21 Why 1 – 6 m? Reference, calculation or explanation for decision should be provided. Where does that assumption 1000 m.a.s.l come from? Please provide reference.

**Response:** The penetration height offsets of 1 to 6 m were assigned based on explanations given in Sections 3.2.1 and 3.2.2 and in (Abdel Jaber, 2016) and are referenced at line P12 L22. See response above. We assign the mentioned systematic error only to altitudes above 1000 m to compensate for undetected regions on the plateau and not on the rough ice of the glacier termini.

Below 1000 m a.s.l. are ice areas where the C-and X-band radar signal does not penetrate. For the SRTM data set (C-band) penetration is no issue because the snow and ice surfaces were wet during the SRTM

mission. Regarding the TanDEM-X data, only a quite small percentage of the total data set exhibited partly frozen or dry snow. As mentioned in the response to P11, line 8, the penetration height offsets for completely dry snow and firn are based on the relation between σ° for dry and wet snow and Δh in Figure 8.12 of (Abdel Jaber, 2016) showing for dry snow a mean offset of 4 m and a maximum offset of 6 m. This is in agreement with the number on X-band one-way signal penetration for dry snow reported by Mätzler (1987) if converted into two-way penetration and accounting for oblique view. The good agreement between our volume change rates 2000 to 2012 over NPI and the results of Dussaillant et al. (2018) based on optical data confirms the validity of our approach regarding signal penetration.

Mätzler, C.; Applications of the interaction of microwaves with the natural snow cover, *Remote Sensing Review, 2*, 259-387, 1987.

**Comment:** P13 ll1 According to this paragraph: for interpolated seasonal correction, the last epsilon term should dominate the quadrature sum and thus the total SECR error ,if I understand correctly. What does 'increase by a factor of three' mean in this context? Times 3 (*3) ? I compared SECR uncertainty value for extrapolated glaciers (e.g. Jorge Montt, Bernardo, Tempano) in Tab. 3 with values for not extrapolated glaciers. First ones are not near triple of latter. And they should even be higher than triple, following this paragraph: scaling by year (divided by 0.27. for 99 days for example) is performed  as well as a *1.5 increase for the timespan difference. Please explain where I've gone wrong and/or revise the explanations in this paragraph.

**Response:** Thanks for pointing this out. It seems there is some misunderstanding regarding the seasonal correction and its impact for the retrieval of SECR. The term seasonal correction refers to the difference between mean annual SECR over epochs spanning 12 years (2000 to 2012) and 4 years (2012 to 2016) without accounting for seasonal differences in SEC of the missing days vs. the mean annual SECR taking seasonal differences into account. For the extrapolated glaciers 53 to 103 summer days are missing in order to cover the full 4 year period (1461 days). This means that the missing days to be substituted correspond to 3.6 % to 7.0 % of the 4 year period for which the mean SECR is computed (and not 27 % which would refer to a single year). The impact of missing days to be substituted for the 12 year period is still much smaller. Our response to the general comments of Referee 1 includes detailed explanations on the seasonal correction. We will revise section 3.1.3 (seasonal correction) accordingly in order to provide better explanations. We will check the uncertainty estimates for the individual glaciers in order to be sure that the number of missing days has been properly taken into account.

**Comment:** P13 ll9 A formula containing the total SECR error would be helpful for traceability. Is it $\delta$SECR = sqrt($\varepsilon b^2$ + SE$^2$). Just to make sure I got the method correctly and the comment above (ll1) is justified.

**Response:** We can add the formula, even if trivial.

**Comment:** P13 ll15 I would assume a factor of 3 to be very low for the icefields concerning a factor 5 was applied e.g. by Brun et al. 2017 in High Mountain Asia, whereas the variability of SECR patterns in the icefields (especially SPI) is rather high.

**Response:** We provided information on this issue in the response to Referee 1 (referring to his comment on page 13), and detailed information on the estimation of the mass balance gradient and its seasonal

ratio in the Appendix which is included in that response. The correction for the (few) glaciers with gaps during summer accounts for the seasonal difference of the mass balance gradient (ratio summer versus full year). The material presented in the Appendix shows that possible differences of this ratio between individual glaciers are rather small so that an increase of the error by a factor of three is a rather conservative estimate. We also want to point out that this factor does not have a large impact on the VCR rates because the extrapolation refers to a limited subset of the total data sets in respect to area and time span.

**Comment:** P13 ll19 A formula for the complete error propagation throughout mass balance computation would be appropriate.

**Response:** We think that a complete error formula is redundant. If deemed necessary, we can include a formula for the various error components in the Supplement.

**Comment:** P15 ll19 The processes described should be perfectly correct. However, I doubt the values found through the seasonal correction analysis are able to significantly support this interpretation. As mentioned, I assume this daily SECR as a study design feature hard to accept. Also, a precise description of the method that smoothed the SECR field in Fig 8 would be of interest– or even better a display of the original data (SECR field). If it is clearly shown, that the process introduces more precision to the data, than it introduces measurement/ interpolation uncertainty (also regarding comments to 3.3.3) I am willing to accept it. So far, I find it difficult to support it.

**Response:** This comment is related to the issue addressed above (referring to page 13, line 15). As mentioned in the response there, further details on the seasonal correction are included in the response to Referee 1. We will provide further information on the correction procedure in the revised manuscript and/or in the Supplement.

**Comment:** P15 ll28 For the subaqueous loss Abdel Jaber (2016) is referenced. But for the basal cross-sections an original source should be cited.

**Response:** The references for the bathymetric data on the four mentioned glaciers (Upsala, Jorge Montt, Tyndall and Ameghino) will be added. The calving cross sections of these glaciers are deduced from bathymetric data in front of the glaciers and the freeboard. References to the bathymetric data:

Ameghino Glacier: Stuefer, M: Investigations on mass balance and dynamics of Moreno Glacier based on field measurements and satellite imagery, PhD Thesis, Univ. Innsbruck, Austria, 1999. The thesis reports also on two field campaigns on Ameghino Glacier, including pre-frontal bathymetric measurements.

Jorge Montt Glacier: Rivera, A., Koppes, M., Bravo, C. and Aravena, J.C.: Little Ice Age advance and retreat of Glaciar Jorge Montt, Chilean Patagonia. *Clim. Past*, 8, 403–414, 2012.

Tyndall Glacier:  Raymond, C., Neumann, T., Rignot, E., Echelmeyer, K., Rivera A., and Casassa, G.: Retreat of Tyndall Glacier, Patagonia, over the last half century, J. Glaciol., 51(173), 239-247, 2005.

Upsala Glacier: Sugiyama, S., Minowa, M., Sakakibara, D., Skvarca, P. et al.: Thermal structure of proglacial lakes in Patagonia, J. Geophys. Res. Earth Surf., 121, 2270 – 2286, doi:10.1002/2016JF00408, 2016.

**Comment:** P18 ll 9 It is unclear here if that paragraph refers to previous work (Abdel Jaber 2016) or a different publication. Any citation would help. Also I would suggest a reference to the Figures displaying those datasets (provided in the supplement if it is own work)

**Response:** A similar remark was made by Referee #1. The analysis refers to ice flow velocity results external to this study and based on TerraSAR-X. We will reformulate this. Within the ESA project SAMBA (mentioned in the Acknowledgements of the paper) we generated digital maps of ice velocities of SPI and SPI derived from TerraSAR-X and Sentinel-1 data, including time series. Parts of this data set are available to the public at http://cryoportal.enveo.at, but we did not yet have time to write this up for a publication. In Section 5.1 we report some numbers out of these results, in support remarks of the discussion. In principle we can add one or two figures on velocities, preferably in the main paper because these are original results.

**Technical Corrections:**

**Comment:** p4 l1 'Method and error estimation'

**Response:** Not clear what is wrong here.

**Comment:** P7 l1 Check formula. This way it says $\delta$hoff is equal $\delta$hoff times the factor.

**Response:** Assuming p7 ll 11: "." is not a multiplication but a punctuation mark. We will avoid starting the sentence with $\delta$hoff.

**Comment:** Also the distinction, when formulas are a) formatted as objects to be numbered b) written as part of continuous text c) omitted, but have a text description instead is not clearly structured. This should be reconsidered thoroughly.

**Response:** We will improve the manuscript on this issue.

**Comment:** P14 l 'Figs' Fig. /Figure consistency Check throughout the text, also Table /Tab.

**Response:** We will check the Fig/Figs vs Figure. "Tab." does not occur.

---

## Author Response (AR1)

**Author's response on "Heterogeneous spatial and temporal pattern of surface elevation change and mass balance of the Patagonian icefields between 2000 and 2016"**

Authors: Abdel Jaber, W., Rott, H., Floricioiu, D., Wuite, J., Miranda, N.

The Cryosphere Discuss., https://doi.org/10.5194/tc-2018-258

Referee comments are shown in **black**, our response in **blue**, changes in **red**. Line numbers refer to the manuscript version (pdf) of 23 November 2018.

**Anonymous Referee #1**

**General comments:**

**Comment (GC1):** This manuscript applies advanced SAR processing techniques in order to derive spatially detailed maps of surface elevation change (SEC) of the Northern and Southern Patagonian icefields, NPI and SPI respectively. The text is well written, the figures are of good quality, the tables are clear and informative, the topic is of high interest and the results are very interesting, confirming previous studies showing as a whole a strong negative ice volume change with high spatial variability. The authors analyze in detail different sources of errors and gives precise estimations of uncertainties for every studied glacier, different data sets and analyzed periods.

**Response:** Firstly, we want to thank the anonymous Referee for the time and effort put in this detailed and thorough review. We carefully evaluated all comments and suggestions, which are extremely valuable in improving the paper. We are very glad about the positive feedback. We particularly value the Referee's appreciation of the error estimation, in which we invested significant effort and which we believe is one of the strong points of this manuscript with respect to previous studies. Throughout this response, as well as in the paper, we refer to the surface elevation change (in m) as "SEC" and to the surface elevation change rate (in m a-1 or m d-1) as "SECR".

We recognize that the focus of this review is on the seasonal correction which we apply for filling gaps in the rate of surface elevation change, in particular for the four-year time span of epoch 2012 to 2016. This is an important side issue aimed at reducing the uncertainty in mean annual SECR. The main contribution of the paper is the generation of spatially detailed data sets on surface elevation and volume change of both NPI and SPI for two different epochs, including the assessment of communalities and differences in glacier behaviour between the two ice fields and the two epochs. We are not aware of any other publication showing homogenous high resolution data sets on surface elevation or volume change of both icefields for two different epochs.

**Comment (GC2):** The results are not totally novel, since the data sets employed in this manuscript were recently used in a paper published by Malz et al., (2018) with some variations in dates, error assessments and study area. Unfortunately, the results are not totally comparable between them since the glacier

basins and dates are not the same. These differences preclude a precise estimation of discrepancies, but in general the results are statistically similar. The main contribution of Abdel Jaber et al, is their claimed much smaller uncertainties due to several correction that Malz et al didn't applied. After considering the analysis performed by Abdel Jaber et al, I think the error assessment is much more rigorous, effectively addressing many error sources of the data sets, but it is too ambitious when trying to extrapolate parameters from single stations/glaciers for correcting some issues related to the whole icefields. In those cases, is better to live with higher uncertainties and not adding more doubts as I think were added when using Perito Moreno as model for altitudinal gradients for example.

Thanks to the detailed error analysis, this manuscript provides a state-of-the art estimation of surface elevation change of both Patagonia Icefields. Unfortunately, the last data set of 2015 does not come from the end of ablation season, therefore, the authors applied a seasonal corrections to the derived surface elevation changes in order to provide estimates that correspond to full seasonal cycle. This seems to be a weak point of the study design. I guess this was caused by the availability of TanDEM-X imagery, however, employing a more recent datatake from the end of ablation season (later than 2015/16), would have considerably strengthen the importance of this contribution.

**Response:** While in fact some TanDEM-X raw data (Level-0) used in this study are the same as those used by Malz et al., (2018) we would like to highlight the main differences with respect to the latter publication. Malz et al., (2018) provides a SECR with full coverage of SPI for the epoch 2000-2015/2016 only, from which they derive the geodetic mass balance at an icefield level (with uncertainty) and at a basin level (without uncertainty). They furthermore provide two SECR maps covering the periods 2000-2012 and 2012-2015 which are based on SRTM and on the same pairs of TanDEM-X raw data that we use but they are restricted to the southern part of SPI (~2106 km2, ~16% of SPI). Furthermore they do not exploit them to provide any mass balance estimation of the covered glaciers.

We argue that one of the strong points of this study is the use of a homogeneous set of methods and data to achieve SECR maps of both Patagonian icefields, featuring a very high coverage, for two similar observation periods. These maps are used to compute the geodetic mass balance (with uncertainty) of all glacier basins larger than 2 km2 on NPI and larger than 9 km2 on SPI. Hypsometric plots (with uncertainty) for both epochs are also reported for 15 main glaciers of NPI and 24 of SPI.

As correctly noted by the Referee a critical point of our study design is linked to the missing summer days in the "slave" elevation mosaic of the 2012-2015 SEC, caused by its acquisition time in December 2015, whereas February 2016 would have constituted the ideal setup. We hence understand that most of the constructive critique is linked to this issue. We firstly confirm the Referee's presumption: the choice of this dataset was indeed forced by the absence of more suitable coverages of the icefields at the beginning of this study.

As described in the manuscript, the seasonal elevation loss occurring on the icefields during the missing summer days is non-negligible. We show a map of measured seasonal changes on most of the SPI for a different summer, that of 2011/2012 (Fig. 8) which, however, was characterized by the same average air temperature as summer 2015/2016. We are not aware of any published seasonal SECR map of SPI.

Neglecting the impact of summer seasonal elevation variations on the SECR maps and on the corresponding geodetic mass balance would cause some bias, in particular for the second epoch. We

hence devise different correction strategies for different sections of the icefields, according to the availability of SECR datasets and complementary data. We are aware of the limitations of such strategies and take into account these limitations in the error budget. Detailed information on the points correctly raised by the Referee is provided below. We are convinced that the seasonal corrections improve the accuracy of the mean annual SECR compared to the approach by filling temporal gaps with average SECR without taking into account the missing days in the melting season.

**Comment (GC3):** Assuming that the surface elevation changes in summers 2011/2012 and 2015/2016 are equal, based on similarity of monthly mean air temperature records in some neighbouring weather stations is very arguable.

**Response:** We agree that these weather stations, in some distance from the icefields, are not the best choice for checking summer temperature over the icefields. We checked the ERA Interim temperature data, showing agreement within 0.1°C for both summers. The 850 hPa mean summer temperatures (December, January, February) for point 47.25 S, 73.55 W over NPI are 5.9° C (2011/2012) and 6.0° C (2015/2016), for point 50.25 S, 73.55 W over southern SPI 3.6° C (2011/2012) and 3.5° C (2015/2016).

**Changes:** We dropped the station data and report the ERA Interim temperature data together with relevant information in Sect. 3.1.3.

**Comment (GC4):** The other weak aspects of the manuscript are the assumptions regarding altitudinal gradients only supported by Perito Moreno glacier data. This glacier cannot be considered representative for the entire icefield due to extreme longitudinal gradients of climate and mass balance associated with orographic barrier of the Patagonian Andes.

**Response:** Many thanks for stressing this topic. We recognise that the rationale and procedures for the seasonal corrections were not well explained in the first version of the manuscript. Section 3.1.3 contains now a completely revised version describing in detail the strategy, procedures and implications of the seasonal correction. In the Supplement to the manuscript (Section S4) we present an analysis on the applicability of the P. Moreno mass balance gradient for supporting the seasonal corrections on Jorge Montt, S. Rafael, S. Quintin and Benito glaciers (the only glaciers for which this type of correction is applied).

**Changes:** Sect. 3.1.3 (Seasonal correction) was completely re-written in order to assure a better explanation of the applied strategy and methods. An analysis on the applicability of the P. Moreno mass balance gradient for supporting the seasonal corrections on Jorge Montt, S. Rafael, S. Quintin and Benito glaciers is included in the Supplement to the manuscript (Section S4).

**Comment (GC5):** I would be keen to see maps of systematic error similar to Fig S7 that shows random error. This is especially important for some major outlet glaciers of SPI (Jorge Montt, Pio XI, O'Higgins, Viedma and Upsala) where seasonal correction was in the order of several meters (Fig. S4b).

**Response:** The systematic error maps do not feature a strong spatial variation, with exception of regions affected by high probability of signal penetration and regions not covered by the summer 2011/2012 SECR. The systematic error maps are per se not very informative and are hence not included in the Supplement, which file size cannot be further increased without affecting the resolution of the other

figures. We instead preferred providing quantitative values of the different systematic error components in Table S7. The systematic error linked to the seasonal correction is considered as a bulk error derived from the systematic error of the summer 2011/2012 SECR (see Sect. 3.3.3 and also the response to specific comment below), which was increased on specific regions to take into account the extrapolation through hypsometric average values on regions not covered by the summer 2011/2012 SECR. The weighting by the number of corrected days was also applied. The systematic error dominates in the average SECR and VCR of individual glaciers reported in Table 3, 4 and S8 and can hence be easily compared among glaciers and epochs. We also noticed that the description of Figure S4 might have caused some confusion and changed it. Figure S4 represents the rasters (in meters) added during processing to the original SEC rasters (covering the entire observation period) to compensate for the missing/exceeding days.

**Changes: Caption of Fig. S4 changed.**

**Comment (GC6):** This work is largely based on the results and methods reported by Abdel Jaber (2016) PhD thesis. In order to avoid undesirable repetitions, I guess the thesis can be quoted only a couple of times, and then assuming that the results are the one obtained in this manuscript.

In synthesis, I think this manuscript is highly valuable and fits very well with the aim and scope of the journal.

**Response:** This work is based on methods applied in the Abdel Jaber (2016) PhD thesis which were further developed and adapted to this study. Abdel Jaber (2016) reported SECR maps of NPI (2000-2014) and SPI (2000-2011/2012) with mass balance only at an icefield level. The results presented in this study are completely novel (we note that a limited number of TanDEM-X raw data are in common but were newly processed). For this reason in the manuscript we treat Abdel Jaber (2016) as a separate scientific study. The good agreement between the results of this study and those obtained by Abdel Jaber (2016) by separate processing represents, in our opinion, an added value for both publications.

The only results reported in the manuscript and stemming from Abdel Jaber (2016) are the subaqueous ice volume changes and frontal distance variation of the main SPI glaciers in the epoch 2000-2011/2012 (Table S9) and the icefield-wide subaqueous volume change rate reported in Sect. 4.

**Changes**: A clarification on the novelty of the results presented in this manuscript and on the relation to the PhD thesis of Abdel Jaber (2016) was added in the Introduction and adequate reference to the thesis is made throughout the paper where relevant.

**Specific comments:**

Comment: P1 L20: ... and ..., respectively

**Response:** Wording changed to: "They stretch from 46.5° S to 47.5° S and from 48.3° S to 51.6° S, respectively, along the..."

**Changes: as above.**

Comment: P2 L9: any other reference? this is only review

Response: References added:

Åström, J. A., Vallot, D., Schäfer, M., Welty, E. Z., O'Neel, S., Bartholomaus, T.. C., Liu,, Y., Riikilä, T. I., Zwinger, T., Timonen, J., and Moore, J. C.: Termini of calving glaciers as self-organized critical systems, Nature Geoscience, 7, DOI: 10.1038/NGE02290, 2014.

Benn, D. I., Warren, C. R. and Mottram, R. H.: Calving processes and the dynamics of calving glaciers, *Earth-Science Reviews*, 82, 143-179, 2007.

Warren, C. R. and Aniya, M.: The calving glaciers of southern South America, Global and Planetary Change, 22, 59-77, 1999.

Changes: as above.

**Comment:** P3 L8-9: You frequently refer to Abdel Jaber (2016), how different is its work from this submission? I presume is roughly speaking the same.

**Response:** Please see the response to the general comment GC6.

**Comment:** P3 L23-24: Later you refer to Abdel Jaber (2016) as a source for subaqueous ice loss estimates

**Response:** Please see the response to the general comment GC6.

**Comment:** P5 L15-20: Crippen et al. (2016) provided only a general description of NASADEM, and as far as I now, its performance has not been thoroughly compared with SRTMGL1. This I guess is one of main differences of this work compared to a very similar study by Malz et al. (2018) - how does it impact your final results? Were there really less voids compared to SRTMGL1?

**Response:** Voids, mostly caused by phase unwrapping errors, were filled using ASTER GDEM2 in the SRTMGL1 dataset (SRTM version 3). In the NASADEM (available in its void-filled and SRTM-only versions) voids were not very critical on great part of the icefield. We agree with the Referee that the performance of NASADEM has not been assessed yet. We hence performed our own assessment with the main focus being on sources of systematic errors, and particularly long-wavelength elevation biases. While not perfect (particularly around NPI) we concluded that NASADEM is certainly a step forward in this regard compared to SRTMGL1. We hence proceeded with this dataset for the production of the SECR maps and the computation of the mass balance. SECR maps based on SRTMGL1 were not produced; a quantitative comparison of volume change rates is hence not available.

**Comment:** P6 L25: How different were your glacier outlines from those used by Malz et al? The Randolph inventory is known to have problems in many places. Maybe you can discuss about this.

**Response:** Assuming reference to P5 L25. For the glacier outlines of NPI and SPI we use the Randolph Glacier Inventory in its latest version 6 with our own modifications. In RGI v6 significant changes compared to the previous version are found on NPI, where a complete new set of outlines was introduced based on those published by Rivera et al. (2007), while no changes are found on SPI. We are aware of the limitations of the RGI outlines, particularly in the definition of the internal divides between adjacent basins. We improved the external borders of the glacier outlines as described in Sect 2.3,

whereas we did not modify the internal divides. The TanDEM-X DEMs have potential for such an improvement but this exceeded the scope of this study. We preferred to use the latest RGI for comparability of results, limiting changes to the termini.

The SPI outlines of Malz et al. (2018) are reportedly based on RGI version 5 with refinements based on optical images. Some non-negligible differences between our and their glacier outlines can be appreciated by comparing our Figure S9a with their Figure 2b. In particular differences are visible on Bernardo, Tempano, Occidental, Greve glaciers (the latter three appear as a single basin in their paper). Other differences are found on Chico/Viedma glaciers, Europa/Guilardi glaciers, and on smaller glaciers.

**Comment:** P7 L10: There is a problem when quoting equations along the whole manuscript. Only Eq. 2 is mentioned in the text and some equations are between lines without label numbers, don't helping in the fluent reading process. You refer here to Eq. (2) well before introducing it (and before Eq. (1))

**Response:** Agreed, many equations were written inline in order to reduce space and provide a flowing narrative. We will improve the manuscript on this issue.

**Changes:** The most relevant equations are now displayed on their own line and numbered, whether they are cited or not.

**Comment:** P8 L20. The temperature and balance altitudinal gradients in the SPI are highly different between east-west margins or northern - southern parts etc. Maybe you can check a recent paper by Bravo et al in JGR (DOI: 10.1029/2018JD028857) and comment on this.

**Response:** Many thanks for referring to this recently published paper providing very valuable data for advancing the modelling of surface/atmosphere exchange processes and surface mass balance across the icefield. We performed computations on mass balance gradients using different lapse rates on east and west coast glaciers and comparisons with Moreno balance gradients which happen to support our approach. See details in response to general comment GC3 and in Supplement Section S4.

**Changes:** We clarified this point in the re-written Sect. 3.1.3 and in the Supplement (S4), including the suggested reference to Bravo et al. (2019).

**Comment:** P8 23-36: Can you please clarify this part? For example Perito Moreno and Jorge Montt have very different climatic setting (Lenaerts et al., 2014), I wonder if Perito Moreno is a best choice for a reference in this case.

**Response:** See the detailed descriptions and checks in Section 3.1.3 and Supplement (S4) and the response to comment P8 L20.

**Changes:** Information on mass balance gradients for different glaciers and checks on representativeness of Moreno data are reported in Supplement S4.

**Comment:** P8 L30-32: Again, issue of transferability of parameters of Perito Moreno to entire SPI. Why was it not necessary for NPI, because of the day of data take, I guess?

**Response:** The 2000-2012 SECR of NPI was not corrected because the SRTM and the TanDEM-X data takes were acquired approximately in the same days of February. The seasonal changes on the scene with index 1, not involving summer days, were deemed to be negligible and much less than the systematic errors on a 12-year time frame. Furthermore, the seasonal ratio in ablation rate (summer vs. full year), based on P. Moreno measurements, is not used for supporting the seasonal correction of the full icefields, but only for four glaciers. See also the detailed descriptions in Section 3.1.3 and Supplement (S4) and the response to comments above.

**Comment:** P8 L36-37; Fig S4: There are sharp boundaries between zones corresponding to different time spans, do they propagate to the final product introducing discontinuities?

**Response:** The discontinuities are caused by the different  $\Delta t$  in days to be corrected between adjacent master-slave pairs. These do not propagate in a noticeable way to the final SECR product. In fact light discontinuities might already be present in the uncorrected SECR, the correction would, at least in theory, reduce such discontinuities by compensating different amplitude of seasonal changes according to the actual  $\Delta t$  in days.

**Comment:** P9 L19: Uncertainty bound on glacier-wide density seems to be too low. Cogley (2009) refer to Sapiano et al. (1998) 6% estimate as reasonable. In similar work, Malz et al. (2018) provide three scenarios of different densities, it is their main source of uncertainty for the final results.

**Response:** The main scope of this study is to provide reliable volume change rate estimates at a basin scale and the core of the error estimation focuses on the volume change rate estimates. (Tables 2, 3, S8). These can then be converted to mass change rates using a constant density assumption with a corresponding error of choice. Furthermore we also provide a reference mass change rate at an icefield level (Table 1). For this purpose we used in the first version of the paper the common scenario of glacierwide density of 900 kg m-3 facilitating the comparison of results with other studies. We agree that the assigned error of  $\pm 17$  kg m-3 is a small one for firn areas. The high mass loss rates on the Patagonian icefields refer to ice areas. Therefore it makes sense employing separate density estimates for the volumes in ice areas and firn areas. We use  $\pm 17$  kg m-3 in the ice areas and  $\pm 54$  kg m-3 in the firn areas, assuming a mean elevation of 1150 m on NPI and 1050 m on SPI for separating ice and firn areas. The resulting uncertainty in density for icefield-wide mass changes is obtained by rounding up to  $\pm 36$  kg m-3 (4 %).

**Changes:** This point is taken into account by employing an error on the density of 36 kg m-3 (4 %) for icefield-wide mass changes reported in Table 1. This value (together with explanation) is not cited anymore in Sect. 3.1.4 but appears only in Sect 3.3.4 to avoid repetition.

Comment: P10 L7-10: Please back it up with some reference

Response: References added:

Garreaud, R., Lopez, P., Minvielle, M., and Rojas, M.: Large Scale Control on the Patagonia Climate, J. Climate, 26, 215–230, 2012.

**Schaefer, M., Machguth, H., Falvey, M., and Casassa, G.: Modeling past and future surface mass balance of the Northern Patagonian Icefield, J. Geophys. Res.-Earth, 118, 571–588, doi:10.1002/jgrf.20038, 2013.**

**Changes: as above.**

**Comment:** P10 L24: Finally, what exactly was the criterion for masking regions prone to penetration? Was it only manually outlined based on expert knowledge?

**Response:** In each DEM mosaic, regions prone to signal penetration were not masked-out, instead they were manually outlined according to backscatter intensity and assigned a certain penetration bias, which was then used to compute the corresponding systematic error component of the SECR (Sect. 3.3.3). The penetration bias was assigned based on the average  $\sigma^0$  within the region. Measurements of signal penetration in TanDEM-X data over NPI at varying  $\sigma^0$  are reported in Abdel Jaber (2016). These, together with knowledge on relations between X-band  $\sigma^{\circ}$  and signal penetration length (see e.g. Mätzler, 1987) were used to assign the elevation bias taking into account the relation between penetration length location of the scattering phase center within the snow pack. During summer the top snow layers on the main ice plateau are either wet (low  $\sigma^{\circ}$ ) or include melt/freeze metamorphic layers with rather small penetration for X-band signals also in frozen state (Reber et al., 1987).

Mätzler, C.: Applications of the interaction of microwaves with the natural snow cover, Remote Sensing Review, 2, 259-387, 1987

Reber, C., Mätzler, C., and Schanda, E.: Microwave signatures of snow crusts, modelling and measurements, Int. J. Remote Sensing, 8 (11), 1649 – 1665, 1987.

Changes: Section 3.2 was improved including the aspects and references reported in this response.

**Comment:** P11 L8-9: See comment above.**

**Response:** The average  $\sigma^0$  of the SRTM acquisitions were analyzed to reach this conclusion. This is another novel aspect of this publication; we are not aware of an empirical assessment of the backscatter of C-band SRTM to assess possible signal penetration in glaciological remote sensing studies. Relations between snow wetness, penetration depth and backscatter intensity can be looked up in radar textbooks (e.g. Ulaby and Long, 2014). Furthermore, the good agreement between our volume change rates 2000 to 2012 over NPI and the results of Dussaillant et al. (2018) based on optical data confirm the validity of our approach regarding signal penetration. See also their conclusion: " our study confirms the lack of penetration of the C-band SRTM radar signal into the NPI snow and firn except for a region above 2,900 m a.s.l.".

**Ulaby, F.T and Long, D.G.: Microwave Radar and Radiometric Remote Sensing, The Univ. of Michigan Press, Ann Arbor, 2014.**

**Comment:** P11 L11: Bippus (2007) assumed this lapse-rate for summer season on Perito Moreno, however as far as I know this value was not based on measurements. Additionally, she accounted for an

off-glacier location of AWS, resulting in additional temperature offset. Maybe you can compare your numbers with Bravo et al 2019

**Response:** Thank you for this input. However, we want to stress that the conclusion about signal penetration is primarily based on the backscatter assessment. Regarding penetration see also the comment on P11, L8-9.

**Changes:** Based on our response the reference to Bippus (2007) was removed and the explanation made clearer. The suggested reference is included in the manuscript.

Comment: P12 L26: Is 0.1 m based on literature?

**Response:** The bulk systematic error of 0.1 m for the remaining pixels above 1000 m a.s.l. accounts for undetected regions (a very small percentage of the total area) and possible small offsets for areas with refrozen upper layer of snow and firn, affecting only very small areas; see response above. The typical X-band one-way penetration depth of a frozen crust of 10 cm thickness is about 0.1 m, increasing with decreasing crust thickness (Reber et al, 1987).

**Changes:** We added here a reference to Sect. 3.2 which refers to the response above, and the corresponding reference to Reber et al. (1987).

**Comment:** P13 L1-5: I think that the error linked to the seasonal correction may be underestimated as it does not seem to cover all uncertainties related to the transferability of hypsometric averages shown in Fig. 8 (see previous comments).

**Response**: For details see the response to the general comment on seasonal correction (Sect. 3.1.3) and Sect. 3.3.3 on error analysis. The systematic error linked to the seasonal correction is derived from the systematic error for the summer 2011/2012 SECR used for the correction on most of the glaciers. This systematic error was increased to account for the different year to correct, although summer air temperatures were nearly the same (see response above). It was also increased on regions where the summer 2011/2012 SECR has no coverage (NPI, north-west SPI and many gaps throughout SPI) in order to account for the transferability of hypsometric averages. Such increases might even lead to an overestimation of this error source; this is accepted as it tends towards a more conservative error budget.

**Changes:** Revisions and clarifications on seasonal correction have been implemented in Sect. 3.1.3 and on related error analysis in Sect. 3.3. See response to related comments above.

**Comment:** P15 L26: It that is true than your seasonal correction should use lower density in accumulation area.

**Response:** P15 L26 does neither refer to seasonal correction nor to density in the accumulation area. It is a statement pointing out that measured (or modelled) surface ablation cannot be directly converted into SEC, as it is necessary to account for emergence or submergence. Regarding the question on the density used: there is no need for using snow or ice density in SEC retrieval by means of DEM differencing. The

correction refers to seasonal differences (summer vs. rest of the year) in surface elevation and fills the missing days in observed SEC.

**Changes:** Revisions and clarifications on seasonal correction have been implemented in Sect. 3.1.3. The procedure of separate accounting for seasonal corrections of SEC due to surface melt and submergence is detailed in Supplement, Section S4. The latter procedure is applied to those four glaciers with significant dynamic downwasting that are not (completely) covered by the summer 2011/12 SEC maps.

Comment: P15 L30: Perito Moreno glacier

Response: Corrected.

Changes: as above.

Comment: P16 L28: Results were recently published in Frontiers - Langhamer et al. (2018)

**Response:** Thank you; we included the reference to this publication in the revised paper.

Changes: as above.

**Comment:** P16 L37: Again, I doubt that Perito Moreno is representative for entire SPI and SPI, Steufer (2007)

**Response:** This comment refers probably to P16 L30 (there is no L37): "Assuming a degree-day factor of 0.7 cmd-1 on ice areas (Stuefer et al., 2007), the melt loss for an increase of surface temperature by 0.7 °C during November to March corresponds to an additional loss ...... ". This average degree-day factor, based on ablation measurements on Perito Moreno Glacier over several years, is similar to the average degree-day factor of 0.65 cmd-1 reported by Rivera (2004) for Chico Glacier. Consequently this factor can be well used for an estimate on the impact of melt due to different summer temperatures in the discussion.

**Comment:** P17 L26: This issue is a critical factor in the whole analysis of the elevation changes in the high plateau of the icefields. We know that the accumulation is extremely high, an in between few days you can have huge accumulation events. I think this high temporal variation of snow fall must be taken into account. See Schwikowski et al 2013 for snow accumulation on the SPI.

**Response:** At first, we want to point out that we are well aware on the spatial and temporal variability of accumulation on SPI. The work reported in this paper (as in other papers on SPI or NPI mass balance) does not deal with single events, but addresses the retrieval of spatially detailed maps on changes in surface elevation and volume over SPI and NPI, providing mean values over epochs of 12 years and 4 years. The contribution of single events during this period is implicitly included in this analysis. Regarding spatial variability, this is fully taken into account by the high spatial resolution of the TanDEM-X elevation data and the high percentage of spatial coverage (Tables 2 and 3). Regarding the summer periods 2011/2012 vs. 2015/2016, there is no indication on exceptional events as according to ERA Interim there is perfect agreement for air temperature and the difference in precipitation between these two

summers is 15 % (within the uncertainty of precipitation estimates for this region of complex topography).

**Comment:** P18 9? What analysis? It is missing in methods and results sections. Maybe you wanted to quote Abdel jabber 2012?

**Response:** Assuming this comment refers to P18 L3, L4. Thanks for pointing out the lack of information on the source of the velocity data used in the Section 5 (Discussion) for supporting the discussion on differences in volume change between the two icefields and two epochs. The analysis refers to novel ice flow velocity results from a study which we performed complementary to the work on DEM differencing. It is based on TerraSAR-X repeat-pass SAR data, as explained now in the revised text. Within the ESA project SAMBA (mentioned in the Acknowledgements of the paper) we generated digital maps of ice velocities of SPI and NPI derived from TerraSAR-X and Sentinel-1 data, including time series. In Section 5.1 we report some numbers out of these results for supporting the discussion. We added one figure with velocities for four main glaciers in different years.

**Changes:** We explain the origin of the velocity results at the beginning of Sect. 5.1 and include a new figure which shows results for four main glaciers of SPI.

**Comment:** P18 L25: See comment above P33 Fig.4: Why is there a sharp transition in the terminal part of the glacier on panel a? Frontal retreat I guess?

**Response:** Yes, exactly. The sharp transition is in fact a physical signal in the elevation difference between the glacier front in 2012 (abrupt step in elevation) and what in year 2000 was the glacier surface (smooth increase in elevation). **References mentioned in this review:**

Bravo, C. et al (2019) Air Temperature Characteristics, Distribution and Impact on Modeled Ablation for the South Patagonia Icefield. JGR, DOI: 10.1029/2018JD028857

Langhamer, L., Sauter, T., & Mayr, G. J. (2018). Lagrangian Detection of Moisture Sources for the Southern Patagonia Icefield (1979-2017). Frontiers in Earth Science, https://doi.org/10.3389/feart.2018.00219

Lenaerts, J. T., Van Den Broeke, M. R., van Wessem, J. M., van de Berg, W. J., van Meijgaard, E., van Ulft, L. H., & Schaefer, M. (2014). Extreme precipitation and climate gradients in Patagonia revealed by highresolution regional atmospheric climate modeling. Journal of Climate, 27(12), 4607-4621.

Sapiano, J.J., W.D. Harrison and K.A. Echelmeyer. 1998. Elevation, volume and terminus changes of nine glaciers in North America. Journal of Glaciology, 44(146), 119-135.

Schwikowski, M., M. Schläppi, P. Santibañez, A. Rivera and Casassa G. (2013): "Net accumulation rates derived from ice core stable isotope records of Pío XI glacier, Southern Patagonia Icefield". The Cryosphere 7, 1635-1644. doi.org/10.5194/tc-7-1635-2013

**Anonymous Referee #2**

**General comments:**

Comment (GC1): The study presented by Wael Abdel Jaber and co-authors is an overview of surface elevation change rate (SECR) and geodetic mass balance (MB) values for the Southern Patagonia Icefield (SPI) and Northern Patagonia Icefield (NPI) for the two epochs 2000-2012 and 2012-2016. The results are calculated on the entire icefield as well as on glacier basis, mean SECR and volume change rates (VCR) are listed in a table including observed area and error budget. For most important glaciers the hypsometric distribution of those variables is depicted in graphs. The study provides a detailed description of the error analysis and several steps to correct for biases and penetration and ablation uncertainties. The language is correct and understandable. The subject is of high interest to the community, the method and study areas are not completely novel. In the last years, there have been publications covering the study area with the same topic (Foresta, Dussaillant, Malz, Abdel Jaber himself), but partly using different approaches. This new study cites and discusses those adequately. I recommend to add the recent work of Braun et al. (2019) which also includes SPI and NPI, but only covers the first observation period (2000-2011/15). The authors point out two aspect as main progress to previous studies: 1) The comprehensive and simultaneous observation of both icefields at two epochs. 2) The variety of corrections and assumptions made to guarantee a precise observation of SECR and following products. The line of argumentation is clear as far as (1) is concerned and thus I support publication in TC. Nevertheless, concerning (2), revisions should be performed to significantly improve the traceability of results and assure the validity of some of the applied steps described in the method section before publication.

**Response:** We thank the anonymous referee for the detailed review and the appreciation of our work. Although similar studies were published already we want to point out the main novelties of our manuscript. We provide the first geodetic mass balance for NPI and SPI also for a recent epoch (2012-2016) by TanDEM-X DEM differencing and discuss causes for differences between the two icefields and the two epochs. Besides the entire icefields we give average SECR and VCR (incl. error) for individual glacier basins (up to 9km2 on SPI and 2km2 on NPI) and hypsometric plots of main glaciers (incl. error bars). We used the same method as for the preceding epoch (2000-2012) and this allows the comparison of individual glacier and icefield behavior in the two epochs. Also we present an up to now unique analysis of the backscatter coefficients of all SAR acquisitions (SRTM and TanDEM-X) to assess the error due to signal penetration, a known issue when using InSAR based DEMs. Abdel Jaber et al. (2016) is a doctoral thesis. It has not been published in any scientific journal, neither in its entirety nor any part of it, but it is available online to everybody. The thesis, reporting many details on the methods used for SECR and VCR, provides also the basis for the technical approach applied in this paper. This review asks for many details on techniques for TanDEM-X DEM differencing and retrieval of SECR which are relevant for a technical paper on DEM differencing. As this is not the main scope of our paper, we tried to focus in the methodology sections on essential points, nevertheless resulting in an already quite comprehensive description. We provide in this response information on specific technical issues raised by the referee and explain how these we taken into account in the revisions.

Thanks also for pointing to the recently published paper by Braun et al. (2019). We make reference to this paper. Regarding DEM differencing SRTM-TanDEM-X, for SPI, the numbers seem to be based on Malz et al. (2018). The reported number for NPI 2000 - 2011/2015 lacks full traceability regarding the TanDEM-X data used and processing methods so that in depth comparison with our results is not possible.

**Changes:** We took these issues into account for paper revision (see response to specific comments). We added the reference to Braun et al. (2019) in Sect. 5.2.

**Comment (GC2):** Methods: The utilization of several thresholds or distinct values is not always transparently explained. At some decisive points, it remains vague if the method or decision follows own reasoning, own previous work or an external reference (cf. specific comments)

**Response:** We thank the referee for pointing out this aspect. We provide relevant information in the response to specific comments and in the revised paper.

**Comment (GC3):** The correction for the observation date in epoch 2, for not being at the end of ablation period, is an unprecedented venture. However, it forms also a weak point of the study. In the reviewer's opinion, the error induced to the SECR (Epoch 2 – Epoch1) by this step is not adequately represented by the mapped datasets nor is it transparently addressed as error contribution in the text. Moreover, an interpolation of missing areas based on only two weather stations and adjusted to sparse hypsometrical patterns has to be regarded rather experimental compared to the robust methodology used for the rest of the study. It is hard to judge the validity of the seasonal correction. A  $\Delta h$  map outside the icefields and the unfiltered dataset  $\Delta h$  could help justifying, at least for the observed parts(cf. specific comments)

**Response:** We are aware that the correction applied for the missing days in the ablation season to complete the 4 years period of epoch 2 has some limitations. On the other hand performing such a correction for the short period is fundamental for obtaining reliable annual SECR for comparisons with other results. This reduces a possible bias compared to the case without seasonal correction. Regarding the station data: we replaced these by ERA Interim data over the icefields.

We recognise that the rationale and procedures for the seasonal corrections were not well explained in the first version of the manuscript. Section 3.1.3 contains now a completely revised version describing in detail the strategy, procedures and implications of the seasonal correction. In the Supplement to the manuscript (Section S4) we present an analysis on the applicability of the P. Moreno mass balance gradient for supporting the seasonal corrections on Jorge Montt, S. Rafael, S. Quintin and Benito glaciers (the only glaciers for which this type of correction is applied). Regarding the impact of the seasonal correction on the error budget, this is specified in Table S4 at the icefield level and taken into account in the total error budget for the icefields and the individual glaciers. The spatial patterns on the magnitude of the seasonal correction are shown in Fig. S4. Outside the icefields no seasonal correction is applied. Therefore additional delta h maps are not deemed to be necessary, in particular as they would further inflate already a long paper and Supplement. Concerning the unfiltered Summer 2011/2012 SECR please see the answer to corresponding specific comment.

**Changes:** Sect. 3.1.3 on seasonal correction was completely re-written. Further information is supplied in the Supplement, Section S4.

**Comment (GC4):** The error indicated for SECR is spectacularly low in this paper. Although there is a section explaining the calculation it is not totally clear, why a DEM comparison could come up with such low elevation error budget. It appears, the systematic error budget, as the main contributor, is calculated partly in favor of a small total error. Some steps along this path should be under discussion or described in more detail for traceability (cf. specific comments).

**Response:** The errors of SECR and of VCR (Table 1) are comparable to most of the recent results obtained by other authors based on elevation change approach (see Table S10). Therefore we do not understand the reason for the reviewer's statements "spectacularly low" error and "the systematic error is calculated in favor of a small total error". Furthermore, the agreement between our volume change rate 2000 to 2012 over NPI and the results of Dussaillant et al. (2018) is well within the combined error bound. This supports the validity of our error estimate, as the results of Dussaillant at al. are based on completely different data sets and methods. As suggested, we provide further clarification on the error estimate in the revised manuscript and Supplement.

**Comment (GC5):** Structure: The work is based on the PhD thesis of Abdel Jaber (2016). However, since it is sometimes difficult to follow what is actually new in contrast to what was already in place, that presents the reader with challenges. A clear line between parts that were newly implemented and those that were adopted needs to be drawn by the authors. I recommend that the authors revise the methods and result section with regard to this aspect to make the paper a full stand-alone document. This also concerns the length of some descriptions that could be kept more concise for this paper, with reference to the thesis (or other original source).

**Response:** In our view the first version of the paper is already a full stand-alone document, as all essential information on scientific background, methods, results, discussion, etc. is presented in a logical and traceable way, including references to the sources. As mentioned before, the PhD thesis of Abdel Jaber (2016), nor parts of it, have been submitted to or published in any journal. In this respect everything reported out of the thesis would be novel for a journal paper (with adequate reference to the thesis). However, the results on surface elevation and volume change presented in this study are completely novel also compared to the thesis which covered SECR maps of NPI (2000-2014) and SPI (2000-2011/2012) with mass balance only at an icefield level. Only a limited number of TanDEM-X raw data are in common, but these were newly processed. For this reason we refer in the manuscript to Abdel Jaber (2016) as a separate scientific study. The good agreement between the results of this study and those obtained by Abdel Jaber (2016) by separate processing represents, in our opinion, an added value for both publications. The methodology builds on the development work for the thesis with some further evolution. This paper includes an overview on key components of the method, with reference to the thesis where the reader can look up details. However, we realize that the related explanations are not clear enough in the first version of the paper and took this into account in the revisions.

**Changes:** A clarification on the novelty of the results presented in this manuscript and on the relation to the PhD thesis of Abdel Jaber (2016) was added in the Introduction and adequate reference to the thesis is made throughout the paper where relevant.

**Specific comments:**

**Comment:** P 6 l27 ...(in order of impact, the latter being negligible in our Raw DEMs)." This and further statements could be corroborated by a similar Figure as Fig S 2 for SRTMTDM, displaying same  $\Delta h$  for outside the icefields for SRTM-TDM(Ep1) and TDM(Ep1)-TDM(Ep2).

**Response:** Off-glacier SECR close to some termini are included for this purpose in the detailed maps from different locations of the icefield shown in Figures 2 - 7. Adding these four figures would inflate the already extensive size of the paper. Furthermore the file size of the Supplement cannot be increased further without affecting the resolution of the other figures.

**Comment:** P 7 II3 -13 The weighted averaging of the offset values leaves the question if a spatial pattern was analysed and fitted by an offset function. A simple averaging could lead to regional maladjustment, if the sign / magnitude of the offset is a function of geographic position (tilted dataset, described in this manuscript p6 I20). For the precision of the applied method a mapped  $\Delta h$  (cf. comment to P6 II26) could be convincing.

**Response:** See previous response.**

**Comment:**\_P7 II13 How is the absence of horizontal shifts checked? The detection is slope dependent (cf. Nuth and Kääb (2011)), thus cannot be efficiently performed on an area without slope as the CRs (avr. Slope below 4.)

**Response:** The horizontal shifts (in our case possibly acting in the ground range direction) were not checked analytically directly on our datasets, but relying on visual analysis of all available off-glacier terrain. Analytical checks using the method of Nuth and Kääb (2011) was done for the TDM-SRTM SEC datasets during the preparation of the thesis (Abdel Jaber et al., 2016) corroborating the validity of this calibration procedure. Because the same method was applied for this paper as for the thesis, the conclusions regarding this procedure can be adopted for this work.

**Comment: p7 II23 Please provide reference**

**Response:** Rivera (2004) Fig. 6.3 shows in January higher density in the upper metres of snowpits in the accumulation area of Chico glacier, compared to density in September and October. We added the reference:

*Rivera, A.: Mass balance investigations at Glaciar Chico, Southern Patagonia Icefield, Chile, Ph.D. thesis, Univ. of Bristol, UK, 2004.*

**Changes: as above.**

**Comment:** P 7 II30 What kind of filtering was applied? It would be interesting to see the original dataset and a  $\Delta$ h map outside the icefields.

**Response:** Since the Summer 2011/2012 daily SECR is used only for the seasonal correction, we applied the following procedure for eliminating outliers: (i) conservative masking on glaciated terrain of regions with high backscattering and peaks in the daily SECR values followed by (ii) 2-step filtering with sliding

window: (a) median filters with kernel size 9 and (b) smoothing with kernel size 9. The raster posting is 0.4 arcsec. This way the localized seasonal changes or outliers were eliminated and thus the SECR map can be used for the purpose of compensating the temporal gap in 2015/2016. Such a figure representing the unfiltered Summer 2011/2012 SECR would not provide any additional significant scientific or methodological contribution as it is a minor technical detail of rather limited impact.

**Comment:** P8 ll10 What does similar mean here? +-0°C? Please add a number for consistency.**

**Response:** The 2 stations data we used are confirmed by ERA Interim temperature data (see also response AC#1 page 3) which provide even higher agreement between the summer epochs. According to ERA Interim the average temperatures of summer 2011/2012 and summer 2015/2016 agree within 0.1°C. This means that the SECR maps of summer 2011/2012 (scaled to the length of the missing period) can well be used as substitute for the missing days in summer 2015/2016. We will add this info in this paragraph.

**Changes:** The reference to the two stations was dropped and reference is made to the ERA Interim data in the re-written Sect. 3.1.3.**

**Comment:**\_P7 II32 -p8 37 A comprehensive series of comments concerning the temperature variability and spatially variable ablation patterns resulting in a rather speculative adjustment in the seasonal correction section is given by referee #1. I agree on those.

**Response:** Please see –our response to your general comment GC3 that refers to the same issue.**

**Changes**: Sect. 3.1.3 on seasonal correction was completely re-written to better explain the rationale and strategy of the seasonal correction. Supplement, Section S4, provides analysis on the applicability of the P. Moreno mass balance gradient for supporting the seasonal corrections on Jorge Montt, S. Rafael, S. Quintin and Benito glaciers (the only glaciers for which this type of correction is applied), confirming the validity of this approach. As mentioned before, there is no other glacier on SPI or NPI (except P. Moreno Glacier) for which multi-year seasonal and annual ablation measurements are available and a seasonal ratio for ablation (summer vs. full year) based on observation is available.

**Comment:**\_P8 ll28-32 Please explain the justification of 20% reduction in correlation to a temperature value. Based on what assumption does it translate into a percentage?

**Response:** For epoch 1, although the daily SECR is from the same year (ablation season 2011/2012) and was obtained from December to March, the days which have to be compensated are in late summer and therefore we reduced the estimate for ablation by 20 % compared to the summer average. As we mentioned in the manuscript, this scaling factor is based on a time series of daily air temperature measurements from 1995 to 2003 near the front of Perito Moreno Glacier and ablation measurements on the terminus (Stuefer et al., 2007). Furthermore, we want to point out that this correction factor, applied on the hypsometric curve in Fig S3, affects only a very small area of the icefield.

Changes: The justification was added in the re-written Sect. 3.1.3.

**Comment:**\_P9 II8 Can you please add more information to increase reproducibility when data gets available: what threshold on SEC values? What morphological operators?

**Response:** We did not include these details because we do not think that this is an interesting point and would inflate an already very long paper. For each of the 4 SECR maps we produced a raster starting from the flag mask (FLM) layer that resulted from the processing with ITP which provides roughly the regions affected by layover and shadow. Thresholds  $\Delta h/\Delta t < -10$  m/a and > +6 m/a were applied. A morphological operator of closing followed by a 5 x 5 median filter was applied on the mask raster in order to "clean" the mask, avoiding noise due to thresholding.

**Changes:** This is a minor methodological step; we avoid entering in such details in the manuscript given its length and scope. The procedure can be found in the response above.

**Comment:** P9 II16-19 Where are the 17 kg m-3 uncertainty taken from? Citation of Cogley et al. (2009) is misleading here, because reader would expect a reference for the density uncertainty. I found it to be mentioned in Abdel Jaber (2016), but it seems to be taken from Gardner et al. (2012) – this is not referenced here. Anyway, why using this value when recent large area studies like Brun et al. (2017), Dussaillant et al. (2018), Malz et al. (2018) use 60 kg m-3? Choosing that latter value would lead to comparable error budget.

**Response:** Yes, the uncertainty  $\pm 17$  kg m-3 comes from (Gardner et al, 2012). Regarding this issue, we want to mention again the statement in response to Referee 1, that he main scope of this study is to provide volume change rate estimates at basin scale (Tables 2, 3, S8). These can then be converted to mass change rates using a constant density assumption (which provides full traceability). Furthermore we provide a reference mass change rate at an icefield level; these are the only four numbers in the paper reporting mass change estimates (Table 1). We decided to use the common scenario of glacier-wide density of 900 kg m-3 facilitating the comparison of results with other studies. We agree that the assigned error of 17 kg m-3 is a small one (1.8%) for firn areas. However, large mass losses on the Patagonian icefields refer to ice areas. Therefore it makes sense employing separate density estimates for the volumes in ice areas and firn areas. We use  $\pm 17$  kg m-3 in the ice areas and  $\pm 54$  kg m-3 in the firn areas, assuming a mean elevation of 1050 m on NPI and 1150 m on NPI for separating ice and firn areas. The resulting uncertainty in density for icefield-wide mass changes is  $\pm 36$  kg m-3 (4 %).

**Changes:** This point is taken into account by employing an error on the density of 36 kg m-3 (4 %) for icefield-wide mass changes. This value (together with explanation) is not cited anymore in Sect. 3.1.4 but appears only in Sect 3.3.4 to avoid repetition.

**Comment: P9 II 31 Please provide reference**

**Response:** The following reference will be added:

DLR-CAF. 2013 (October).TerraSAR-X Ground Segment Basic Product Specification Document. 1.9 edn. German Aerospace Center (DLR) - Cluster Applied Remote Sensing (CAF). TX-GS-DD-3302.

Changes: as above.

**Comment:**\_p10 ll21-24 and P11 ll8 Why manual outlines? What is the decision to delimit these areas based on? If that information can be found in Abdel Jaber (2016) it should be indicated (or the original study it referes to).

**Response:** The outlining was performed manually based on the backscatter coefficient ( $\sigma^0$ ) and taking also into account the corresponding elevation to ensure that areas of high backscattering are on the smooth firn plateau and not in the ice areas. A fixed thresholding of the  $\sigma^0$  layers would have added noise because this would include regions of rough glacier ice. The penetration height offsets assigned to each region are based on the relation between difference in  $\sigma^{\circ}$  for dry and wet snow and  $\Delta h$  (Figure 8.12 of Abdel Jaber (2016), Sect. 8.4).

**Comment:**\_P11 II3-7 Is any of the values mentioned in these paragraphs used for determining the outlines? What is the interpretation of the sigma0 ranges based on? Abdel Jaber (2016) / other? Please reference it.

**Response:** The  $\sigma^{\circ}$ -values and related interpretation are based on multi-year experimental and theoretical work on X-band and C-band radar signal interaction with snow and ice by two of the co-authors, including several field campaigns on Alpine glaciers related to ERS-1/ERS-2, ASAR of Envisat, Shuttle Radar SIR-C/ X-SAR SRL-1 and -2, and TerraSAR-X. See e.g. Nagler and Rott (2000); Floricioiu and Rott (2001). Relations between snow wetness, snow morphology, penetration depth and backscatter intensity can also be looked up in radar textbooks (e.g. Ulaby and Long, 2014). Concerning the SRTM data (Sect.3.2.2), to which this comment refers, such an analysis was already performed in (Abdel Jaber, 2016) and the processing was not repeated for his study. Only the analysis of TDM backscatter (Sect. 3.2.1) is new because the data used in this study have not been used in Abdel Jaber (2016).

Floricioiu D. and Rott, H.: Seasonal and short-term variability of multifrequency, polarimetric radar backscatter of alpine terrain from SIR-C/X-SAR and AIRSAR data. IEEE Trans. Geosc. Rem. Sens., Vol.39(12), 2634–2648, 2001.

*Nagler T. and Rott, H.: Retrieval of wet snow by means of multitemporal SAR data. IEEE Trans. Geosc. Rem. Sens., Vol 38(2), 754-76, 2000.*

Ulaby, F.T and Long, D.G.: Microwave Radar and Radiometric Remote Sensing, The Univ. of Michigan Press, Ann Arbor, 2014.

Changes: This issue has been clarified in Sect. 3.2 and references have been added.

**Comment:** P11II 17 First sentence would be well supported by a formula. Is the HEM for the TDM elevations calculated by the phase difference to the interferometric phase of 12 m TDM products? Is it always TDM 12m as a reference (also in 3.3.3 (1))? It is mentioned once briefly in 2.1, but I think I should be emphasised there, that it is especially used as reference for elevation error assessment.

**Response:** The HEM does not depend on the reference DEM (the global TDM DEM) but it is processed by ITP for each TDM RAW DEM. The HEM is only the interferometric error and reflects point per point the actual error. This is the alternative of computing it over ice free terrain, as other authors did. This error is

used to compute the random error of each sample. When averaging on an elevation bin this error becomes negligible compared to the systematic components.

In 2.1 (p 4 lines 13-14) we state only that the global TDM DEM is used as reference for the processing, the details of how this is used are given in 3.1.1. It is also used for the DEM coregistration (see sect. 3.1.2 p 6 lines 24-25). We never mention that we used the global TDM DEM in error calculation.

**Changes:** We improved the explanation and added the requested formula even if it is a basic uncertainty propagation rule, but we will avoid repeating the quadrature sum formula in the rest of the Section.

**Comment:** P11 II25: How was it included? Add some mathematical explanation of the error propagation through seasonal correction. Is it sqrt( $\sigma$ t1 2 +  $\sigma$ t2 2 + $\sigma$ seas 2) for each pixel?

**Response:** The random error (addressed in this comment), using pixelwise correction, is applied for the portions of SPI which are covered by the summer 2011/2012 SECR. It is included in the formula for the total random error as shown in the comment above. We do not see the need for adding also this formula as it refers to basic uncertainty propagation, already described by the (standard) formula provided in response to the comment P11 L17.

**Changes:**. We changed the text to better clarify this operation.

**Comment:**\_P12 II14 Enhance precise and illustrative explanation to this whole section 3.3.3. The reader is interested how exactly the systematic error is calculated, for it is key to the low elevation error budget presented. Please provide formulas to enhance comprehensibility. That could spare some explanatory text passages, that are less illustrative.

**Response:** As already explained in the response to general comment GC4, the elevation error budget is not particularly low, being in line with several other studies. We agree with the request to improve the comprehensibility of Section 3.3.3 and revised this section, trying to address a broad community interested in mapping of glacier volume change.

Changes: Sect. 3.3.3 was revised in order to improve its clarity.

**Comment:** p 12 II14 Is the IQR of the areas that were adjusted (CR, calibration) addressed as the measure of error (validation) on each DEM? I do not agree with this method from a scientific perspective. On top, choosing the IQR reduces or eliminate slope dependent effects (avr. Slope below 4°, IQR slope?). But on glacier these are present for sure, so they are a source of systematic error to be addressed in the budget. It would be more reliable if validation is performed on the entire DEM (glaciers excluded), but assessed with regard to absolute elevation and slope.

**Response:** Assuming the comment concerns p 12 ll17-20.**

The IQR is chosen to characterize the spread of the elevation difference between the TDM Raw DEM and the global TDM DEM within each scene over calibration regions covered by that scene. Large difference means larger uncertainty of the DEM calibration. The IQR was chosen instead of standard deviation because the distribution is not Gaussian. A tilt of a specific Raw DEM was excluded through the

comparison to the reference global TDM DEM. If still present it would lead to an increase of the spread, since the calibration regions (visible in Figure S1) are relatively well distributed geographically.

Slope dependent elevation offsets are caused by horizontal shifts between the two DEMs, which according to our checks are negligible compared to vertical ones. The glaciers are rather flat, except on the small regions of the mountain ranges sticking out of the plateau. Therefore, the combination of small shifts and small slopes would make these effects negligible. We focus on vertical offsets. Evaluating on the entire off-glacier surface would have been another possibility. But this would have been biased by the higher slopes of the off glacier mountains of Patagonia and would lead to a larger error than what we found on the glaciers which have low and moderate slopes. Since this error is linked to the calibration procedure we compute it on the calibration regions themselves.

**Comment:** P12 II21 Why 1 - 6 m? Reference, calculation or explanation for decision should be provided. Where does that assumption 1000 m.a.s.l come from? Please provide reference.

**Response:** The penetration height offsets of 1 to 6 m were assigned based on explanations given in Sections 3.2.1 and 3.2.2 and in (Abdel Jaber, 2016) based on the observed backscatter coefficient. See response above. We assign the mentioned systematic error only to altitudes above 1000 m (as a conservative threshold) to compensate for undetected regions on the plateau.

Below 1000 m a.s.l. are ice areas where the C-and X-band radar signal does not penetrate. For the SRTM data set (C-band) penetration is no issue because the snow and ice surfaces were wet during the SRTM mission. Regarding the TanDEM-X data, only a quite small percentage of the total data set exhibited partly frozen or dry snow. The penetration height offsets for completely dry snow and firn are based on the relation between  $\sigma^{\circ}$  for dry and wet snow and  $\Delta h$  in Figure 8.12 of (Abdel Jaber, 2016) showing for dry snow a mean offset of 4 m and a maximum offset of 6 m. This is in agreement with the number on X-band one-way signal penetration for dry snow reported by Mätzler (1987) if converted into two-way penetration and also with the TanDEM-X penetration bias in the percolation zone of the Greenland ice sheet (Rizzoli et al., 2017). The penetration depth of refrozen snow crust is smaller (see response to Referee 1, P12, L26). The good agreement between our volume change rates 2000 to 2012 over NPI and the results of Dussaillant et al. (2018) based on optical data confirms the validity of our approach regarding signal penetration.

Mätzler, C.; Applications of the interaction of microwaves with the natural snow cover, Remote Sensing Review, 2, 259-387, 1987

Rizzoli, P., Martone, M., Rott, H., and Moreira, A..: Characterization of snow facies on the Greenland Ice Sheet observed by TanDEM-X interferometric SAR data. Remote Sens., 9(4), 315; doi:10.3390/rs9040315, 2017.

Changes: Relevant explanation was added to Sect. 3.2 and 3.3.

**Comment:** P13 II1 According to this paragraph: for interpolated seasonal correction, the last epsilon term should dominate the quadrature sum and thus the total SECR error ,if I understand correctly. What does 'increase by a factor of three' mean in this context? Times 3 (\*3) ? I compared SECR uncertainty value for

extrapolated glaciers (e.g. Jorge Montt, Bernardo, Tempano) in Tab. 3 with values for not extrapolated glaciers. First ones are not near triple of latter. And they should even be higher than triple, following this paragraph: scaling by year (divided by 0.27. for 99 days for example) is performed as well as a \*1.5 increase for the timespan difference. Please explain where I've gone wrong and/or revise the explanations in this paragraph.

**Response:** Thanks for pointing this out. It seems there is some misunderstanding regarding the seasonal correction and its impact for the retrieval of SECR. The term seasonal correction refers to the difference between mean annual SECR over epochs spanning 12 years (2000 to 2012) and 4 years (2012 to 2016) without accounting for seasonal differences in SEC of the missing days vs. the mean annual SECR taking seasonal differences into account. For the extrapolated glaciers 53 to 103 summer days are missing in order to cover the full 4 year period (1461 days). This means that the missing days to be substituted correspond to 3.6 % to 7.0 % of the 4 year period for which the mean SECR is computed (and not 27 % which would refer to a single year). The impact of missing days to be substituted for the 12 year period is still much smaller. This is now made clear in the revised section 3.1.3 and in Supplement S4.

**Changes:** Sect. 3.1.3 was revised providing a better explanation of the seasonal correction. We updated the results (tables and plots) to correctly take into account the spatial variability of the systematic error.

**Comment:** P13 II9 A formula containing the total SECR error would be helpful for traceability. Is it  $\delta$ SECR = sqrt( $\varepsilon$ b2 + SE2). Just to make sure I got the method correctly and the comment above (II1) is justified.

**Response:** We do not see the need for adding this formula as it refers to basic error propagation, already described by the formula added in response to comment P11 L17.

**Comment:** P13 II15 I would assume a factor of 3 to be very low for the icefields concerning a factor 5 was applied e.g. by Brun et al. 2017 in High Mountain Asia, whereas the variability of SECR patterns in the icefields (especially SPI) is rather high.

**Response:** This comment seems to be based on some misunderstanding, as already explained in the response to P13, L1. The revised Section 3.1.3 and Supplement S4 provide full traceability on this issue. The analysis on the validity of the mass balance gradients (in S4) used for estimating the seasonal difference in the ablation component of the four unsurveyed glaciers in S4, for example, indicates that the increase of error by a factor of 3 for unsurveyed regions is an overestimate. Nevertheless, we keep this (rather conservative) factor as it anyway does not have a large impact on the VCR rates because the extrapolation refers to a limited subset of the total data sets in respect to area and time span.

**Comment:** P13 II19 A formula for the complete error propagation throughout mass balance computation would be appropriate.

**Changes:** We improved the text in Section 3.3, added and highlighted formulae so that it should provide adequate understandability and traceability of the error estimation procedure.

**Comment:**\_P15 II19 The processes described should be perfectly correct. However, I doubt the values found through the seasonal correction analysis are able to significantly support this interpretation. As

mentioned, I assume this daily SECR as a study design feature hard to accept. Also, a precise description of the method that smoothed the SECR field in Fig 8 would be of interest— or even better a display of the original data (SECR field). If it is clearly shown, that the process introduces more precision to the data, than it introduces measurement/ interpolation uncertainty (also regarding comments to 3.3.3) I am willing to accept it. So far, I find it difficult to support it.

**Response:** This comment refers to the seasonal correction which is not the topic addressed on page 15, line 19 to 26 (In Section 4, "Results"). Here we discuss processes of relevance for the observed SECR pattern during the summer period, pointing out the main factors responsible for surface lowering. The reference to a daily ablation rate provides a hint on the magnitude to be expected for the contribution of surface melt during summer, based on measurements. There is no claim for measuring ablation by DEM differencing because these data provide the sum of SEC due to ablation and emergence/submergence (as explained in the paper). In our view the information provided in this section is clear and without any fault. On suggestion of Referee 1 we added a reference on firn densification. During summer firn densification is a general feature in firn areas of temperate glaciers.

Changes: Reference on firn densification on SPI in summer added (Rivera; 2004)

**Comment:**\_P15 II28 For the subaqueous loss Abdel Jaber (2016) is referenced. But for the basal cross-sections an original source should be cited.

**Response:** The references for the bathymetric data on the four mentioned glaciers (Upsala, Jorge Montt, Tyndall and Ameghino) will be added. The calving cross sections of these glaciers are deduced from bathymetric data in front of the glaciers and the freeboard. References to the bathymetric data:

Ameghino Glacier:

*Stuefer, M: Investigations on mass balance and dynamics of Moreno Glacier based on fieldmeasurements and satellite imagery, PhD Thesis, Univ. Innsbruck, Austria, 1999.*

The thesis reports also on two field campaigns on Ameghino Glacier, including pre-frontal bathymetricmeasurements.

Jorge Montt Glacier:

Rivera, A., Koppes, M., Bravo, C. and Aravena, J.C.: Little Ice Age advance and retreat of Glaciar Jorge Montt, Chilean Patagonia. Clim. Past, 8, 403–414, 2012.

Tyndall Glacier:

Raymond, C., Neumann, T., Rignot, E., Echelmeyer, K., Rivera A., and Casassa, G.: Retreat of Tyndall Glacier, Patagonia, over the last half century, J. Glaciol., 51(173), 239-247, 2005.

Upsala Glacier:

Naruse, Renji and Skvarca, Pedro, "Dynamic features of thinning and retreating Glaciar Upsala, a lacustrine calving glacier in southern Patagonia", Arctic, Antarctic, and Alpine Research (2000), 485--491.

**Skvarca, P and De Angelis, H and Naruse, R and Warren, CR and Aniya, M, "Calving rates in fresh water: new data from southern Patagonia", Annals of Glaciology (2002), 379--384.**

**Changes: The original sources of bathymetric data have been cited in the Supplement.**

**Comment:**\_P18 II 9 It is unclear here if that paragraph refers to previous work (Abdel Jaber 2016) or a different publication. Any citation would help. Also I would suggest a reference to the Figures displaying those datasets (provided in the supplement if it is own work)

**Response:** A similar remark was made by Referee #1. The analysis refers to ice flow velocity results which we performed complementary to this study. It is based on TerraSAR-X as explained now in the revised text. Within the ESA project SAMBA (mentioned in the Acknowledgements of the paper) we generated digital maps of ice velocities of SPI and SPI derived from TerraSAR-X and Sentinel-1 data, including time series. In Section 5.1 we report some numbers out of these results, for supporting the discussion on differences between epoch 1 and 2. We added one figure with velocities for four main glaciers in different years. Including a comprehensive report on all these results is not the objective of this paper and would overrun the maximum length of TC papers.

**Changes:** Information has been added in Section 5 to clearly indicate the origin of all cited ice flow velocity results. Furthermore, a new figure displaying novel unpublished plots of TerraSAR-X ice flow velocities along the central flowlines of four main SPI glaciers was included.

**Technical Corrections:**

Comment: p4 l1 'Method and error estimation'

**Response:** Not clear what is wrong here.

**Comment:** P7 I1 Check formula. This way it says  $\delta$  hoff is equal  $\delta$  hoff times the factor.

**Response:** Assuming p7 II 11: "." is not a multiplication but a punctuation mark. We will avoid starting the sentence with  $\delta$  hoff.

Changes: Formula was isolated and sentence not starting with a Greek letter.

**Comment:** Also the distinction, when formulas are a) formatted as objects to be numbered b) written as part of continuous text c) omitted, but have a text description instead is not clearly structured. This should be reconsidered thoroughly.

**Changes:** Improvements were implemented as suggested. Formulae which are relevant to the method and error estimation are now displayed as object and numbered. Generic or less important formulae are left inline. Text description of operation is used to avoid repeating with variations a numbered formula which is already provided (ex. Quadrature sum for error propagation).

**Comment:**\_P14 I 'Figs' Fig. /Figure consistency Check throughout the text, also Table /Tab.

**Response:** We checked the Fig/Figs vs Figure. "Tab." does not occur.

**Changes:** We checked this issue according to the rules of the Journal, one instance of Figure was corrected.

**Heterogeneous spatial and temporal pattern of surface elevation change and mass balance of the Patagonian icefields between 2000 and 2016**

Wael Abdel Jaber1, Helmut Rott2,3, Dana Floricioiu1, Jan Wuite2, and Nuno Miranda4

1Remote Sensing Technology Institute (IMF), German Aerospace Center (DLR), Oberpfaffenhofen, Germany 2ENVEO IT GmbH, Innsbruck, Austria

3Institute of Atmospheric and Cryospheric Sciences, University of Innsbruck, Innsbruck, Austria

4European Space Agency (ESA) - ESRIN, Frascati, Italy

Correspondence to: Wael Abdel Jaber (wael.abdeljaber@dlr.de)

Abstract. The Northern and Southern Patagonian icefields (NPI and SPI) have been subject to accelerated retreat during the last decades with considerable variability in magnitude and timing among individual glaciers. We derive spatially detailed maps of surface elevation change (SEC) of NPI and SPI from bistatic SAR interferometry data of SRTM and TanDEM-X for two epochs, 2000-2012 and 2012-2016 and provide data on changes in surface elevation and ice volume for the individual glaciers and for the icefields at large. We apply advanced TanDEM-X processing techniques allowing to cover 90 % and 95 % of the area of NPI and 97 % and 98 %-of the area of SPI for the two epochs, respectively. Particular attention is paid to precisely coregistering the DEMs, assessing and accounting for possible effects of radar signal penetration through backscatter analysis, and correcting for seasonality biases in case of deviations in repeat DEM coverage from full annual time spans. The results show a different temporal trend between the two icefields and reveal a heterogeneous spatial pattern of SEC and mass balance caused by different sensitivities in respect to direct climatic forcing and ice flow dynamics of individual glaciers. The estimated volume change rates for NPI are  $-4.26\pm0.20$  km3 a-1 for epoch 1 and  $-5.60\pm0.71$  km3 a-1 for epoch 2, while for SPI these are  $-14.87 \pm 0.51 - 14.87 \pm 0.52$  km3 a-1 for epoch 1 and  $-11.86 \pm 1.90 - 11.86 \pm 1.99$  km3 a-1 for epoch 2. This amcorresponds for both icefields to  $\frac{0.047 \pm 0.005 \text{ mm a}^{-1}}{4}$  an eustatic sea level rise of  $0.048 \pm 0.002 \text{ mm a}^{-1}$  for bepotch icefiel1 and during  $0.043 \pm 0.005 \text{ mm a}^{-1}$  the 
[revised manuscript text omitted]

 $SE - \sigma_r$

To derivSe-ansonual variateions of SECR shourface-eld bev tatioken chaingeto accoundt massfor balanede;riving seasonnual variationes of speciarfie mass balance shoueld be-vataken-ionto accouhantge if the time span of the repeat DEMs does not exactly match yearly intervals. TCommonly the meannu daily mean-SECR of the giveneo timplete speriodan is commonly extrapolatused tfor the mfisslling temporal gap:s However; infor summer incbtreased melcting andthe compaentribution of excesnow/firms edauyse. Thisignifie approach int-roducevs a biats ions from the annual mean SECR. In ordcaser tof mseasonitor suchal devariations. The magnitude tof comthe bias dependsate on them poercental gape of miss-ing (our excess) datys and sets whe coamplituted our SPIthe seasonal higcycle. Th-re seasonalu correction, asumm elaborated SECRhere, refers mapto by the differencinge bethrween mean additionnual SECRaw DEMs of December 2011 versus the corresponding chsame-beam Raw DEM of Januar4 y/Mearehs (2012 to 2016), alreadspectively p12 yearts of(2000 theo 2012), DEM-mostakieng (Tseasonable S2). Thdiffe Drencembes in SECR for 2011 missing days intao account versus SPImean annualm SECR withosut accompleuntelying for soutch odif-49ferences.4o

The <del>S,</del>temporal mismand tch versus eentrxactl/y 4 yearst and 12 yearns pvarties for the two epochs and forth different section.s of Tthe timce span fis 99 eldays. fFor 2012 to 2016 the impactw of wseasteonal corrections bis moreams, importand 33t daysue ftor the eashortern timest be spamn. TFor NPI the numbere of missulting dailys SECR, corresponds to 3.6 % to 4.8 % aof ther cons4-yervar tivme maskipang, of aor thefae tswo maind filteracks acquing (mred ian Februandry 2012. A smooall secthiong), of NPI (coverisng sthe lowner termin Fig of S. 8Rafael, S.

I Quin-thisn stuandy thBe 2012–2016 SECR nisto glaffectied byrs) was significanst temporald gap dcqurired ong summer28 May 2015/2016, whi(NPI schene Non. 1; Table shortS1, tFimeg. spS1an). Fof 4 years makthes main corrections nof SPI thee pessary. Thce-numbtager of missing days (5ranges from 3.6 % to 75.0 on NPI%, 53-excepto 103 fonr SPI)a vsmall subaries-aeeo wherde ingt is 7 %. tFor SPI the e2000 to 2012 mbinsmatch ion percentage of TDM datatakhes (Tabfull period S1 randges S2from and Fig0.1 S1)% to 1.0 In% of the 12-yeard period. Nevertheless, we applied seasonal comprections also to this data set. fFor the mistwo tracks covering summther dmays wein rselctiedons onf NPI the filtegap corresponds daito only SECR0.1% of summther 2011/2012. This wyears, usgived pixthe lwimited surface wheovered availably the onthird SPItrack, ino pacorrecticulon was applied to theis 33-daytaset.

Here bwe explam coinei details on the seasonal corregction whefore 53 the epoch 592012–2016 dbecayuse musof their blarger coimpensact. Depend-ing Forn the mavaisslabing parlity of Padditional XITDM GIDEM datacier, the hyfollowing psromcetdurices mwean (aggreg atpplied for different 100 m clsevaction bins) of low-loss the glaicefierIds:

- Three (Padditional XI,TDM Peracquisitions Morf Deno,cember Grey,2011 (Tyndabll, HPSe 13,S2) Eurcopa, Pvenguin,r Guilardi)the was-ousthed (green, curve-in-Fig. S3).tral Oand north-western resections of SPI (except59.4 % Jorgef MonttSPI, GlaeFierg. 8.). tThe hypsome data werice museand tof computhe 99-dailys summer SECR wasover susmmedr (2011/2012 bluey eurveDEM differen-Fcing vs. S3).TDM Ondata NPIof (exMarcepth 2012 (99 days) coverine-1)g the hypsometraien gpareet of SPI and evs. TDM data of 31 January 2012 (33 days) coverin Fig. S3 was sub-sedction sine the south-erast. The eSECR malvingps of summes are a2011/2012, smcalled etompon the length of totalhe maiss-balaineg period, wherea used fon SPIr substroituting dynathe miessing downwayst ing lsummer 2015/2016, a valids tappro-achig as ther nmegan tive SECR on glacimper-atongures.

In of the two seummenarios addgressed above-we used ithein 2011/20.12 sum°C. Air temperatures SECRof tohe sEubsropean Centre for Meditutm-Range Weather miForecasts Interingm dRe-Analysis of (ERA-Interim) (Dee et al., 2011; Berrisford et al., 2011) summhow at ther 20185/2016. At thPa level synfoptier stathe grid poin-Balmacedat (457.925° S;, 713.6855° W) near (NPI) thea moeanthly summean air temperatures of summer5.9 °C in 2011/2012 comparend to6.0 summer°C in 2015/2016, wefore hpoigher in December and Januaryt (+250.25° °CS, +1.5-73.55°C), 1W (sowuther in FSPI) thebr values arye (-1.53.6°C) in 2011/2012 and similar3.5 °C in Ma2015/2016.

Foreh. ASPI simiglaciers not coverend wasby mthea summered at2011/2012 PuntaSEC Arenmasp (53.00° S;except 70.85°Pio W),XI leanding usJorge Montot glacierssum) we ruseld datively sSECR imilarn mdependean ice abof elevation deraived from the 99-day SECR maps in bothf summers 2011/2012 (blue curve in Fig. S3).

On For the ablasection-area of JPiorge MonttXI Glacier (that is not covered by the Dsummeer SEC map we use the hypsombetric 2011SECR curve of low-loss glaciers (green curves) itn Fisg. importaS3). Ont toNPI, except onsider the termini croveasred dby scenamice dNo. 1, wnwe alstingo comparused to the avhypsometraic gre of then icurve of Fieldg. ToS3 thsis endce wthe sepmajoraitely of glaecouiers are noted calving for the surfcace lowerving fluxes are a smalated tcompo-neinther surof totacel mass balance or ice flow dynamics(Schaefer et al., 2013).-

- For the lathree NPI termini wcove-assumred aby scene Nonst. 1 antd vefortical velJoeity throughe Mount-the yGlaciear, forwhich is nothe fcovermerd by the summeasonalr cycle2011/2012 SECR wmaps takend subject to significanto dynaemic douwnwast-ing Wwe addsed-paratoe the dynamicSEC component in the seablastion areas related eto surface melt from SEC duee tio dynamic fdor-wnwablastiong. TFor this copurrectipon-wase estimateds by usiong the elevation adependenualce of the lspecifinc surface marss balance (SMB) during summer adiend the ofull -1.37year ma-1 are pner 100eded. mTo ofur eknowlevationdge, up to now the eqonly muiltib-year tiume lserines (of ablased-tion field measurements on Perito Moreanoy Gglacier (Stuefer et al., 2007))of sSPI and NPI, incaleuding withe thseparation of summer-to-and annual peratiods, ohas been performed ablation Moreno Glatcier (Stuefer et al., 2007).

W The applied ca-simbilarity approach to be selforene 1-of NPI mass balancover elevationg mgradienlyt thas been tchercked by mieanis of S. Rafamodel, S. Qoutpuintin aond SMB for NPI wenist coast glaciers (Fig.(Schaefer et al., 2013) S1).and Beingmass balancquire data onf 28Chico MGlaycier 2011(Rivera, 2004), accounting for the wexest/eedast by 189difference aind 200temperature dlaypse rathe 4-yeacross the imcefield span(Bravo et al., 2019). Furto-ther ovderltapping sls avre seengives 4-and 5-(3-aind 14-Dthee Supplember 2015)nt, respSectively. S4. The eratio between daily SMB-relatribed SEC dutriong dsummer vs. to be rest bof the dyneamr ien imbaladependence and of melt-wevas-obtaion is used byfor esetimalting the uincorrectased 2011–2015-SECR contribution thdue acto surfal-excess pmelt during summer od.n Tthe dJorgevia Montt termionus and from the mreduceand annuamelt acontriblaution during May teo was-Decoemputbed for withe thre-balance gNPI teradmient of i. MFor theno Gldynaemier (Stuefer et al., 2007);c downwhastiehng accoumponents forwe reduesed ablationhe duaverinage SECR of the f-summl period.

For months.

[revised manuscript text omitted]
 densificompaction and melting of the top snow layers (va(Rivera, 2004). On the firn plateau, at es-levartiounds  $-0.03 \ge 1200 \text{ md}^{-1}\text{m}$ , arthe obsavervaged oSECR in thsummer pl2011/2012 wates abou)t wh-0.03 md-1 (blue curveas in Fig. S3). In the ablation areas ice melt and dynamic downwasting (varying from glacier to glacier) are the main factors. High loss rates (SECR  $\ge -0.08 \le -0.08 \text{ md}^{-1}$ ) refer to areas that are subject to significant dynamic thinning, such as the lower terminus of Upsala and Viedma glaciers. Average summer melt rates for ice on the lower terminus of Perito Moreno Glacier (at 300 m altitude) are about 0.05 md-1 (Stuefer et al., 2007). On a glacier in balanced state surface lowering due to melt is in summer partly compensated by uplift due to emergence.

[revised manuscript text omitted]

On lower sections of the main calving glaciers temporal variations of flow velocities are a main factor for the differences in SECR during the two epochs. FIIn owrder vto support theloe interpretation of differences in SECR beartween the two epochs, we derived maps of surface velvocity gridded at 50 m for main glaciers from TerraSAR-X 11-day repeat pass data on various dates ofbetween 2010 Sand R2016, applying the offaselt Gltracking techniquer. Thave runcertaeinty of thed velocity magnitudes inof exethesse prof 7 kma-1ducts ins 20.075 (Willis et al., 2012a; Abdel Jaber et al., 2014),md-1(Wuite et al., 2015). buPlots of have-locities alowng cedntral dflowlines, extracted from 4.4the kma-1velocity mafterwps, ardse shown ing IFig. 11 for Jorge Montt, Pio XI, Upsala and Viedma glaciers.

Flow velocities near the calving front of San Rafael bGlacier reached magnitwudes in excess of 18 md-1 in April 2007 (Willis et al., 2012a), dropping to 126 md-1 in Mandy 20162 (Abdel Jaber et al., 2014). Whouginot and Rignot (2015) report for the velocitiesy at 10 km from the ice front show-a temporal peak in 2005 and a decrease by about 20 % until 2014 (Mouginot and Rignot, 2015). The drop in velocisty between epoch 1 and 2 is reflected in the hypsometric curve of SECR, showing reduced loss rates below 800 m elevation during epoch 2 (Figs. 3 and 9). San Quintin Glacier, the largest glacier of NPI, reaches its maximum speed of about 13 kma-1 at a distance ofm d-1 2730 km from the front (Abdel Jaber et al., 2014; Mouginot and Rignot, 2015). BOur analysis of TerraSAR-X data shows between May 200512 and June 20146 on the flow vtelocrmity 1 km-nupstream of a thme froant increased byin abveloucity 5by 10 %. However, this caused only a minor additional increase of surface lowering on the glacier terminus (Fig. 9) because for this glacier the ice export due to calving accounts only for a very small part of total mass turnover (Schaefer et al., 2013).

On SPI calving fluxes play a larger role for mass turnover than on NPI. This is reflected in the change of the average hypsometric curve of SECR of the icefield between the two epochs (Fig. 10). In spite of slightly hmighelar air temperatures during epoch 2 the average rate of surface lowering decreased at elevations below 400 m. Between 400 m and 1000 m elevation the differences between the two epochs are very small. On the ice plateau, between 1000 m and 2000 m, the loss rate decreased slightly, mainly brought about by minor changes on the southern sector of the icefield. Local increase in snow accumulation may play a role.

For six glacier basins the VCR between the two epochs changed by more than +0.2 km3 a-1, summing up to a combined decrease of volume losses by 2.20 km3 a-1 (Table 3). The change of VCR from epoch 1 to epoch 2 amounted for Pio XI Glacier to +0.74 km3 a-1, for Grey & Dickson to +0.37 km3 a-1, for Upsala & Cono to +0.33 km3 a-1, for Tyndall to +0.30 km3 a-1, for Europa to +0.24 km3 a-1, for Penguin to +0.22 km3 a-1. There are three glaciers with major increase of losses during epoch 2 (VCR becoming more negative by  $\ge 0.2 \text{ km}^3 \text{ a}^{-1}$ ): the change of VCR for Jorge Montt is  $-0.36 \text{ km}^3 \text{ a}^{-1}$ , for Viedma  $-0.28 \text{ km}^3 \text{ a}^{-1}$ , for Bernardo  $-0.20 \text{ km}^3 \text{ a}^{-1}$ .

The behaviour of Pio XI Glacier, with frontal advance and positive mass balance since many years is opposed to the general trend of SPI glaciers. The recent frontal advance trend started at the northern section of the terminus in 2006 and at the

southern section in 2000 (Wilson et al., 2016). Between 2000 and 2014 a general-slowdown of velocity was observed on the eentral and southern sections of the terminus (Mouginot and Rignot, 2015; Wilson et al., 2016). The slowdown went on until 2016, whereas the velocity of the northern section more thaecen doublerad between 2013 and 2016 (Fig. 11). **T**Bathymetric data show ishallow rwateflr with ridges running across the fjord inat the mapresent position of thel-iceva front Dowdeswell and Vásquez (2013). Thions impedes chalving at the sof-uthern twice front, causing during epoch 1 a main increas, e of showurface elevationg on the southeronge sect-ionere, laster of SECRn shifting towards the northern section that calves into Lago Greve (Fig. 6).

On Upsala Glacier the front retreated by 4 km between 2000 and 2014. and tThe calving velocity reached a maximum in 2009/2010 (Abdel Jaber et al., 2012; Mouginot and Rignot, 2015) and decreased significantly afterwards, dreachopping a maximufrom 8 m d-1 in March 20011 to 5.9/ m d-1 in August 2014 and 4.8 m d-1 in August 2016 (Abdel Jaber et al., 2012; Mouginot and R 11). This caused a major decrease in the thinning rate of the lower terminus during epoch 2 (Fig. 4).

[revised manuscript text omitted]

Our result for NPI during epoch 1 (VCR =  $-4.26 \pm 0.20 \text{ km}^3 \text{a}^{-1}$ )-is in complines with the numbers resulportsed ofby Abdel Jaber (2016) for the period 2000 to 2014 ( $-4.40 \pm 0.13 \text{ km}^3 \text{a}^{-1}$ ) and Willis et al. (2012a) for 2000 to 2011 ( $-4.06 \pm 0.12 \text{ km}^3 \text{a}^{-1}$ ), the latter based on SRTM and ASTER DEMs. Willis et al. (2012b) recomputed their previous estimate applying a 2 m offset to the SRTM DEM to account for signal penetration which resultings in larger losses (VCR =  $-4.9 \pm 0.3 \text{ km}^3 \text{a}^{-1}$ ). This correction is not comprehensible given the wet status of the snow surface during the summer acquisition of SRTM as evident from the backscatter data (Sect. 3.2.2, and Abdel Jaber (2016)). Braun et al. (2019) report for NPI a VCR  $-4.65 \pm 0.17 \text{ km}^3 \text{a}^{-1}$  over the period 2000 (SRTM data) to 2011/2015 (TDM data).

Our VCR for epoch 1 is slightly lower than the results of Dussaillant et al. (2018) who applied btwo methods: differencing of SPOT and SRTM DEMs (VCR =  $-4.55 \pm 0.41$  km3 a-1) and derivation of temporal elevation trends from ASTER DEM time series (VCR =  $-4.72 \pm 0.34$  km3 a-1). Our hypsometric curve of SEC shows up to 2800 m of elevation a similar behaviour as their ASTER\_trend results, although with slightly lower losses at most elevations. Above 1000 m Dussaillant et al. (2018) report 35 % and 22 % of unsurveyed area for the SPOT-SRTM analysis and ASTER\_trend respectively, mostly due to the lack of contrast or the presence of clouds in the optical stereo images. For the same elevation band the unsurveyed area in our 2000–2012 SECR map of NPI is 6 % on NPI. On glaciers larger than 100 km2 the SEC rates with both methods applied by Dussaillant et al. (2018) agree within errour baresults within errour baresults. On two medium-sized glaciers, Exploradores (86 km2) and Grosse (67 km2), the average SECR of their two methods differs by more than 1.0 ma-1, their ASTER\_trend being ~ 0.8 ma-1 higher than our SECR and ~ 0.6 ma-1 higher than those of Willis et al. (2012a).

On SPI Willis et al. (2012b) estimate a VCR of  $-21.2\pm0.5$  km3 a-1 for the period 2000–2011, a much larger value compared to the onvalue reported here for epoch 1 (VCR =  $-14.87\pm0.51-14.87\pm0.52$  km3 a-1) and to that of Abdel Jaber (2016) for

[revised manuscript text omitted]

Acknowledgements. The work was supported by the European Space Agency, ESA Contract No. 4000115896/15/I-LG, High Resolution SAR Algorithms for Mass Balance and Dynamics of Calving Glaciers (SAMBA). The TanDEM-X data were made available by DLR through the projects DEM\_GLAC0787, XTI\_GLAC0495, XTI\_GLAC6663. We would like to thank Lukas Langhamer (Humboldt-Universitaet zu Berlin) for providing processed climate re-analysis data over the icefields.

<The following bibliography contains references included in both the old and the new version on the manuscript.>

[revised manuscript text omitted]

| Icefield | Period    | Area [km 2 ] | Cov.
[%] | Average SECR $\begin{bmatrix} m a^{-1} \end{bmatrix}$ | Volume change $\left[ \mathrm{km}^3  \mathrm{a}^{-1} \right]$ | Mass change $\left[\operatorname{Gta}^{-1}\right]$ | Sea level rise $\left[\mu m a^{-1}\right]$ |
|----------|-----------|-------------------------|-------------|-------------------------------------------------------|---------------------------------------------------------------|----------------------------------------------------|--------------------------------------------|
| NPI      | 2000–2012 | 3975.3                  | 95.4        | $-1.072 \pm 0.049$                                    | $-4.261 \pm 0.196$                                            | $-3.835 \pm 0.236$                                 | $10.594 \pm 0.653$                         |
| NPI      | 2012–2016 | 3914.2                  | 89.8        | $-1.431 \pm 0.188$                                    | $-5.602 \pm 0.735$                                            | $-5.042 \pm 0.693$                                 | $13.927 \pm 1.915$                         |
| SPI      | 2000–2012 | 12999.0                 | 98.0        | $-1.143 \pm 0.040$                                    | $-14.874 \pm 0.518$                                           | $-13.386 \pm 0.712$                                | $36.979 \pm 1.966$                         |
| SPI      | 2012–2016 | 12846.8                 | 97.0        | $-0.923 \pm 0.155$                                    | $-11.860 \pm 1.987$                                           | $-10.674 \pm 1.839$                                | $29.485 \pm 5.079$                         |

[revised manuscript text omitted]

---

## Referee Report (RR1)

**2nd Referee report on "Heterogeneous spatial and temporal pattern of surface elevation change and mass balance of the Patagonian icefields between 2000 and 2016"**

*Authors*: Abdel Jaber, W., Rott, H., Floricioiu, D., Wuite, J., Miranda, N.

The Cryosphere Discuss., https://doi.org/10.5194/tc-2018-258

Referee comments are shown in **black**, cited aurhor's responses in **blue**

General Response:

Thank you for providing these informative comments and the clearly improved manuscript. The general structure has improved through clarifications of sources of work / data and of the application of certain methodological steps. Also, the rearrangement of the methods section gives the opportunity to quickly trace the workflow and the error estimation. This is a great feature for a scientific publication. Yet, not all discussed issues have been resolved.

Generally, previous review comments were perceived more technically oriented than intended. The idea was to get more transparency to methods, where possible. ITP processing will not be re-trackable for most readers – o.k. But when it comes to SECR / VCR /geodetic mass balance, the community will be able and willing to re-track how the results have been produced and evaluated. Therefore, the reviewer's intention was to point out passages, where methods can be described in a more accessible way to scientific readers from related disciplines. There were also misconceptions from my side about specific points, that have been clarified and solved by now. Thank you for correcting the mistaken statements and revising the corresponding passages in the text. A few comments on methods remain for this second round.

Thus, I want to explain: I do accept the scope of the paper and do not ask to inflate the manuscript with a lot of method details. I would rather ask to substitute some phrasings with more precise ones. Even the extensive error estimation terms do not necessarily need to be in the manuscript as it now resulted from the first review. Rather they could be listed in the supplement – to enable traceability without emphasizing methodological parts of the study too much. Please consider this option for each of the subsequent detailed comments as well.

Another major concern of this and the first review was to make the paper more easily comprehensible without having to read the thesis Abdel Jaber (2016) in parallel. With respect to this, comments reappear.

Apart from the systematic error estimation, the rest is to be considered minor revisions. The paper has improved to an almost publishable version. But the reappearing concern about an appropriate coregistration validation should be seriously addressed.

General comment 3.1.2 Coregistration of DEMs and following error contribution in 3.3.3

I summarize: Coregistration is done by manual APO updates through ITP processing with regard to ref. DEM (TDX 0.4 arcsec). Further efforts are claimed to not be necessary for TanDEM products. NASADEM was vertically adjusted, horizontal adjustment was checked to be needless. I suggest support the precision of this process, which is of key importance to geodetic method, with statistical values more adequate than the provided ones.

Therefore, I want to emphasize again: If you assess the error on the same CRs, where you corrected the offset, you get the residual error of the values you calibrated your model with. Not the error of the values you claim it to be valid for which is all the rest of each DEM. It for sure is possible to find some VRs (validation regions) with similar properties as the CRs. Show the spread of height diff to reference DEM there. Including slopes up to 20° in this validation process would solve two problems: 1) the dependence of the elevation difference to slope could be addressed. 2) the claim that horizontally no shifts remain, could be padded, since they appear more significantly with slope. Values would point to a positive /negative distribution center corresponding to the spread of aspects in the corresponding scene if residual hor. shifts remain.

Specific comments: 3.1.2 Coregistration of DEMs and following error contribution in 3.3.3

revise the systematic error contribution of $\varepsilon_{reg}$:

Do not use error values on CRs. Statement above.

Do not apply IQR: The process does not cut outliers but reduces the systematic error component to its lower half. You responded: 'The IQR was chosen instead of standard deviation because the distribution is not Gaussian' → If DEMs are hor. and vert. coregistered, the residual height error should be normally distributed.

General comment: 3.2 Impact of radar penetration

The section should concentrate on /emphasize a conclusion on the extensively described distribution of σ0. In the end, a mask with σ0 thresholds as a function of elevation, look angle incidence angle, date/time etc. would be precious. It would enable applicability of this new method for future studies. At the moment the procedure leaves the reader with the barrier 'done by expert knowledge' and no chance to use or reproduce the method. If changing that is out of scope, at least the numeric outcome of the method should be supported by more than one resulting number which up to now is limited to the final $\varepsilon_{pen}$ in the supplement.
So far, the end of 3.2.1 reads (p11ll7): 'The offsets are based on empirical observations of the relationship between σ0 and height offset performed on multiseasonal TDM Raw DEMs of NPI, showing a mean penetration bias of 4 m for an increase of σ0 by 10 dB from wet to dry snow (Abdel Jaber, 2016).'

Specific comment: 3.2 Impact of radar penetration
    Please, add more precision to the cited paragraph: what exactly was applied then? A 4 m mean penetration depth assumed for all masked areas? Or if not, how is it calculated/modelled? Moreover, the description in p13ll15 stays vague, except for 6 m as a maximum.
    Provide a conclusion about applied thresholds (σ0 etc.) in the text at the end of 3.2.2 similar to the cited one.
    Provide numeric values for the assumed (mean?) penetration depth at each location in Fig. S5 referring to the manually outlined areas they are applied to.

Re: Specific Comment: P 7 ll30

What kind of filtering was applied? It would be interesting to see the original dataset and a Δh map outside the icefields.

Response: Since the Summer 2011/2012 daily SECR is used only for the seasonal correction, we applied the following procedure for eliminating outliers: (i) conservative masking on glaciated terrain of regions with high backscattering and peaks in the daily SECR values followed by (ii) 2-step filtering with sliding 15window: (a) median filters with kernel size 9 and (b) smoothing with kernel size 9. The raster posting is 0.4 arcsec. This way the localized seasonal changes or outliers were eliminated and thus the SECR map can be used for the purpose of compensating the temporal gap in 2015/2016.

→ A shortened description would complement the caption of Fig8. Please consider reproducibility, not only answering to me.

Re: Specific Comment: P7 ll13 (update p7 ll3)

How is the absence of horizontal shifts checked? The detection is slope dependent (cf.Nuth and Kääb (2011)), thus cannot be efficiently performed on an area without slope as the CRs (avr. slope below 4)

Response: The horizontal shifts (in our case possibly acting in the ground range direction) were not checked analytically directly on our datasets, but relying on visual analysis of all available off-glacier terrain. Analytical checks using the method of Nuth and Kääb (2011) was done for the TDM-SRTM SEC datasets during the preparation of the thesis (Abdel Jaber et al., 2016) corroborating the validity of this calibration procedure. Because the same method was applied for this paper as for the thesis, the conclusions regarding this procedure can be adopted for this work.

→ Concerning the last two sentences, the thesis reads:

'The horizontal coregistration with respect to the SRTM appears to be achieved as well: the difference images do not display a hillshade effect (see Figure 7.9) and the method of Nuth & Kääb (2011) confirmed the validity of the correction. The quality of the coregistration was furthermore confirmed by comparing on stable terrain (ice-free) the corrected TanDEM-X Raw DEMs and the SRTM C-band DEM to several ICESat GLAS altimetry tracks acquired between 2003 and 2009. Nevertheless a certain amount of residual horizontal misregistration can be expected, given the uncertainty linked to the height offset estimation. Its contribution will be accounted in the mass balance error budget'(Abdel Jaber 2016, p 91)

There is no statistical / numerical evidence referenced, but a Figure to check the absence of aspect dependent slope offsets visually. It is due to such a response, that I must insist in you taking the suggestion seriously that I repeat here: Put more effort to divide the thesis and the paper, to make latter a scientific stand-alone document. For this specific case: Even if the method is applicable, please show this, by validating your data independently as suggested in >Comments on 3.1.2 Coregistration of DEMs and following error contribution in 3.3.3<.

Re: Comment: P9 ll8

Can you please add more information to increase reproducibility when data gets available: what threshold on SEC values? What morphological operators?

Response: We did not include these details because we do not think that this is an interesting point and would inflate an already very long paper. For each of the 4 SECR maps we produced a raster starting from the flag mask (FLM) layer that resulted from the processing with ITP which provides roughly the regions affected by layover and shadow. Thresholds h/ t < -10 m/a and > +6 m/a were applied. A morphological operator of closing followed by a 5 x 5 median filter was applied on the mask raster in order to "clean" the mask, avoiding noise due to thresholding.

→ Eliminating the phrase from the manuscript does not support transparency.

Please integrate answer in manuscript /supplement. Especially the asymmetrical thresholding is interesting. Please provide reasoning for that. A symmetrical cut-off for outlier elimination would be methodically sounder. Executed like provided, statistically appearing (therefore normally distributed) residual noise error resulting in higher/lower rates gets dragged to more negative rates.

Re: comment: P13 ll1

According to this paragraph: for interpolated seasonal correction, the last epsilon term should dominate the quadrature sum and thus the total SECR error, if I understand correctly. What does 'increase by a factor of three' mean in this context? Times 3 (*3) ? I compared SECR uncertainty value for extrapolated glaciers (e.g. Jorge Montt, Bernardo, Tempano) in Tab. 3 with values for not extrapolated glaciers. First ones are not near triple of latter. And they should even be higher than triple, following this paragraph: scaling by year (divided by 0.27. for 99 days for example) is performed as well as a *1.5 increase for the timespan difference. Please explain where I've gone wrong and/or revise the explanations in this paragraph.

Response: Thanks for pointing this out. It seems there is some misunderstanding regarding the seasonal correction and its impact for the retrieval of SECR. The term seasonal correction refers to the difference between mean annual SECR over epochs spanning 12 years (2000 to 2012) and 4 years (2012 to 2016) without accounting for seasonal differences in SEC of the missing days vs. the mean annual SECR taking seasonal differences into account. For the extrapolated glaciers 53 to 103 summer days are missing in order to cover the full 4 year period (1461 days). This means that the missing days to be substituted correspond to 3.6 % to 7.0 % of the 4 year period for which the mean SECR is computed (and not 27 % which would refer to a single year). The impact of missing days to be substituted for the 12 year period I still much smaller. This is now made clear in the revised section 3.1.3 and in Supplement S4.

→ Thank you for correcting my wrong assumption. It really helped understanding the seasonal correction – as well as the revision of the respective section. Yet, it also emphasized the relatively small impact of the correction to the results. After reviewing the paragraph of the related error estimation again, I now understood correctly what has been done and am sorry about misconceptions. Still I suggest a little step to improve the manuscript.

The paragraph p13ll26 reads:

To compute the systematic error linked to the seasonal correction (Sect. 3.1.3), the previous three systematic error components ($\epsilon$ reg , $\epsilon$ pen and $\epsilon$ add ) were estimated separately for the summer 2011/2012 SECR. Here $\epsilon$ add was increased by a factor of 1.5 to account for the different temporal coverage. All three components were summed in quadrature and conservatively further increased by

a factor of 3.0 on extrapolated regions (north of SPI and NPI). A pixelwise scaling by the number of corrected days and by the appropriate Δt in years was applied, leading to a fourth systematic error

The calculation in the cited paragraph is:

(1)  $\varepsilon_{seas}(x, y) = sqrt(\varepsilon_{reg}^2 + \varepsilon_{pen}^2 + (1.5*\varepsilon_{add})^2)*days)/\Delta t$

(2)  $\varepsilon_{seas}(x, y) = sqrt(\varepsilon_{reg}^2 + \varepsilon_{pen}^2 + (1.5*\varepsilon_{add})^2)*3*days)/\Delta t$ (for extrapolated regions)

I assume. since you involved this error contribution from a seasonal yearly change rate scaled to the dh error, in this case the Δt=365days. This practice is fine, and I misunderstood it in the first place. That might be, due to the fact, that the phrasing in the last cited sentence indicates, that here a division by entire epoch would appear (Δt=4*356 d or 12*365 d respectively). Latter would be questionable, not representing what has been done, since you subsequently quadrature sum the epsilon components as

(3)  $\varepsilon = sqrt(\varepsilon_{reg}^2 + \varepsilon_{pen}^2 + \varepsilon_{add}^2 + \varepsilon_{seas}^2)/\Delta t$

where the division by entire epoch appears as Δt to give an appropriate error budget for the SECR.

→ Please correct 'and by the appropriate Δt in years' to a phrasing that clarifies Δt to be 365d in this case.

Formal comments:

P5 l16: 'attitude adjustment'

---

## Author Response (AR2)

**TOPIC "Impact of radar penetration" (Sect 3.2)**

**Comment of the referee:**

The section should concentrate on /emphasize a conclusion on the extensively described distribution of σ0. In the end, a mask with σ0 thresholds as a function of elevation, look angle incidence angle, date/time etc. would be precious. It would enable applicability of this new method for future studies. At the moment the procedure leaves the reader with the barrier 'done by expert knowledge' and no chance to use or reproduce the method. If changing that is out of scope, at least the numeric outcome of the method should be supported by more than one resulting number which up to now is limited to the final $\varepsilon_{pen}$ in the supplement.

So far, the end of 3.2.1 reads (p11ll7): 'The offsets are based on empirical observations of the relationship between σ0 and height offset performed on multiseasonal TDM Raw DEMs of NPI, showing a mean penetration bias of 4 m for an increase of σ0 by 10 dB from wet to dry snow (Abdel Jaber, 2016).

Please, add more precision to the cited paragraph: what exactly was applied then? A 4 m mean penetration depth assumed for all masked areas? Or if not, how is it calculated/modelled? Moreover, the description in p13ll15 stays vague, except for 6 m as a maximum.
Provide a conclusion about applied thresholds (σ0 etc.) in the text at the end of 3.2.2 similar to the cited one.
Provide numeric values for the assumed (mean?) penetration depth at each location in Fig. S5 referring to the manually outlined areas they are applied to.

**Response:**

We agree on this point raised by the referee. In order to make the publication increasingly independent of other publications we rephrased the last sentence and added a section on estimation of penetration offset on the critical regions to the Supplement (included in Sect. S5). We explain the relationship between σ0 and the penetration bias which was used to derive the penetration offsets for subareas covered by non-melting snow surfaces according to the increased backscatter coefficient. The percentage of NPI and SPI accumulation area included in the critical regions and the related elevation biases are specified in Table S6 for the SRTM and TDM DEM mosaics.

**TOPIC "DEM Coregistration" (sect. 3.1.2) and following error contribution (Sect. 3.3.3)**

**Comment of the referee:**

I summarize: Coregistration is done by manual APO updates through ITP processing with regard to ref. DEM (TDX 0.4 arcsec). Further efforts are claimed to not be necessary for TanDEM products. NASADEM was vertically adjusted, horizontal adjustment was checked to be needless. I suggest support the precision of this process, which is of key importance to geodetic method, with statistical values more adequate than the provided ones.

Therefore, I want to emphasize again: If you assess the error on the same CRs, where you corrected the offset, you get the residual error of the values you calibrated your model with. Not the error of the values you claim it to be valid for which is all the rest of each DEM. It for sure is possible to find some VRs (validation regions) with similar properties as the CRs. Show the spread of height diff to reference DEM there. Including slopes up to 20° in this validation process would solve two problems: 1) the dependence of the elevation difference to slope could be addressed. 2) the claim that horizontally no shifts remain, could be padded, since they appear more significantly with slope. Values would point to a positive /negative distribution center corresponding to the spread of aspects in the corresponding scene if residual hor. shifts remain.

revise the systematic error contribution of $\varepsilon_{reg}$:
Do not use error values on CRs. Statement above.
Do not apply IQR: The process does not cut outliers but reduces the systematic error component to its lower half. You responded: 'The IQR was chosen instead of standard deviation because the distribution is not Gaussian' → If DEMs are hor. and vert. coregistered, the residual height error should be normally distributed.

*How is the absence of horizontal shifts checked? The detection is slope dependent (cf.Nuth and Kääb (2011)), thus cannot be efficiently performed on an area without slope as the CRs (avr. slope below 4)*
*Response: The horizontal shifts (in our case possibly acting in the ground range direction) were not checked analytically directly on our datasets, but relying on visual analysis of all available off-glacier terrain. Analytical checks using the method of Nuth and Kääb (2011) was done for the TDM-SRTM SEC datasets during the preparation of the thesis (Abdel Jaber et al., 2016) corroborating the validity of this calibration procedure. Because the same method was applied for this paper as for the thesis, the conclusions regarding this procedure can be adopted for this work.*

→ Concerning the last two sentences, the thesis reads:
'The horizontal coregistration with respect to the SRTM appears to be achieved as well: the difference images do not display a hillshade effect (see Figure 7.9) and the method of Nuth & Kääb (2011) confirmed the validity of the correction. The quality of

the coregistration was furthermore confirmed by comparing on stable terrain (ice-free) the corrected TanDEM-X Raw DEMs and the SRTM C-band DEM to several ICESat GLAS altimetry tracks acquired between 2003 and 2009. Nevertheless a certain amount of residual horizontal misregistration can be expected, given the uncertainty linked to the height offset estimation. Its contribution will be accounted in the mass balance error budget'(Abdel Jaber 2016, p 91)
There is no statistical / numerical evidence referenced, but a Figure to check the absence of aspect dependent slope offsets visually. It is due to such a response, that I must insist in you taking the suggestion seriously that I repeat here: Put more effort to divide the thesis and the paper, to make latter a scientific stand-alone document. For this specific case: Even if the method is applicable, please show this, by validating your data independently as suggested in >Comments on 3.1.2 Coregistration of DEMs and following error contribution in 3.3.3<.

**Response:**

The error estimation intends to model the error associated, in this case, with the coregistration procedure. It is not intended as a validation. The number of calibration regions is limited for each Raw DEM (less than 20, less than 10 for some Raw DEMs). The IQR is calculated on their $\mu_r$ (mean of Δh with respect to the TDM global DEM on the CR pixels) as a measure of spread, the rationale being that the larger the spread within a Raw DEM, the less certain is the obtained height offset and the corresponding APO value used for coregistration of that Raw DEM. The distribution of these limited number of $\mu_r$ is not Gaussian and the IQR was chosen over the standard deviation as it leads to a more conservative error estimate for almost all Raw DEMs.

Concerning the issue of horizontal shifts we would like to emphasize that the slope distribution on the icefields is significantly skewed towards low slopes. Figure 1 shows the slope distribution and cumulative slope distribution of NPI and SPI. It can be appreciated that 80% of the icefield surface has a slope lower than 15° on NPI and lower than 23° on SPI.

The small residual horizontal shifts in ground range direction which might affect each Raw DEM have consequently a negligible vertical offset effect on most of the icefield surface.

Nevertheless in order to fully tackle the concerns raised by the referee, we include for each SEC dataset the maps of the elevation difference on stable terrain (except some small glaciers not belonging to SPI and NPI which were not masked out). These are show in Fig. 2 for the datasets of NPI 2000-2012 (a), NPI 2012-2015 (b) and in Fig. 3 for the datasets of SPI 2000-2012 (a), SPI 2012-2015 (b). The images further highlight in magenta the location and coverage on the calibration regions (CRs). These images were requested by both referees, we cannot at the moment include them in the Supplement since this has already reached the maximum allowed size (50 MB). We will gladly include them if this size limitation can be technically overcome. The TDM vs. TDM differences display very good alignment according to Fig. 2(b) and Fig. 3(b). In the NASADEM vs. TDM some vertical biases (of varying amplitude and sign) as well as some horizontal shift of varying direction are visible in different regions of the off-glacier terrain. These are mostly due to long-wavelength biases still affecting the NASADEM caused by the mosaicking

procedure and by oscillations of the mast on the Space shuttle, nevertheless we noticed a significant improvement compared to the previous SRTM v3 (see Manuscript Sect. 2.2 and Supplement Section S2). Furthermore on the steep terrain surrounding the icefields the elevation difference caused by the different resolution of the TanDEM-X and of the NASADEM DEM is noticeable. This bias is however are not critical on the icefield, characterized mostly by low slope and curvature.

We computed some statistical values, as requested by the referee. In particular we extracted mean and standard deviation of the elevation difference on off-glacier samples with slope below 20° for each SEC dataset.

*Table 1: off-glacier, off-water mean and standard deviation for SEC samples with slope below 20° for all SEC datasets.*

| Dataset | **NPI 2000 – 2012** | **NPI 2012 – 2015** | **SPI 2000 – 2012** | **SPI 2012 – 2015** |
|---|---|---|---|---|
| Samples | 13 969 156 | 11 486 375 | 37 455 616 | 22 332 512 |
| Mean Δh [m] | 0.060 | 0.210 | 0.094 | 0.104 |
| Std. Dev. Δh [m] | 6.316 | 3.977 | 6.741 | 4.328 |
| Mean slope [°] | 9.191 | 8.942 | 11.465 | 11.356 |
| Std. Dev. slope [°] | 6.011 | 5.977 | 5.52988 | 5.531 |
| Mean NASADEM [m] | 367.445 | 320.093 | 506.158 | 449.338 |
| Std. Dev. NASADEM [m] | 395.951 | 381.113 | 400.440 | 360.240 |

It must be noted that the off-glacier areas, especially those featuring a rough topography have not been subject to a thorough outlier masking. Small areas of phase unwrapping error, layover and shadow are still present and might affect the statistics. These are hence to be considered a worst-case scenario compared to the SEC on-glacier.

[Figure]

Figure 1: Slope distribution of NPI (a) and SPI (c). Cumulative slope distribution of NPI (b) and SPI (d).

[Figure]

(a)                                                        (b)

*Figure 2: off-glacier elevation difference for NPI 2000-2012 (a) and NPI 2012-2015 (b). The color scale is the same as in Fig.3.*

[Figure]

*Figure 3: off-glacier elevation difference for the SEC datasets SPI 2000-2012 (a) and SPI 2012-2015 (b).*

**TOPIC "Seasonal correction" (Sect. 3.1.3)**

**Comment of the referee:**

What kind of filtering was applied? It would be interesting to see the original dataset and a Δh map outside the icefields.
Response: Since the Summer 2011/2012 daily SECR is used only for the seasonal correction, we applied the following procedure for eliminating outliers: (i) conservative masking on glaciated terrain of regions with high backscattering and peaks in the daily SECR values followed by (ii) 2-step filtering with sliding 15window: (a) median filters with kernel size 9 and (b) smoothing with kernel size 9. The raster posting is 0.4 arcsec. This way the localized seasonal changes or outliers were eliminated and thus the SECR map can be used for the purpose of compensating the temporal gap in 2015/2016.
→ A shortened description would complement the caption of Fig8. Please consider reproducibility, not only answering to me.

**Response:**

Our response was not intended exclusively for the referee but rather to the general public reading the author's response available online. The caption was changed as suggested by the referee.

**TOPIC "Filtering of outliers" (Sect. 3.1.4)**

**Comment of the referee:**

*Can you please add more information to increase reproducibility when data gets available: what threshold on SEC values? What morphological operators?*
*Response: We did not include these details because we do not think that this is an interesting point and would inflate an already very long paper. For each of the 4 SECR maps we produced a raster starting from the flag mask (FLM) layer that resulted from the processing with ITP which provides roughly the regions affected by layover and shadow. Thresholds h/ t < -10 m/a and > +6 m/a were applied. A morphological operator of closing followed by a 5 x 5 median filter was applied on the mask raster in order to "clean" the mask, avoiding noise due to thresholding.*
→ Eliminating the phrase from the manuscript does not support transparency.
Please integrate answer in manuscript /supplement. Especially the asymmetrical thresholding is interesting. Please provide reasoning for that. A symmetrical cut-off for outlier elimination would be methodically sounder. Executed like provided, statistically appearing (therefore normally distributed) residual noise error resulting in higher/lower rates gets dragged to more negative rates

**Response:**

We added the requested reasoning to the Supplement (in Sect. S3) and referenced it in the manuscript. The asymmetrical thresholds wrongly given in the previous version of the response have been corrected.

**TOPIC "Seasonal correction systematic error" (Sect. 3.3.3)**

**Comment of the referee:**

*According to this paragraph: for interpolated seasonal correction, the last epsilon term should dominate the quadrature sum and thus the total SECR error, if I understand correctly. What does 'increase by a factor of three' mean in this context? Times 3 (\*3) ? I compared SECR uncertainty value for extrapolated glaciers (e.g. Jorge Montt, Bernardo, Tempano) in Tab. 3 with values for not extrapolated glaciers. First ones are not near triple of latter. And they should even be higher than triple, following this paragraph: scaling by year (divided by 0.27. for 99 days for example) is performed as well as a \*1.5 increase for the timespan difference. Please explain where I've gone wrong and/or revise the explanations in this paragraph.*

*Response: Thanks for pointing this out. It seems there is some misunderstanding regarding the seasonal correction and its impact for the retrieval of SECR. The term seasonal correction refers to the difference between mean annual SECR over epochs spanning 12 years (2000 to 2012) and 4 years (2012 to 2016) without accounting for seasonal differences in SEC of the missing days vs. the mean annual SECR taking seasonal differences into account. For the extrapolated glaciers 53 to 103 summer days are missing in order to cover the full 4 year period (1461 days). This means that the missing days to be substituted correspond to 3.6 % to 7.0 % of the 4 year period for which the mean SECR is computed (and not 27 % which would refer to a single year). The impact of missing days to be substituted for the 12 year period I still much smaller. This is now made clear in the revised section 3.1.3 and in Supplement S4.*

→ Thank you for correcting my wrong assumption. It really helped understanding the seasonal correction – as well as the revision of the respective section. Yet, it also emphasized the relatively small impact of the correction to the results. After reviewing the paragraph of the related error estimation again, I now understood correctly what has been done and am sorry about misconceptions. Still I suggest a little step to improve the manuscript.

The paragraph p13ll26 reads:
To compute the systematic error linked to the seasonal correction (Sect. 3.1.3), the previous three systematic error components ($\varepsilon$ reg , $\varepsilon$ pen and $\varepsilon$ add ) were estimated separately for the summer 2011/2012 SECR. Here $\varepsilon$ add was increased by a factor of 1.5 to account for the different temporal coverage. All three components were summed in quadrature and conservatively further increased by a factor of 3.0 on extrapolated regions (north of SPI and NPI). A pixelwise scaling by the number of corrected days and by the appropriate $\Delta t$ in years was applied, leading to a fourth systematic error
The calculation in the cited paragraph is:
(1) $\varepsilon$seas (x, y) = sqrt($\varepsilon$reg² + $\varepsilon$pen² + (1.5\*$\varepsilon$add)² ) \*days)/$\Delta$t
(2) $\varepsilon$seas (x, y) = sqrt($\varepsilon$reg² + $\varepsilon$pen² + (1.5\*$\varepsilon$add)² )\*3 \*days)/$\Delta$t (for extrapolated regions)
I assume. since you involved this error contribution from a seasonal yearly change rate scaled to the

dh error, in this case the Δt=365days. This practice is fine, and I misunderstood it in the first place. That might be, due to the fact, that the phrasing in the last cited sentence indicates, that here a division by entire epoch would appear (Δt=4*356 d or 12*365 d respectively). Latter would be questionable, not representing what has been done, since you subsequently quadrature sum the epsilon components as

(3) $\varepsilon = \mathrm{sqrt}(\varepsilon_{reg}^2 + \varepsilon_{pen}^2 + \varepsilon_{add}^2 + \varepsilon_{seas}^2)/\Delta t$

where the division by entire epoch appears as Δt to give an appropriate error budget for the SECR.

→ Please correct 'and by the appropriate Δt in years' to a phrasing that clarifies Δt to be 365d in this case.

**Response:**

The applied calculation is the following:

$\varepsilon_{seas}''' (x, y)$ **[m]** $= \mathrm{sqrt}(\varepsilon_{reg}^2 + \varepsilon_{pen}^2 + (1.5 \cdot \varepsilon_{add})^2)$

in order to have the error on the elevation change rate we divide this by Δt_summer [d], the number of days separating master and slave of the summer SECR (either 99 or 33 days)

$\varepsilon_{seas}'' (x, y)$ **[m/d]** $= (\mathrm{sqrt}(\varepsilon_{reg}^2 + \varepsilon_{pen}^2 + (1.5 \cdot \varepsilon_{add})^2)) / \Delta t\_summer$ [d]

Now for each pixel we multiply this error on the rate by the number of corrected days Δt_corr [d] for each multiyear dataset, and where extrapolation was performed also multiply it by a factor of 3.0.

$\varepsilon_{seas}' (x, y)$ **[m]** $= (\mathrm{sqrt}(\varepsilon_{reg}^2 + \varepsilon_{pen}^2 + (1.5 \cdot \varepsilon_{add})^2)) / \Delta t\_summer$ [d] $* \Delta t\_corr$ [d]

$\varepsilon_{seas}' (x, y)$ **[m]** $= (\mathrm{sqrt}(\varepsilon_{reg}^2 + \varepsilon_{pen}^2 + (1.5 \cdot \varepsilon_{add})^2)) / \Delta t\_summer$ [d] $* \Delta t\_corr$ [d] $* 3.0$

(for extrapolated regions)

Finally we obtain an error on the yearly rate, like with the other components dividing by the Δt [y].

We slightly modified the text in order to make the applied procedure even clearer.

[revised manuscript text omitted]
 8 $\text{m}\,\text{d}^{-1}$ in March 2011 to 5.9 $\text{m}\,\text{d}^{-1}$ in August 2014 and 4.8 $\text{m}\,\text{d}^{-1}$ in August 2016 (Fig. **??**). This caused a major decrease in the thinning rate of the lower terminus during epoch 2 (Fig. **??**).

The hypsometric curves of Grey and Tyndall glaciers show little change in SECR on the lower terminus close to the calving front and decreasing loss rates in the upper reaches of the terminus and in the accumulation area, an indication for surface mass balance as main cause for the change in SECR (Fig. **??**). This is in line with TerraSAR-X surface velocity results between December 2011 and August 2016 showing only modest changes near the ice front and slowdown upstream. On Tyndall Glacier the velocity on the central flowline 0.5 km from the front was 0.96 $\text{m}\,\text{d}^{-1}$ in December 2011, 0.88 $\text{m}\,\text{d}^{-1}$ in October 2013 and 0.96 $\text{m}\,\text{d}^{-1}$ in August 2016. On Grey Glacier at the central flowline 3 km from the front, where the glacier splits into three branches, the velocity was 1.13 $\text{m}\,\text{d}^{-1}$ in December 2011, 1.11 $\text{m}\,\text{d}^{-1}$ in October 2013 and 1.02 $\text{m}\,\text{
[revised manuscript text omitted]